# Quantitative live-cell imaging and computational modeling shed new light on endogenous WNT/CTNNB1 signaling dynamics

Saskia MA de Man[1], Gooitzen Zwanenburg[2]*, Tanne van der Wal[1], Mark A Hink[3,4]†*, Renée van Amerongen[1]†*

[1]Developmental, Stem Cell and Cancer Biology, Swammerdam Institute for Life Sciences, University of Amsterdam, Amsterdam, Netherlands; [2]Biosystems Data Analysis, Swammerdam Institute for Life Sciences, University of Amsterdam, Amsterdam, Netherlands; [3]Molecular Cytology, Swammerdam Institute for Life Sciences, University of Amsterdam, Amsterdam, Netherlands; [4]van Leeuwenhoek Centre for Advanced Microscopy, Swammerdam Institute for Life Sciences, University of Amsterdam, Amsterdam, Netherlands

**\*For correspondence:**
Gooitzen.zwanenburg@gmail.com (GZ);
m.a.hink@uva.nl (MAH);
r.vanamerongen@uva.nl (RVA)

†These authors contributed equally to this work

**Competing interests:** The authors declare that no competing interests exist.

**Abstract** WNT/CTNNB1 signaling regulates tissue development and homeostasis in all multicellular animals, but the underlying molecular mechanism remains incompletely understood. Specifically, quantitative insight into endogenous protein behavior is missing. Here, we combine CRISPR/Cas9-mediated genome editing and quantitative live-cell microscopy to measure the dynamics, diffusion characteristics and absolute concentrations of fluorescently tagged, endogenous CTNNB1 in human cells under both physiological and oncogenic conditions. State-of-the-art imaging reveals that a substantial fraction of CTNNB1 resides in slow-diffusing cytoplasmic complexes, irrespective of the activation status of the pathway. This cytoplasmic CTNNB1 complex undergoes a major reduction in size when WNT/CTNNB1 is (hyper)activated. Based on our biophysical measurements, we build a computational model of WNT/CTNNB1 signaling. Our integrated experimental and computational approach reveals that WNT pathway activation regulates the dynamic distribution of free and complexed CTNNB1 across different subcellular compartments through three regulatory nodes: the destruction complex, nucleocytoplasmic shuttling, and nuclear retention.

## Introduction

WNT signaling is one of the most ancient pattern-forming cell signaling cascades. It drives many biological processes from the onset of embryogenesis until adulthood in all multicellular animals (reviewed in *van Amerongen and Nusse, 2009*; *Holstein, 2012*; *Loh et al., 2016*). WNT signaling remains important throughout the lifespan of the organism and controls stem cell maintenance in many mammalian tissues, including the intestine (*Barker et al., 2007*). Disruption of the pathway causes disease, with hyperactivation being a frequent event in human colorectal and other cancers (reviewed in *Nusse and Clevers, 2017*; *Wiese et al., 2018*).

The key regulatory event in WNT/CTNNB1 signaling (traditionally known as 'canonical WNT signaling') is the accumulation and nuclear translocation of the transcriptional co-activator β-catenin (Catenin beta-1, hereafter abbreviated as CTNNB1) (*Figure 1A*). In the absence of WNT signaling, rapid turnover by the so-called destruction complex maintains low levels of CTNNB1. This cytoplasmic complex consists of the scaffold proteins Adenomatous Polyposis Coli Protein (APC) and

**Figure 1.** Generation of HAP1$^{SGFP2-CTNNB1}$ cell lines. (**A**) Cartoon depicting the current model of the WNT/CTNNB1 pathway. In the absence of WNT ligands (left, 'OFF'), free cytoplasmic CTNNB1 is captured by the destruction complex consisting of AXIN, APC, CSNK1A1, and GSK3, which leads to its phosphorylation, BTRC-mediated ubiquitination and subsequent proteasomal degradation, resulting in low levels of CTNNB1 in the cytoplasm and nucleus. Binding of the WNT protein (right, 'ON') to the FZD and LRP receptors inhibits the destruction complex through DVL. CTNNB1 accumulates in the cytoplasm and subsequently translocates to the nucleus, where it promotes the transcription of target genes, such as *AXIN2*, as a co-activator of TCF/LEF transcription factors. (**B**) Cartoon depicting exon 2 of the *CTNNB1* locus, which contains the start codon, and the CTNNB1 protein before (top) and after (bottom) introduction of the SGFP2 by CRISPR/Cas9-mediated homology directed repair. (**C**) Schematic of the experimental workflow and timeline for generating HAP1$^{SGFP2-CTNNB1}$ clones. Cas9, gRNA and repair templates are transfected as plasmids. The repair template contains the coding sequence of SGFP2 surrounded by 800 bp homology arms on either side and lacks the gRNA recognition site (see supplement 2 of this figure). A short puromycin selection step is included from 24 to 48 hr after transfection to enrich for transfected cells. Haploid, GFP-positive cells are sorted, and single cell clones are expanded for further analysis. (**D–F**) FACS plots illustrating control (**D**) and SGFP2-CTNNB1-tagged cells (**E–F**). (**D**) Cells

*Figure 1 continued*

transfected with Cas9 and gRNA in the absence of a repair template were used to set the gate for SGFP2-positive events. (E) A small population of cells expressing low levels of SGFP2 can be detected when cells are transfected with Cas9, gRNA, and repair template. (F) Treatment for 24 hr of cells similar to those depicted in (D) with 8 µM CHIR99021 does not change the amount of cells that are SGFP2 positive, but increases the SFP2 signal, most likely reflecting an increase in SGFP2-tagged beta catenin levels on a per cell basis and supporting the notion that the gated events indeed represent successfully tagged cells.

The online version of this article includes the following figure supplement(s) for figure 1:

**Figure supplement 1.** *SGFP2-CTNNB1* locus.

**Figure supplement 2.** FACS Gating strategy for haploid HAP1 cells.

Axis Inhibition Proteins 1 and 2 (AXIN), which bind CTNNB1, and the serine/threonine kinases Casein kinase I isoform alpha (CSNK1A1) and Glycogen Synthase Kinase-3 alpha and beta (GSK3), which subsequently phosphorylate residues S45, T41, S37, and S33 (*Amit et al., 2002*; *Liu et al., 2002*). This primes CTNNB1 for ubiquitination by E3 Ubiquitin Protein Ligase beta-TrCP 1 and 2 (BTRC and FBXW11) and subsequent proteasomal degradation (*Aberle et al., 1997*; *Latres et al., 1999*). In the current working model for WNT/CTNNB1 signaling, binding of WNT ligands to the Frizzled (FZD) and low-density lipoprotein receptor-related protein 5 and 6 (LRP) receptor complex sequesters and inhibits the destruction complex at the membrane in a process that involves Disheveled (DVL) (*Bilic et al., 2007*; *Kan et al., 2020*; *Ma et al., 2020*; *Schwarz-Romond et al., 2007*). This allows newly synthesized CTNNB1 to accumulate and translocate to the nucleus, where CTNNB1 binds to TCF/LEF transcription factors (TCF7, TCF7L1, TCF7L2, and LEF1) to regulate target gene transcription as part of a larger transcriptional complex (*Behrens et al., 1996*; *Fiedler et al., 2015*; *Molenaar et al., 1996*; *van Tienen et al., 2017*).

The working model for WNT/CTNNB1 signaling described above is the result of almost 40 years of research. The use of traditional genetic and biochemical approaches has allowed identification of the core players, as well as dissection of the main signaling events. However, multiple aspects of WNT/CTNNB1 signaling remain poorly understood. For instance, the exact molecular composition of the destruction complex as well as the mechanism for its inhibition remain unclear (reviewed in *Tortelote et al., 2017*). How WNT/CTNNB1 signaling regulates the subcellular distribution of CTNNB1 also requires further scrutiny.

Most biochemical techniques lead to loss of spatial information and averaging of cell-to-cell heterogeneity, since proteins are extracted from their cellular context. Additionally, temporal information is usually limited to intervals of several minutes or hours. Live-cell microscopy offers better spatiotemporal resolution. Currently, however, many of these studies are conducted by overexpressing the protein(s) of interest. This can severely affect activation, localization, and complex formation (*Gibson et al., 2013*; *Mahen et al., 2014*). Although stabilization of CTNNB1 by WNT signaling has been extensively studied, very few studies have focused on the spatiotemporal dynamics of this process – especially at the endogenous level (*Chhabra et al., 2019*; *Massey et al., 2019*; *Rim et al., 2020*).

Here, we use CRISPR/Cas9-mediated genome editing in haploid cells to generate clonal cell lines that express fluorescently tagged CTNNB1. Using confocal imaging and automated cell segmentation, we quantify the dynamic subcellular increase of endogenous CTNNB1 upon WNT stimulation. Moreover, using Fluorescence Correlation Spectroscopy (FCS) and Number and Brightness (N and B) analysis, we measure the mobility and concentration of CTNNB1, providing detailed information on CTNNB1 containing complexes in the cytoplasm and nucleus. Next, we use these biophysical parameters to build a computational model of WNT/CTNNB1 signaling that predicts the levels and subcellular distribution of CTNNB1 across its cytoplasmic and nuclear pools. Using this integrated experimental and computational approach, we find that WNT regulates the dynamic distribution of CTNNB1 across different functional pools by modulating three regulatory nodes: cytoplasmic destruction, nucleocytoplasmic shuttling, and nuclear retention. Finally, we strengthen the link between our data and the model via specific experimental perturbations, which shows that the regulatory nodes responsible for nuclear retention and nuclear shuttling of CTNNB1 are equally important under physiological and oncogenic conditions.

# Results

## Generation and functional validation of clonal HAP1<sup>SGFP2-CTNNB1</sup> cell lines

To visualize and quantify the spatiotemporal dynamics of WNT/CTNNB1 signaling at the endogenous level, we fluorescently tagged CTNNB1 in mammalian cells using CRISPR/Cas9-mediated homology directed repair (*Ran et al., 2013*; *Figure 1*). To preserve the existing (epi)genetic control mechanisms of *CTNNB1* expression, only the coding sequence for SGFP2, a monomeric, bright and photostable green fluorescent protein (*Kremers et al., 2007*), was seamlessly inserted at the starting ATG of the *CTNNB1* coding sequence in HAP1 cells, a WNT-responsive near haploid cell line (*Figure 1B*, *Figure 1—figure supplement 1A*; *Andersson et al., 1987*; *Carette et al., 2011*; *Kotecki et al., 1999*; *Lebensohn et al., 2016*). The choice for this haploid cell line ensured tagging of the complete CTNNB1 protein pool (*Figure 1C*), thus overcoming the limitations of polyploid cell lines where genome editing often results in a combination of correctly and incorrectly edited alleles (*Canaj et al., 2019*).

We isolated clonal cell lines with the desired modification by FACS sorting (*Figure 1D–F*) with a gating strategy that specifically selected for haploid cells (*Figure 1—figure supplement 2*), since HAP1 cells can become diploid or polyploid over time (*Essletzbichler et al., 2014*; *Yaguchi et al., 2018*). Genome editing of wild-type HAP1 (HAP1<sup>WT</sup>) cells resulted in a small population with low SGFP2 fluorescence (0.2%) (*Figure 1D–E*). The intensity, but not the number of cells in this population increased upon treatment with CHIR99021, a potent and selective GSK3 inhibitor (*Bain et al., 2007*), providing a strong indication that these fluorescent events corresponded to HAP1 cells in which the *SGFP2* sequence was successfully knocked into the endogenous *CTNNB1* locus (HAP1<sup>SGFP2-CTNNB1</sup>) (*Figure 1F*). While scarless tagging of endogenous genes in HAP1 cells was relatively cumbersome (only 0.2% gated events), PCR-based screening and sanger sequencing revealed that the desired repair occurred with almost 90% efficiency within this population (*Figure 1—figure supplement 1*).

To verify that the SGFP2 tag did not interfere with CTNNB1 function, three clonal HAP1<sup>SGFP2-CTNNB1</sup> cell lines were further characterized using established experimental readouts for WNT/CTNNB1 signaling (*Figure 2* and *Figure 2—figure supplement 1*). Western blot analysis confirmed that the HAP1<sup>SGFP2-CTNNB1</sup> clones did not contain any untagged CTNNB1 but only expressed the SGFP2-CTNNB1 fusion protein (*Figure 2A*). Moreover, the total levels of SGFP2-CTNNB1 in tagged cell lines increased to the same extent as wild-type CTNNB1 in untagged cells in response to CHIR99021 treatment (*Figure 2A–B*). Similarly, untagged and tagged CTNNB1 induced target gene expression in equal measure, as measured by a TCF/LEF responsive luciferase reporter (*Hu et al., 2007*; *Figure 2C*), and increased transcription of the universal WNT/CTNNB1 target *AXIN2* (*Lustig et al., 2002*; *Figure 2D*). Finally, while unstimulated cells mainly showed SGFP2-CTNNB1 localization at adherens junctions, treatment with purified WNT3A protein (*Figure 2E*) and CHIR99021 (*Figure 2—figure supplement 1E*) increased SGFP2-CTNNB1 levels in the cytoplasm and nucleus consistent with its signaling function.

Taken together, WNT-responsive changes in CTNNB1 levels, localization and activity are preserved after CRISPR/Cas9-mediated fluorescent tagging of the entire CTNNB1 protein pool. Although there is some variation between the three clones with respect to CTNNB1 stabilization and target gene activation, this is likely due to the sub-cloning of these cell lines rather than the targeting per se.

## Live imaging of endogenous SGFP2-CTNNB1 during WNT pathway activation

To better understand the temporal dynamics of endogenous CTNNB1 stabilization, we performed live-cell imaging over 12 hr in HAP1<sup>SGFP2-CTNNB1</sup> clone 2 (*Figure 3*, *Video 1*; *Video 2*; *Video 3*) with different levels of WNT stimulation. Unstimulated cells showed a stable CTNNB1 signal at the cell membrane throughout the imaging time course (*Figure 3A*, *Video 1*). The membrane localization of CTNNB1 is consistent with its structural role in adherens junctions (*Valenta et al., 2012*; *Yap et al., 1997*), which we will not consider further in the current study. Stimulation with different concentrations of purified WNT3A resulted in a heterogeneous response pattern, with some cells in the

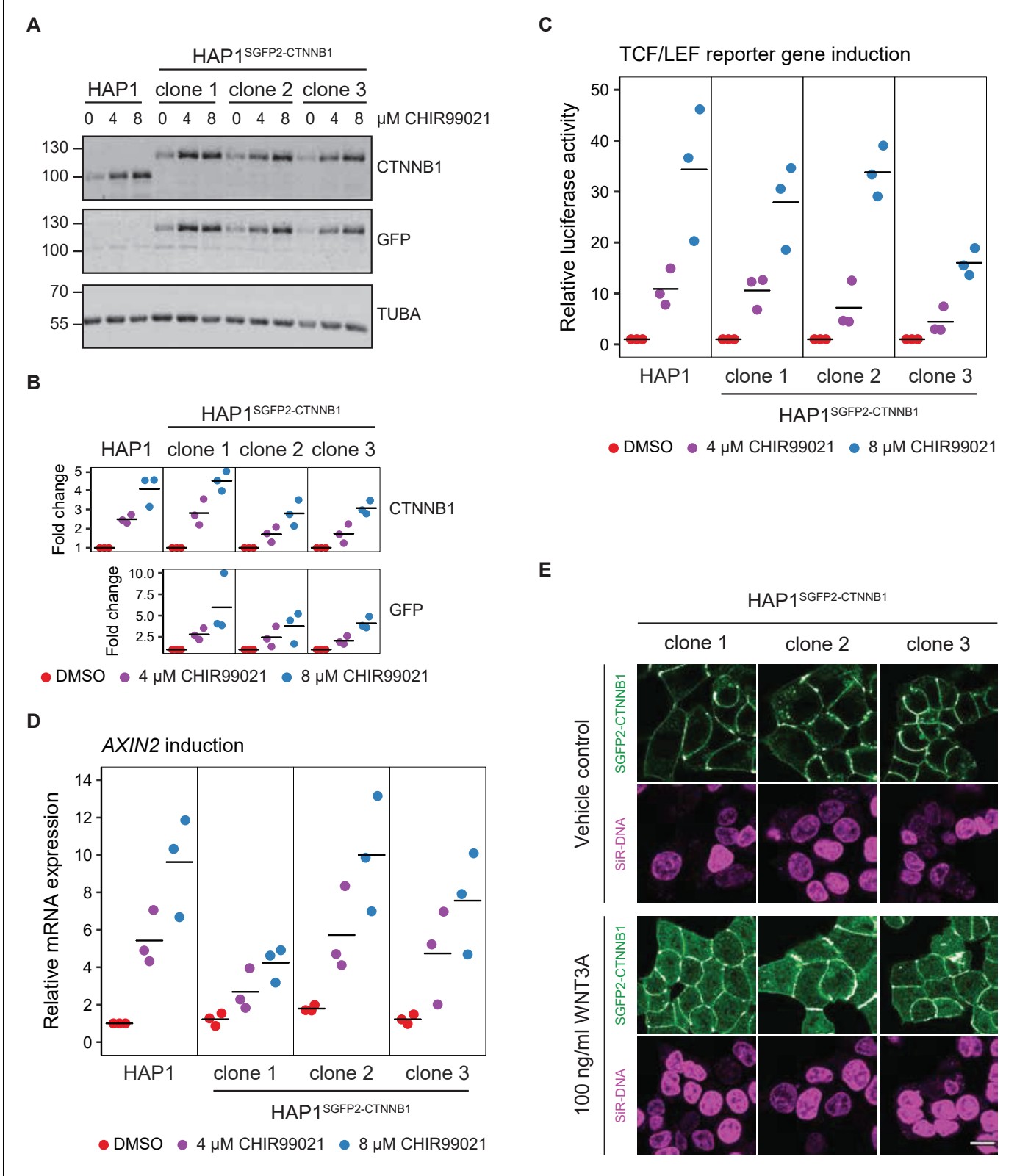

**Figure 2.** Functional validation of three independent HAP1^SGFP2-CTNNB1 clones. (**A**) Western blot, showing CTNNB1 (HAP1^WT) and SGFP2-CTNNB1 (HAP1^SGFP2-CTNNB1 clones 1, 2, and 3) accumulation in response to CHIR99021 treatment. All panels are from one blot that was cut at the 70 kDa mark and was stained with secondary antibodies with different fluorophores for detection. Top: HAP1^WT cells express CTNNB1 at the expected wild-type size. Each of the three clonal HAP1^SGFP2-CTNNB1 cell lines only express the larger, SGFP2-tagged form of CTNNB1, that runs at the expected height (~27

*Figure 2 continued on next page*

*Figure 2 continued*

kDa above the wild-type CTNNB1). Middle: Only the tagged clones express the SGFP2-CTNNB1 fusion protein, as detected with an anti-GFP antibody at the same height. Bottom: alpha-Tubulin (TUBA) loading control. A representative image of n=3 independent experiments is shown. (B) Quantification of Western blots from n=three independent experiments, including the one in (A), confirming that the accumulation of CTNNB1 in response to WNT/CTNNB1 pathway activation is comparable between HAP1WT and HAP1SGFP2-CTNNB1 cells. Horizontal bar indicates the mean. (C) Graph depicting the results from a MegaTopflash dual luciferase reporter assay, showing comparable levels of TCF/LEF reporter gene activation for HAP1WT and HAP1SGFP2-CTNNB1 cells in response to CHIR99021 treatment. Data points from n=3 independent experiments are shown. Horizontal bar indicates the mean. Values are depicted relative to the DMSO control, which was set to one for each individual cell line. (D) Graph depicting *AXIN2* mRNA induction in response to CHIR99021 treatment, demonstrating that induced expression of an endogenous target gene is comparable between HAP1WT and HAP1SGFP2-CTNNB1 cells. Data points represent n=3 independent experiments. Horizontal bar represents the mean. *HPRT* was used as a reference gene. Values are depicted relative to the HAP1WT DMSO control, which was set to 1. (E) Representative confocal microscopy images of the three HAP1 SGFP2-CTNNB1 clones after 4 hr vehicle control or 100 ng/ml WNT3A treatment from n=1 biological experiment, revealing intracellular accumulation of SGFP2-CTNNB1 (green). Nuclei were counterstained with SiR-DNA dye (magenta). Scale bar is 10 μm.

The online version of this article includes the following source data and figure supplement(s) for figure 2:

**Source data 1.** Numerical data for *Figure 2B,C,D* and *Figure 2—figure supplement 1A,C,D*.

**Figure supplement 1.** Verification of the WNT/CTNNB1 responsiveness of HAP1 cells.

population showing a far more prominent increase in CTNNB1 levels in the cytoplasm and nucleus than others (*Figure 3A*, *Figure 3—figure supplement 1A–B*, *Video 2*).

To quantify these dynamic changes, we developed a custom-built automated segmentation pipeline in CellProfiler (*Figure 3D*). Quantification showed that the temporal dynamics of CTNNB1 accumulation were independent of the dose of WNT3A (*Figure 3B–C*, *Videos 4–5*), although this could be different for lower doses and other cell types (*Massey et al., 2019*). Treatment with 100 ng/ml WNT3A increased SGFP2-CTNNB1 fluorescence 1.74-fold (mean, 95% CI 1.73–1.76) in the cytoplasm and 3.00-fold (mean, 95% CI 2.97–3.03) in the nucleus, with similar results in the other two HAP1SGFP2-CTNNB1 clones (*Figure 3—figure supplement 2*).

Our quantification further shows that nuclear accumulation of CTNNB1 is favored over a cytoplasmic increase (compare the fold-changes in *Figure 3B–C*). Moreover, the first statistically significant increases in fluorescence intensity in the cytoplasm could be detected after ~45 min of treatment (*Video 4*, *Figure 3—figure supplement 1C*), whereas in the nucleus an increase was first statistically significant after ~30 min (*Video 5*, *Figure 3—figure supplement 1D*). To examine the relation between the cytoplasmic and nuclear CTNNB1 pools more closely, we calculated the ratio between nuclear and cytoplasmic intensities of SGFP2-CTNNB1 (*Figure 3D*, *Video 6*). In untreated cells, the nuclear/cytoplasmic ratio was 0.652 (mean [3–5 hr], 95% CI 0.649–0.657), showing that SGFP2-CTNNB1 was preferentially localized to the cytoplasm (*Figure 3D*, *Figure 3—figure supplement 3*). For the first 3 hr after WNT3A, nuclear CTNNB1 levels rose considerably faster than cytoplasmic CTNNB1 levels until the nuclear/cytoplasmic ratio showed a slight nuclear enrichment of 1.08 (mean [3–5 hr] 95% CI 1.07–1.10) for 100 ng/ml WNT3A. This indicates that not only the turnover, but also the subcellular localization of CTNNB1 is actively regulated both before and after WNT pathway activation.

## Establishing a fitting model for SGFP2-CTNNB1 diffusion

Having measured the relative changes in the cytoplasmic and nuclear levels of CTNNB1 in response to WNT3A stimulation, we next sought to exploit our experimental system to quantify additional molecular properties of CTNNB1 in each of these subcellular compartments using Fluorescence Correlation Spectroscopy (FCS). FCS is a powerful method to measure the mobility and absolute levels of fluorescent particles in a nanomolar range, compatible with typical levels of signaling proteins in a cell (reviewed in *Hink, 2014*). It has for instance been used to gain insight into the assembly of DVL3 supramolecular complexes (*Yokoyama et al., 2012*), the endogenous concentrations and mobility of nuclear complexes (*Holzmann et al., 2019*; *Lam et al., 2012*), and most recently, to quantify ligand-receptor binding reactions in the WNT pathway (*Eckert et al., 2020*). In point FCS, the fluorescence intensity is measured in a single point (*Figure 4A,D–E*). Diffusion of labeled particles, in this case SGFP2-CTNNB1, causes fluctuation of the fluorescence signal over time (*Figure 4B*). By correlating the fluorescence intensity signal to itself over increasing time-intervals, an autocorrelation curve is generated (*Figure 4C*). To extract relevant biophysical parameters, such as mobility (a measure for

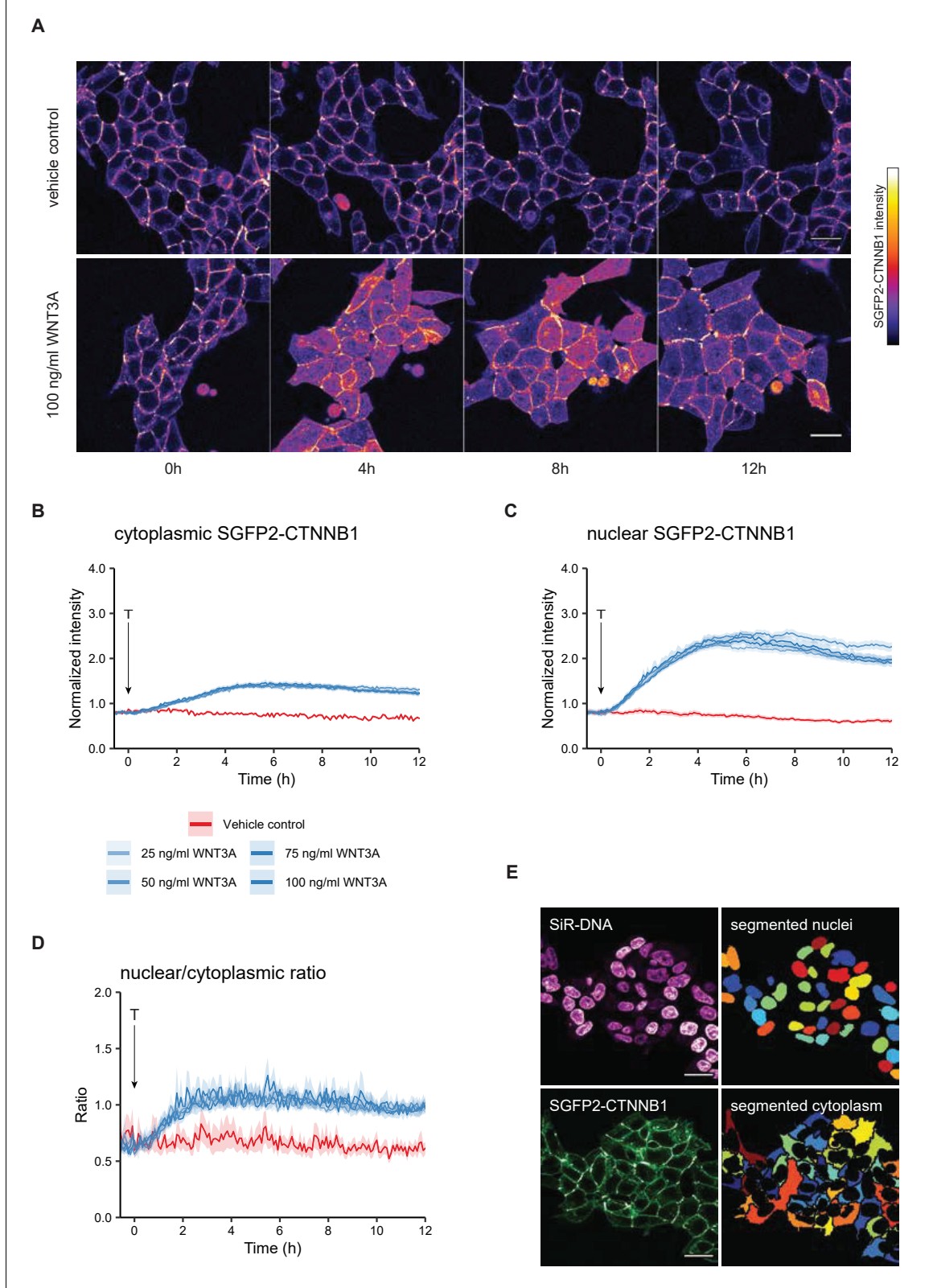

**Figure 3.** Live imaging of HAP1SGFP2-CTNNB1. (**A**) Representative stills from confocal time-lapse experiments corresponding to *Videos 1–2*, showing an increase of SGFP2-CTNNB1 after treatment with 100 ng/ml WNT3A (bottom) relative to a vehicle control (BSA)-treated sample (top). Scale bar = 20 µm. (**B–D**) Quantification of time-lapse microscopy series, using the segmentation pipeline shown in (**E**). Arrow indicates the moment of starting the different treatments (T, see legend in B for details). (**B–C**) Graph depicting the normalized intensity of SGFP2-CTNNB1 in the cytoplasm (**B**) or nucleus

*Figure 3 continued on next page*

*Figure 3 continued*

(C) over time. Solid lines represent the mean normalized fluorescence intensity and shading indicates the 95% confidence interval. n=155–393 cells for each condition and time point, pooled data from n=three independent biological experiments. (D) Graph depicting the nuclear/cytoplasmic ratio of SGFP2-CTNNB1 over time, calculated from raw intensity values underlying (B) and (C). (E) Segmentation of nuclei (top) and cytoplasm (bottom) based on the SiR-DNA signal and SGFP2-CTNNB1 signal. Scale bar = 20 μm.

The online version of this article includes the following figure supplement(s) for figure 3:

**Figure supplement 1.** Difference analysis of SGFP2-CTNNB1 fluorescence.

**Figure supplement 2.** Graphs showing quantification of time-lapse microscopy experiments with three independent HAP1[SGFP2-CTNNB1] clones.

**Figure supplement 3.** Unnormalized nuclear and cytoplasmic intensity measurements.

size) and the absolute numbers of the fluorescent particles (corresponding to their concentration), this autocorrelation curve is fitted with an appropriate model.

We first attempted to fit the autocorrelation curves obtained with point FCS measurements in HAP1[SGFP2-CTNNB1] cells with a one-component model (i.e. containing one single diffusion speed for SGFP2-CTNNB1). This model was unable to fit most of our data (*Figure 4F*). The current literature suggests that while a large portion of CTNNB1 is present as a monomer (*Gottardi and Gumbiner, 2004*; *Maher et al., 2010*), CTNNB1 is also present in multiprotein complexes in the cytoplasm and in the nucleus (reviewed in *Gammons and Bienz, 2018*). We therefore tested the fit of a two-component model. To this end, we deduced the theoretical diffusion speed of monomeric, unbound SGFP2-CTNNB1 to be 14.9 μm²/s. This theoretical speed was confirmed by fitting an unbiased two-component model to our experimental data (*Figure 4—figure supplement 1*). To limit variability due to noise in the measurements, we proceeded with the two component model in which the first diffusion component was fixed to the theoretically determined diffusion speed of monomeric SGFP2-CTNNB1 (14.9 μm²/s) and with the second diffusion component limited to slower speeds compatible with point-FCS imaging (see Materials and methods for details). This model provided good fits for our autocorrelation curves obtained in both cytoplasmic and nuclear point FCS measurements (*Figure 4G*). Together this is consistent with the presence of free monomeric CTNNB1 (first, fast component) and larger CTNNB1 containing complexes (second, slow component) in both the nucleus and cytoplasm.

## Quantification of absolute SGFP2-CTNNB1 concentrations

Using this fitting model, we determined, for the first time, the absolute concentrations of endogenous CTNNB1 in living cells in presence and absence of a physiological WNT stimulus (*Figure 5A*, *Table 1*). In the absence of WNT3A, we determined the total concentration of SGFP2-CTNNB1 to be 180 nM (median, 95% CI 127–218) in the cytoplasm and 122 nM (median, 95% CI 91–158) in the nucleus. This is consistent with the nuclear exclusion we observed with confocal imaging (*Figure 3*).

In the presence of WNT3A, we measured a 1.2-fold increase in the total SGFP2-CTNNB1 concentration to 221 nM (median, 95% CI 144–250 nM) in the cytoplasm. This increase was smaller than expected from fluorescence intensity measurements (*Figure 3B*). We excluded that this was caused by photo-bleaching and other photophysical effects, and currently have no explanation for this discrepancy (*Figure 5—figure supplement 1*, see also Materials and methods and discussion). In the nucleus the concentration increased 2.0-fold to 240 nM (median, 95% CI 217–325) upon pathway activation. Nuclear concentrations of SGFP2-CTNNB1 therefore exceed cytoplasmic concentrations after WNT3A treatment, consistent with the nuclear accumulation observed with live imaging (*Figure 3*). These concentrations are in a similar range as those previously determined by quantitative mass spectrometry in different

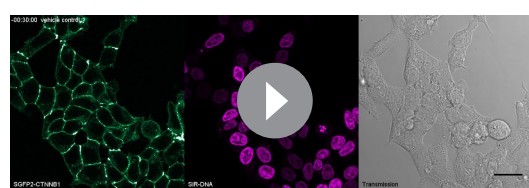

**Video 1.** Representative video of confocal time-lapse experiments, showing SGFP2-CTNNB1 (left, green), SiR-DNA staining (middle, magenta), and transmission image (right, gray) after treatment with vehicle control (BSA). Time of addition is at 00:00:00 (indicated at the top left). Scale bar in the lower right represents 20μm. https://elifesciences.org/articles/66440#video1

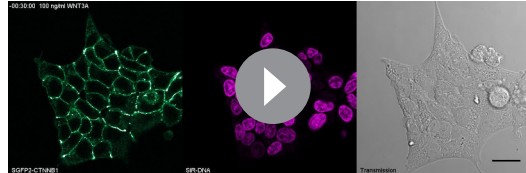

**Video 2.** Representative video of confocal time-lapse experiments, showing SGFP2-CTNNB1 (left, green), SiR-DNA staining (middle, magenta), and transmission image (right, gray) after treatment with 100 ng/ml WNT3A. Time of addition is at 00:00:00 (indicated at the top left). Scale bar in the lower right represents 20μm.
https://elifesciences.org/articles/66440#video2

**Video 3.** Representative video of confocal time-lapse experiments, showing SGFP2-CTNNB1 (left, green), SiR-DNA staining (middle, magenta), and transmission image (right, gray) after treatment 8 μM CHIR99021. Time of addition is at 00:00:00 (indicated at the top left). Scale bar in the lower right represents 20μm.
https://elifesciences.org/articles/66440#video3

mammalian cell lines (*Kitazawa et al., 2017*; *Tan et al., 2012*). Of note, the exact concentrations will likely vary between cell types and their calculated values may also be dependent on the intricacies and assumptions that underlie each individual measurement technique.

Our two-component fitting model also allowed us to discriminate between pools of SGFP2-CTNNB1 with different mobility (*Table 2*), that is fast diffusing monomeric CTNNB1 (*Figure 5B*) and slow diffusing complexed CTNNB1 (*Figure 5C*). In the nucleus, the concentration of fast moving CTNNB1 increased 2.0-fold from 87 nM (median, 95% CI 78–119) to 170 nM (median, 95% CI 147–214), while slow moving CTNNB1 concentration increased 3.9-fold from 22 nM (median, 95% CI 4–40) to 86 nM (median, 95% CI 67–114). This is also reflected by the increase in the bound fraction of SGFP2-CTNNB1 the nucleus (*Figure 5D*). The preferential increase of the slow-moving fraction is consistent with the notion that upon WNT stimulation CTNNB1 will become associated with the chromatin in a TCF-dependent transcriptional complex (or 'WNT enhanceosome').

Of note, in the cytoplasm, the concentration of both fast and slow SGFP2-CTNNB1 increased upon WNT3A treatment (*Figure 5B–C*), with the fraction of bound SGFP2-CTNNB1 remaining equal between stimulated (median 0.38, 95% CI 0.29–0.46) and unstimulated cells (median 0.34, 95% CI 0.31–0.4) (*Figure 5D*). The fact that a large portion of CTNNB1 remains in a complex after WNT stimulation, challenges the notion that mainly monomeric CTNNB1 accumulates, as commonly depicted in the textbook model (*Figure 1A*).

## Quantification of SGFP2-CTNNB1 mobility

While we cannot determine the exact composition of the SGFP2-CTNNB1 complex, we do obtain biophysical parameters that are linked to its size. For instance, the diffusion coefficient of the nuclear SGFP2-CTNNB1 complex was 0.17 $\mu m^2 s^{-1}$ (median, 95% CI 0.14–0.22) in cells treated with purified WNT3A (*Figure 5E*). This is comparable to the diffusion coefficients measured for other chromatin-bound transcriptional activators (*Lam et al., 2012*), which further

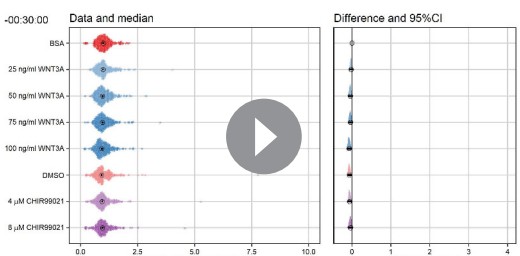

**Video 4.** Video showing the quantification of the normalized intensity of SGFP2-CTNNB1 in the cytoplasm of time-lapse microscopy series (from *Figure 4* and *Videos 1–3*) at each time point showing all individual cells from three biological experiments. Time of addition of the indicated substances is at 00:00:00 (indicated at the top left). The left graph represents the raw data (colored dots, each dot is one cell, n=155–400 cells for each condition and time point), the median (black circle) and the 95% CI of the median (black bar). The right graph represents the median difference (black circle) from the treatments to the control (BSA). When the 95% CI (black bar) does not overlap 0, the difference between the two conditions is significant. Significant changes in intensity can first be observed after 40 min of 8 μM CHIR99021, and after 70–80 min of 4 μM CHIR99021 or 25–100 ng/ml WNT3A treatment.
https://elifesciences.org/articles/66440#video4

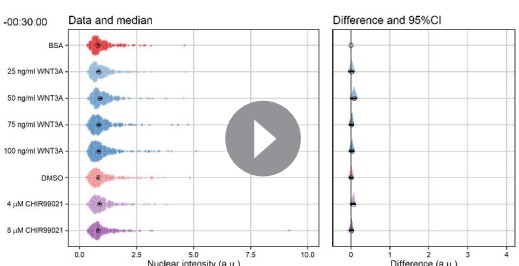

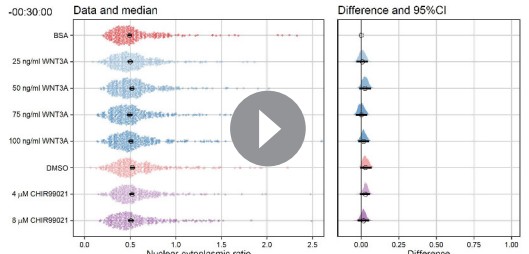

**Video 5.** Video showing the quantification of of the normalized intensity of SGFP2-CTNNB1 in the nucleus of time-lapse microscopy series (from *Figure 4* and *Videos 1–3*) at each time point showing all individual cells from three biological experiments. Time of addition of the indicated substances is at 00:00:00 (indicated at the top left). The left graph represents the raw data (colored dots, each dot is one cell, n=155–400 cells for each condition and time point), the median (black circle) and the 95% CI of the median (black bar). The right graph represents the median difference (black circle) from the treatments to the control (BSA). When the 95% CI (black bar) does not overlap 0, the difference between the two conditions is significant. Significant changes in intensity can be observed for all treatments (but not controls) after 20–50 min.

https://elifesciences.org/articles/66440#video5

**Video 6.** Video showing the quantification of the nuclear-cytoplasmic ratio of SGFP2-CTNNB1, calculated from raw intensity values underlying *Videos 4* and *5*. At each time point showing all individual cells from three biological experiments. Time of addition of the indicated substances is at 00:00:00 (indicated at the top left). The left graph represents the raw data (colored dots, each dot is one cell, n=155–400 cells for each condition and time point), the median (black circle) and the 95% CI of the median (black bar). The right graph represents the median difference (black circle) from the treatments to the control (BSA). When the 95% CI (black bar) does not overlap 0, the difference between the two conditions is significant. Significant changes in the nuclear-cytoplasmic ratio can be observed for all treatments (but not controls) after 20–50 min.

https://elifesciences.org/articles/66440#video6

supports that this pool represents the WNT enhanceosome.

In the cytoplasm, we determined the second diffusion coefficient of SGFP2-CTNNB1 to be 0.13 $\mu m^2 s^{-1}$ (median, 95% CI 0.13–0.17) in the absence of WNT3A stimulation (*Figure 5E*). This is indicative of very large complexes containing SGFP2-CTNNB1 that move with diffusion kinetics comparable to those previously observed for the 26S proteasome (*Pack et al., 2014*). Of note, the speed of the cytoplasmic complex increased 3.5-fold to 0.46 $\mu m^2 s^{-1}$ (95% CI of the median 0.37–0.57) after WNT3A treatment. Because changes in diffusion coefficient are typically indicative of much larger changes in molecular weight (i.e. three-dimensional protein complex size, see Materials and methods section FCS data acquisition and analysis for details), this indicates that the size of the cytoplasmic CTNNB1 complex drastically changes when the WNT pathway is activated. Thus, although the fraction of CTNNB1 that resides in a complex remains the same (34–38%), the identity of the cytoplasmic complex is quite different in unstimulated and WNT3A-stimulated cells.

## Determining the multimerization status of SGFP2-CTNNB1

Recent work suggests that the CTNNB1 destruction complex (also known as the 'degradosome') is a large and multivalent complex, mainly as the result of AXIN and APC multimerization (reviewed in *Schaefer and Peifer, 2019*). The speed of the slow CTNNB1 component, determined by the second diffusion coefficient in our FCS measurements (*Figure 5E*), is consistent with this model. Such a large, multivalent destruction complex would be expected to have multiple CTNNB1-binding sites. To measure the number of bound SGFP2-CTNNB1 molecules within this cytoplasmic complex, we performed Number and Brightness (N and B) analysis (*Figure 5F–G*, *Figure 5—figure supplement 2*). N and B is a fluorescence fluctuation spectroscopy technique similar to point FCS, but it makes use of image stacks acquired over time rather than individual point measurements (*Digman et al., 2008*). By quantifying the variance in fluorescence intensity of this stack, not only the number of particles but also their brightness can be determined.

Because brightness is an inherent property of a fluorophore, a change in brightness is a measure of the number of fluorophores per particle. In our case, the brightness is indicative of the number of SGFP2-CTNNB1 molecules per complex. As N and B does not incorporate diffusion kinetics, we

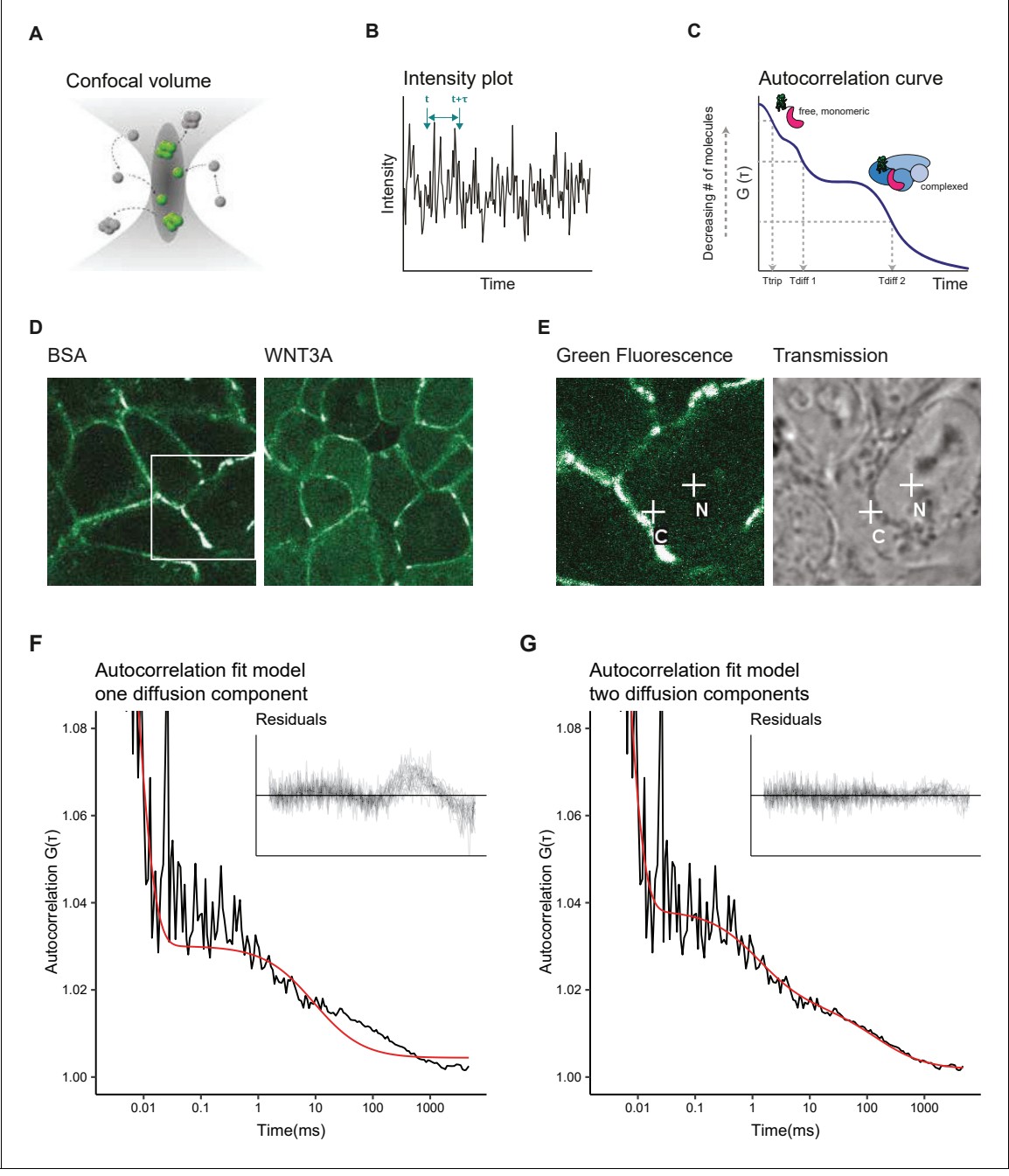

**Figure 4.** Two diffusion-component fit-model for SGFP2-CTNNB1 FCS measurements. (**A**) Schematic representation of the point FCS technique, depicting the confocal volume with fluorescent particles diffusing in and out. Particles in FCS are defined by their coherent movement; therefore, a particle can be made up of monomers or multimers in isolation or complexed to unlabeled molecules. (**B**) Schematic representation of intensity fluctuations over time as measured in the confocal volume. Fluctuations are the result of both photo-physics (e.g. blinking of the fluorophore), diffusion and the number of particles in the confocal volume. (**C**) Graphical representation of the two diffusion-component fitting model used for our autocorrelation curves. $T_{trip}$ describes the blinking of the SGFP2 fluorophore and the after-pulsing artefact. $T_{diff1}$ and $T_{diff2}$ describe the monomeric and complexed form of SGFP2-CTNNB1, respectively. Details of all fitting parameters are described in Materials and methods. (**D**) Representative confocal images of HAP1$^{SGFP2-CTNNB1}$ cells treated for 4 hr with BSA (left) or 100 ng/ml WNT3A (right). (**E**) Zoom in of the white rectangle in (**D**), with representative locations of FCS measurement points for cytoplasm (C) and nucleus (N) indicated with white crosses in the SGFP2-CTNNB1 channel and transmission channel. (**F–G**) Fitting of a representative autocorrelation curve with one unfixed diffusion-component (**F**) or a two diffusion-component model (**G**), where the first diffusion component was fixed to the speed of free monomeric SGFP2-CTNNB1 (14.9 $\mu m^2$/s) and the second diffusion component was unfixed. The black line represents the autocorrelation curve generated from the FCS measurement; the red line represents the fitted model. The residuals after fitting of 25 individual curves are shown in the upper right corner of the graphs.

*Figure 4 continued on next page*

*Figure 4 continued*

The online version of this article includes the following source data and figure supplement(s) for figure 4:

**Source data 1.** Numerical data for *Figure 4F,G* and *Figure 4—figure supplement 1*.

**Figure supplement 1.** Quantification of SGFP2-CTNNB1 fast component using an unfixed two diffusion component model.

cannot differentiate between monomeric (which would have a brightness of one) and complexed CTNNB1 (which would have a brightness exceeding one if multiple CTNNB1 molecules reside in a single complex). Therefore, the measured brightness of SGFP2-CTNNB1 in our N and B analysis is an average of both fractions.

First, we confirmed that the number of particles we determined using N and B, were highly similar to those obtained with FCS (compare *Figure 5—figure supplement 1A* with *Table 1*). Second, we observe that the total pool of SGFP2-CTNNB1 in both the cytoplasm and nucleus has a brightness similar to EGFP and SGFP2 monomers (*Figure 5H*, *Table 3*). Because we found a substantial fraction (34–38%) of SGFP2-CTNNB1 to reside in a large complex using point FCS (*Figure 5C–D*), this suggests that few, if any, of these complexes contain multiple SGFP2-CTNNB1 molecules. If the cytoplasmic SGFP2-CTNNB1 containing complex indeed represents a large, multivalent destruction complex, this would imply that under physiological conditions, quite unexpectedly, most CTNNB1-binding sites are unoccupied in both the absence and presence of WNT3A.

## A minimal computational model of WNT/CTNNB1 signaling

Quantitative measurements and physical parameters of WNT pathway components and their interactions remain limited (*Kitazawa et al., 2017*; *Lee et al., 2003*; *Tan et al., 2012*), especially in living cells. As we obtained absolute measurements of different functional pools of CTNNB1, we next sought to integrate these biophysical parameters in a minimal computational model of WNT signaling to identify the critical nodes of regulation of subcellular CTNNB1 distribution (*Figure 6A*, *Tables 4–5*, Materials and methods). This minimal model is based on a previous model of Kirschner and colleagues (*Lee et al., 2003*), and incorporates the new data obtained in our study, supplemented with parameters from the literature (*Lee et al., 2003*; *Tan et al., 2012*).

Our model diverges from the model presented by Lee et al. on two major points. First, the model is simplified by replacing the details of the destruction complex formation cycle and the individual actions of APC and AXIN with a single, fully formed destruction complex. We chose this option because our study does not provide new quantitative data on the formation and dynamics of the destruction complex, but does provide absolute concentrations of CTNNB1 in a bound state in the cytoplasm. Second, we explicitly include shuttling of CTNNB1 between the cytoplasm and nucleus in both directions (*Schmitz et al., 2013*; *Tan et al., 2014*).

Thus, our model (*Figure 6A*) describes the binding of cytoplasmic CTNNB1 ('CB') to the destruction complex ('DC') leading to its phosphorylation and degradation (described by k3), which releases the DC. Transport of CTNNB1 from the cytoplasm to the nucleus, allows nuclear CTNNB1 ('NB') to bind to TCF/LEF forming a transcriptional complex ('NB-TCF'). When WNT is present in the system, we describe the inactivation of the destruction complex ('DC*') by DVL through the parameter *w* (see Materials and methods section model description). The model is available as interactive app at https://wntlab.shinyapps.io/WNT_minimal_model/ and allows users to explore the effects of modulating different equilibria and constants in an intuitive way.

Our model faithfully recapitulates the dynamic changes observed with functional imaging (compare *Figure 6B–F* to *Figures 3* and *5*). Moreover, it reveals two critical regulatory nodes in addition to the requisite inactivation of the destruction complex (described by k5/k4). The first additional node of regulation is nuclear import and export (or 'shuttling', described by k6/k7). Upon WNT stimulation, the ratio of k6/k7 (nuclear shuttling) needs to increase in order for the model to match the free CTNNB1 concentrations we measured by FCS (*Table 5*, *Figure 5B*). Thus, the balance shifts from nuclear export before WNT, to nuclear import after WNT. The second additional node of regulation is the association of CTNNB1 with the TCF transcriptional complex (or 'retention'), described by k9/k8. Upon WNT stimulation, the ratio of k9/k8 (nuclear retention) needs to decrease by almost a factor of 10 in order for the model to reproduce the concentrations of free and bound CTNNB1 in

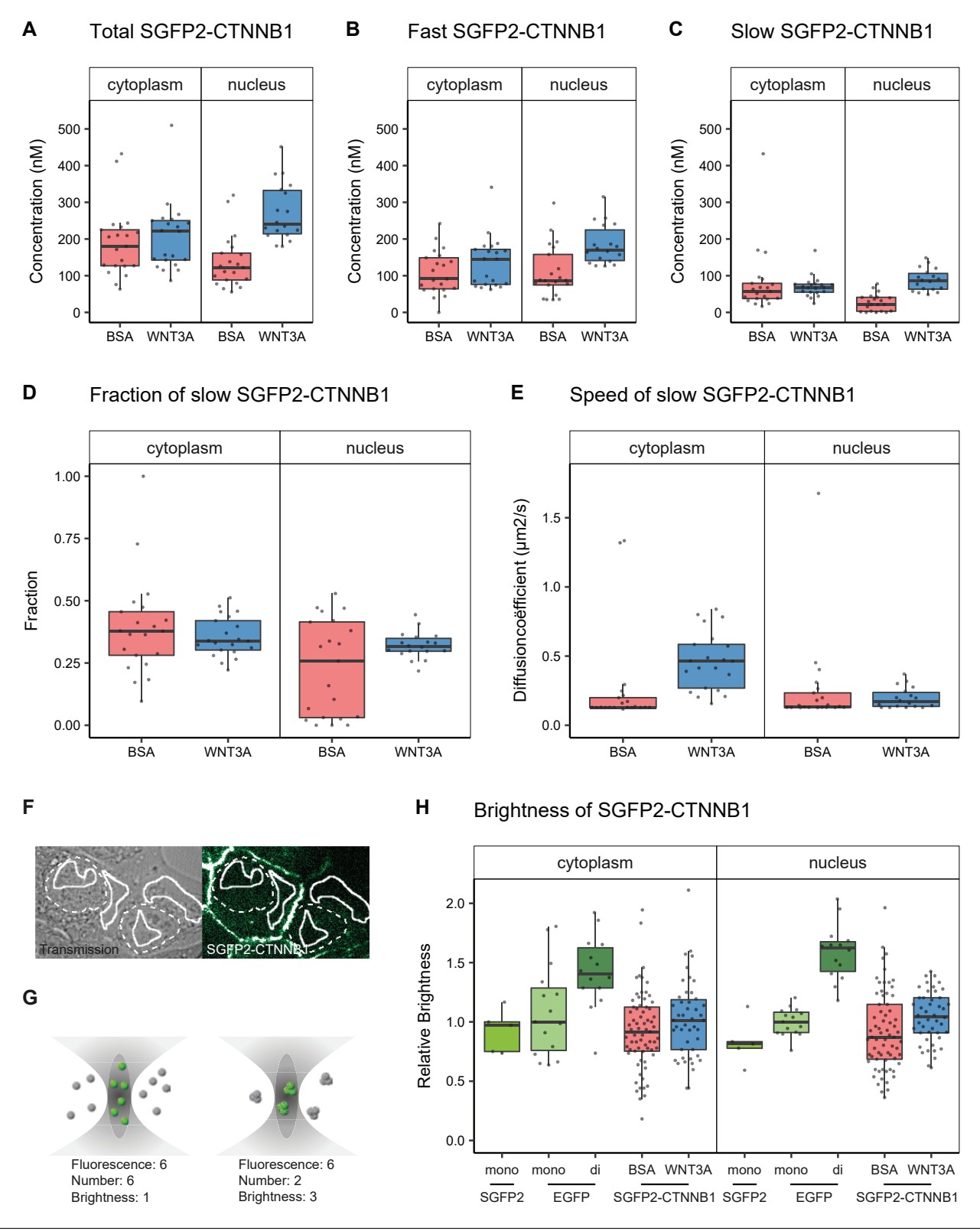

**Figure 5.** Abundance and mobility of SGFP2-CTNNB1 molecules in living cells after 4 hr WNT3A treatment or control. Details on sample size and statistics can be found in *Supplementary file 1*. (**A**) Graph depicting the total concentration of SGFP2-CTNNB1 particles (monomeric plus complexed) as measured with FCS. (**B**) Graph depicting the concentration of SGFP2-CTNNB1 particles with the fast diffusion component (i.e. free monomeric). (**C**) Graph depicting the concentration of SGFP2-CTNNB1 containing particles with the slow diffusion component (i.e. complex associated). (**D–E**) Graphs

*Figure 5 continued on next page*

*Figure 5 continued*

depicting the fraction (D) and speed (E) of the second diffusion component (i.e. SGFP2-CTNNB1 containing complex) measured by FCS. (F) Example of typical regions of interest in two cells used in N and B analysis. Solid line represents the analysis ROI, dashed line, marks the outline of the nuclear envelope. (G) Schematic representation of a confocal volume with different brightness species. On the left are six monomers with a brightness of 1, on the right two trimers with a brightness of 3, both result in a fluorescence of 6. N and B analysis is able to extract the number and the brightness of such samples, for more detail see supplement 1 of this figure. (H) Graph depicting the molecular brightness of SGFP2-CTNNB1 in the cytoplasm and nucleus relative to controls as measured with N and B in the same subcellular compartments. EGFP monomer was used for normalization and EGFP dimer as a control for N and B measurements.

The online version of this article includes the following source data and figure supplement(s) for figure 5:

**Source data 1.** Numerical data for *Figure 5* (all graphs) and *Figure 5—figure supplement 1* (all panels).
**Figure supplement 1.** Quantification of SGFP2-CTNNB1 particles, fluorescence and fluorescence lifetime.
**Figure supplement 2.** Number and Brightness analysis.

the nucleus as measured by FCS (*Table 5*, *Figure 6F*, *Figure 5B–C*). Thus, association of CTNNB1 to the TCF transcriptional complex is favored after WNT stimulation. In summary, our model suggests that WNT/CTNNB1 signaling is regulated at three distinct levels of the signal transduction pathway: destruction complex inactivation, nucleocytoplasmic shuttling and nuclear retention. How WNT signaling influences nuclear shuttling and nuclear retention is an open question and both are areas of active research (*Anthony et al., 2020*; *Söderholm and Cantù, 2020*).

## Perturbing the system to mimic oncogenic WNT signaling

WNT signaling is often disrupted in cancer (reviewed in *Polakis, 2000*; *Zhan et al., 2017*), frequently due to inactivating mutations in negative regulators or due to activating mutations in CTNNB1 itself (*Bugter et al., 2021*). One of the earliest identified mutations in CTNNB1 was a substitution of serine-45 for a phenylalanine (S45F) (*Morin et al., 1997*). This mutation removes the CSNK1A1 priming phosphorylation site on CTNNB1 that is needed for sequential phosphorylation by GSK3, and thus blocks its proteasomal degradation (*Amit et al., 2002*; *Liu et al., 2002*).

We generated the S45F mutation in one of our HAP1[SGFP2-CTNNB1] cell lines through a second step of CRISPR/Cas9-mediated genome editing (*Figure 7—figure supplement 1A–D*). As expected, the mutation resulted in higher CTNNB1 levels (*Figure 7—figure supplement 1E–F*) and constitutive downstream activation of the pathway (*Figure 7—figure supplement 1G–H*). Next, we used this cell line for two purposes. First, we used FCS and N and B to compare the complex-state of wild-type and mutant CTNNB1 in the cytoplasm (*Figure 7*). Second, we reproduced the same perturbation in silico to strengthen the link between our experimental data and the computational model (*Figure 8*).

Similar to the situation detected under physiological conditions (*Figure 5D*, slow fraction: median 0.38), we find a large fraction (median 0.402, 95% CI 0.363–0.471) of SGFP2-CTNNB1[S45F] to reside in a cytoplasmic complex (*Figure 7A*). As observed for physiological stimulation with WNT3A

**Table 1.** Total number of SGFP2-CTNNB1 molecules in the confocal volume and corresponding calculated concentrations obtained from FCS measurements in n=3 independent experiments.

The concentration is calculated from the number of molecules in the confocal volume and the calibrated confocal volume (see Materials and methods). The number of molecules is consistent with those measured with N and B analysis (*Figure 5—figure supplement 1A*, *Supplementary file 1*).

| Compartment | Treatment | N | Number of molecules | | Concentration (nM) | |
| --- | --- | --- | --- | --- | --- | --- |
| | | | Median | 95% CI | Median | 95% CI |
| Cytoplasm | BSA | 21 | 80 | 70–116 | 180 | 127–218 |
| | WNT3A | 21 | 95 | 85–122 | 221 | 144–250 |
| Nucleus | BSA | 21 | 63 | 53–72 | 122 | 91–158 |
| | WNT3A | 18 | 135 | 127–150 | 240 | 217–325 |

**Table 2.** Number of SGFP2-CTNNB1 molecules in the confocal volume and corresponding calculated concentration of SGFP2-CTNNB1 molecules with the fast or slow diffusion coefficient obtained from FCS measurements in n=3 independent experiments. The concentration is calculated from the number of molecules and the calibrated confocal volume (see Materials and methods).

| | | | Fast SGFP2-CTNNB1 | | | | Slow SGFP2-CTNNB1 | | | |
| | | | Number of molecules | | Concentration (nM) | | Number of molecules | | Concentration (nM) | |
| Compartment | Treatment | n | Median | 95% CI | Median | 95% CI | Median | 95% CI | Median | 95% CI |
|---|---|---|---|---|---|---|---|---|---|---|
| Cytoplasm | BSA | 21 | 51 | 40–63 | 91 | 66–139 | 29 | 20–37 | 57 | 38–76 |
| | WNT3A | 21 | 60 | 47–80 | 145 | 76–168 | 35 | 30–41 | 68 | 57–76 |
| Nucleus | BSA | 21 | 48 | 41–66 | 87 | 78–119 | 13 | 2–22 | 22 | 4–40 |
| | WNT3A | 18 | 96 | 81–101 | 170 | 147–214 | 47 | 37–49 | 86 | 64–104 |

(*Figure 5E*), the speed of this complex is significantly increased (approximately threefold) in SGFP2-CTNNB1$^{S45F}$ (median 0.589 $\mu m^2 s^{-1}$, 95% CI 0.585–0.691) compared to unstimulated HAP1$^{SGFP2-CTNNB1}$ cells (median 0.191 $\mu m^2 s^{-1}$, 95% CI 0.115–0.29) (*Figure 7B*). We find similar behavior when we block the GSK3-mediated phosphorylation of wild-type CTNNB1 using CHIR99021 (*Figure 7C–D*). The reduction in cytoplasmic complex size therefore must occur downstream of CTNNB1 phosphorylation. Intriguingly, our N and B analyses suggest that these smaller S45F mutant cytoplasmic complexes have a higher occupancy of CTNNB1 (*Figure 7E*) than the those in WNT3A (*Figure 5H*) or CHIR99021 (*Figure 7F*) stimulated wildtype cells. The S45F mutant (median 1.304, 95% CI 1.139–1.418, p=0.002) was significantly brighter than the SGFP2 monomer control (median 0.866, 95% CI 0.573–0.949), where the WT-tagged HAP1 cells again did not diverge from the monomer (0.886, 95% CI 0.722–1, p=0.845) (file 1). Thus, under oncogenic conditions more binding sites in the cytoplasmic CTNNB1 complex may be occupied than in physiological circumstances.

The S45F mutant shows a substantial increase in SGFP2-CTNNB1 levels in the cytoplasm and nucleus (*Figure 8A*). As this constitutive mutation does not provide any kinetic information, we also measured the dynamic response of SGFP2-CTNNB1 to CHIR99021-mediated GSK3 inhibition (*Figure 8—figure supplement 1*, *Video 3*). We see similar initial kinetics as for WNT3A stimulation. However, in contrast to what is observed for WNT3A treatment, no plateau was reached at the highest concentration of CHIR99021 (8 µM). Of note, the quantification also confirms that there is cell to

**Table 3.** Brightness of SGFP2-compared relative to EGFP-monomer and -dimer controls in n=2 independent experiments. N is the number of analyzed cells.
p-Values were calculated using PlotsOfDifferences that uses a randomization test (*Goedhart, 2019*).Note that only the EGFP-dimer is significantly different to the EGFP-monomer control, while SGFP2-CTNNB1 is not.

| Fluorophore | Compartment | Treatment | N | Median | 95 CI median | p-value to matched control (EGFP monomer in the nucleus or cytoplasm) |
|---|---|---|---|---|---|---|
| EGFP-monomer | Cytoplasm | NA | 15 | 1 | 0.79–1.34 | 1.000 |
| EGFP-dimer | Cytoplasm | NA | 14 | 1.4 | 1.29–1.60 | 0.011* |
| SGFP2-CTNNB1 | Cytoplasm | BSA | 69 | 0.92 | 0.83–1.00 | 0.738 |
| SGFP2-CTNNB1 | Cytoplasm | 100 ng/ml WNT3A | 46 | 1.01 | 0.93–1.11 | 0.919 |
| EGFP-monomer | Nucleus | NA | 15 | 1 | 0.91–1.07 | 1.000 |
| EGFP-dimer | Nucleus | NA | 14 | 1.62 | 1.44–1.69 | <0.001* |
| SGFP2-CTNNB1 | Nucleus | BSA | 69 | 0.87 | 0.78–0.96 | 0.192 |
| SGFP2-CTNNB1 | Nucleus | 100 ng/ml WNT3A | 46 | 1.05 | 0.95–1.15 | 0.578 |

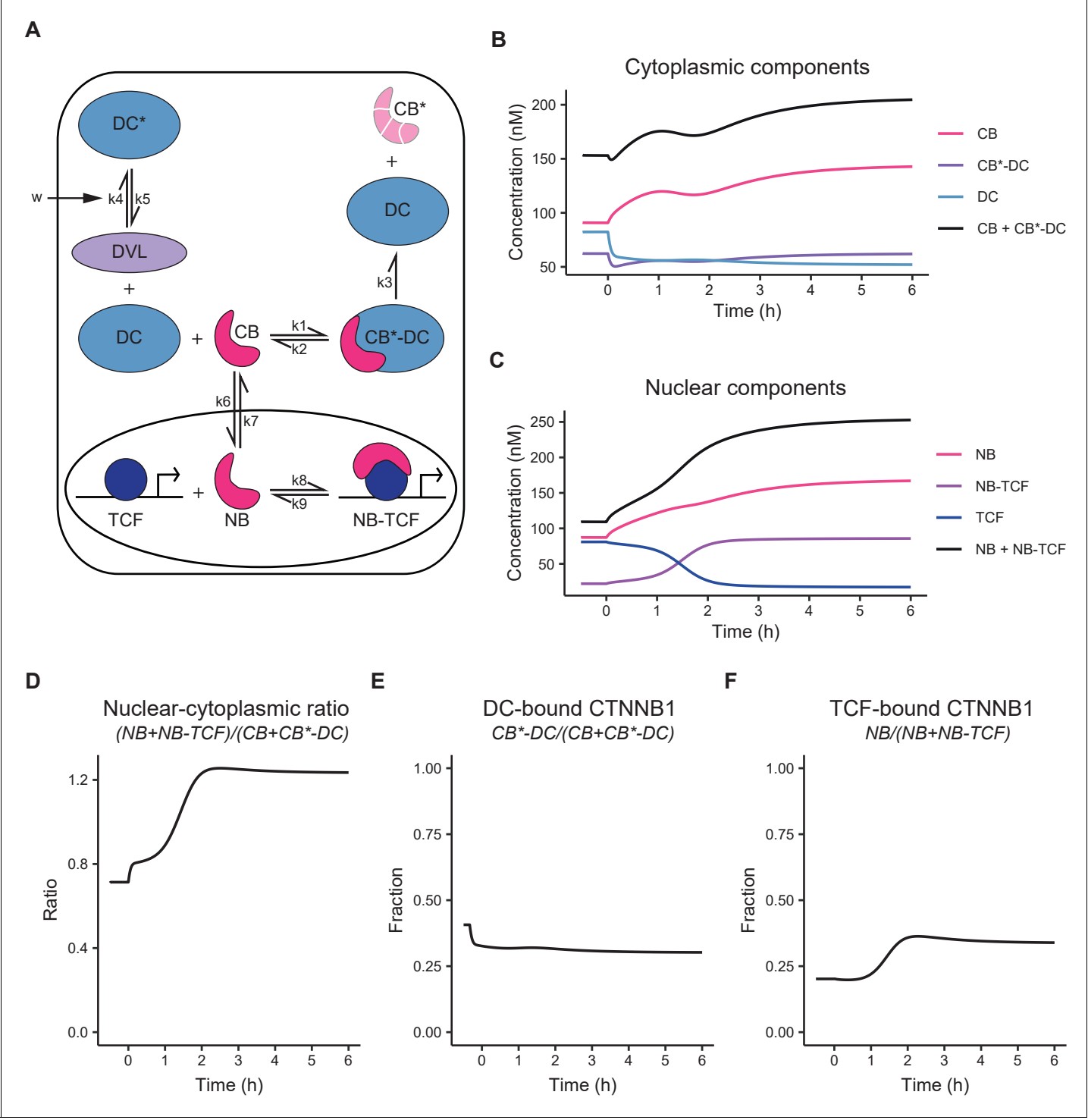

**Figure 6.** Computational model of WNT/CTNNB1 based on FCS concentrations for free and complexed CTNNB1 (*Tables 1–2*). (**A**) Schematic overview of the model. DC=destruction complex, DC* = DVL-inactivated DC, CB=cytoplasmic CTNNB1, CB*=phosphorylated CB, NB=nuclear CTNNB1, TCF=TCF/LEF transcription factors, DVL=WNT-activated DVL. In WNT OFF, w=0, and therefore k5/k4 does not play a role and no inactivated destruction complex is formed. In WNT ON, w=1, and k5/k4 is put into action, resulting in an increase in DC* at the expense of DC. Under the assumption that k3 remains equal and given that CB*-DC was experimentally determined to be the same in WNT ON and WNT OFF, removal of DC, results in an increase in CB. Changes in k6/k7 and k9/8 further increase NB and NB-TCF in WNT ON. Note that CB* is degraded and therefore plays no role in the model. (**B**) Graph depicting the modeled concentrations of cytoplasmic components over time. The black line indicates total concentration of cytoplasmic CTNNB1, corresponding to *Figure 3B*. (**C**) Graph depicting the modeled concentrations of nuclear components over time. The black

*Figure 6 continued on next page*

*Figure 6 continued*

line indicates total concentration of nuclear CTNNB1, corresponding to *Figure 3C*. (D) Graph depicting the ratio of total nuclear and cytoplasmic CTNNB1 over time, corresponding to the measurements in *Figure 3D*. (E) Graph depicting the DC-bound CTNNB1 fraction ratio over time. (F) Graph depicting the TCF-bound CTNNB1 fraction ratio over time.

cell heterogeneity in the response, regardless of whether WNT/CTNNB1 signaling is activated at the level of the receptor (WNT3A treatment) or at the level of the destruction complex (CHIR99021 treatment), as can be seen from the spread of intensities measured from individual cells (*Figure 3— figure supplement 1A–B*).

Finally, we compared our biological measurements from these perturbation experiments to our computational model predictions. Both the S45F mutation and CHIR99021 treatment disrupt degradation of phosphorylated CTNNB1 (corresponding to k3, *Figure 6A*). With FCS and N and B we quantified the accumulation of CTNNB1 levels of mutant SGFP2-CTNNB1$^{S45F}$ (*Figure 8B*, *Figure 8— figure supplement 2A*) and wild-type SGFP2-CTNNB1 upon CHIR99021 treatment (*Figure 8—figure supplement 2B–C*). Both exceeded the levels observed with physiological WNT3A stimulation (*Figure 3A–C*, *Figure 5A*). Specifically, the absolute concentration of SGFP2-CTNNB1$^{S45F}$ in the cytoplasm (median 351 nM, 95% CI 276–412) exceeded that of SGFP2-CTNNB1 in WNT3A-treated cells (median 221 nM, 95% CI 144–250). In the nucleus, the concentration of SGFP2-CTNNB1$^{S45F}$ reached 429 nM (median, 95% CI 387–481), as opposed to 240 nM (median, 95% CI 217–325) for SGFP2-CTNNB1 in WNT3A-treated cells as a result of losing its priming phosphorylation site. While this further increase in concentration is evident, it should be noted that in both the cytoplasm and the nucleus CTNNB1 levels thus rise less than twofold in an oncogenic setting compared to WNT3A treatment.

In our computational model, we simulated reduced degradation by lowering the value of k3. A reduction in k3 from its initial value (k3=0.0068, *Table 5*) to k3=0.0043, accurately predicted the higher cytoplasmic concentration measured for the S45F mutant (*Figure 8C*), but a further reduction to k3=0.0038 was needed to match the measured nuclear concentration (*Figure 8D*). However, reducing k3 alone was not sufficient to reproduce either the fraction of CTNNB1 that is bound in the nucleus (*Figure 8E–G*) or the overall nuclear enrichment of CTNNB1 (*Figure 8H*). The latter requires a predicted nuclear/cytoplasmic (N/C) ratio greater than one, as observed in both physiological and constitutively active WNT/CTNNB1 signaling (*Figure 8I–K*).

The fraction of bound SGFP2-CTNNB1 in the nucleus was comparable between our HAP1$^{SGFP2-CTNNB1(S45F)}$ mutant cell line (median 0.29, 95% CI 0.27–0.33) (*Figure 8E*), 8 µM CHIR99021 (median 0.38, 95% CI 0.29–0.46) (*Figure 8F*) and WNT3A (median 0.32, 95% CI 0.30–0.34) (*Figure 5D*) treated wild-type HAP1$^{SGFP2-CTNNB1}$ cells. This experimental observation can be matched by adjusting the k9/k8 (nuclear retention) ratio, as was also required for

**Table 4.** Variables Minimal Model of WNT signaling.

| Model name | Variable | Compound | Values obtained from | WNT OFF (nM) | WNT ON (nM) |
|---|---|---|---|---|---|
| CB | $x_1$ | Free cytoplasmic CTNNB1 | FCS data this report | 91 | 145 |
| DC | $x_2$ | Free destruction complex | Model equations | 82.4 | 52 |
| CB*-DC | $x_3$ | DC-bound phosphorylated CTNNB1 | FCS data this report* | 62.5* | 62.5* |
| DC* | $x_4$ | Inactivated destruction complex | Model equations | 0** | 30.5** |
| NB | $x_5$ | Free nuclear CTNNB1 | FCS data this report | 87 | 170 |
| TCF | $x_6$ | Free TCF | Model equations | 81 | 17 |
| NB-TCF | $x_7$ | TCF-bound nuclear CTNNB1 | FCS data this report | 22.2 | 86 |
| TCF$^0$ | TCF$^0$ | Total TCF | $x_7$ and *Tan et al., 2012* - Figure 11 | 103 | 103 |

*Under the assumption that k3 does not change, the levels of CB*-DC remain equal. Since there was no significant difference between the concentration of slow SGFP2-CTNNB1 in the absence or presence of WNT3A (57 nM versus 68 nM, not significant, *Table 2*) the average of both medians (62.5 nM) was used. ** In WNT OFF, w=0, and no inactivated destruction complex is formed. In WNT ON, w=1, which induces the formation of inactivated destruction complex at the expense of free destruction complex (see *Equations 7a and 9* in the model description in Materials and methods).

**Table 5.** Equilibrium conditions for the Minimal Model of WNT signaling.

All rates are multiplied with factor R=20, so that the equilibrium is reached at 4.5 hr according to *Figure 4C–D*.

| Rate constant | | Biological process | Values based on | Wnt off | Wnt on |
|---|---|---|---|---|---|
| b | nMmin$^{-1}$ | CTNNB1 synthesis | $v_{12}$ from Lee | 0.423 | 0.423 |
| $\frac{k_2}{k_1}$ | nM | Binding to and phosphorylation by the destruction complex of cytoplasmic CTNNB1 | K8 from Lee | 120 | 120 |
| $k_3$ | min$^{-1}$ | Dissociation and degradation of phosphorylated CTNNB1 from the destruction complex | Deduced from $b$ and $x_3$ | 0.0068 | 0.0068 |
| $\frac{k_5}{k_4}$ | nM | Inactivation of the destruction complex by activated DVL | Fitted to $x_1$ and $x_7$ | N.A.* | 1.7 |
| $\frac{k_6}{k_7}$ | | Ratio between nuclear import and export of CTNNB1 | Deduced from $x_1$ and $x_5$ | 0.96 | 1.17 |
| $\frac{k_9}{k_8}$ | nM | Dissociation of nuclear CTNNB1 from TCF | Deduced from $x_5$, TCF$^0$, $x_7$ | 320 | 33.6 |

*In WNT OFF, w=0, and no inactive destruction complex is formed. Only in WNT ON, w=1, which induces the formation of inactivated destruction complex at the expense of free destruction complex (see *Equations 7a and 9* in the model description in Materials and methods).

physiological WNT3A signaling (*Table 5*, *Figure 8G*). This shows the importance of this regulatory node not only in physiological, but also in oncogenic signaling. At the same time, the adjustment of nuclear retention (k9/k8) on top of a reduction in degradation (k3) still does not predict the observed nuclear enrichment of CTNNB1 (*Figure 8H*). After changing the nuclear shuttling ratio (k6/k7) to the ratio we fitted for the WNT ON situation (*Table 5*), the model now also reproduces the nuclear enrichment of CTNNB1 (*Figure 8L*). In *Figure 8M–N*, we show that these additional changes in nuclear shuttling (k6/k7) and nuclear retention (k9/k8) have little effect on the CTNNB1 concentration in the cytoplasm, but do substantially affect the nuclear concentrations of CTNNB1. This suggest that processes downstream of CTNNB1 degradation play a significant and active role in the CTNNB1 dynamics of the cell.

Taken together, our computational model can describe both physiological and oncogenic signaling. Moreover, it underlines the importance of CTNNB1 regulation downstream of destruction complex activity and confirms a critical role for nuclear import and nuclear retention.

## Discussion

WNT signaling is critical for tissue development and homeostasis. Although most core players and many of their molecular interactions have been uncovered, dynamic spatiotemporal information with sufficient subcellular resolution remains limited. As both genome editing approaches and quantitative live-cell microscopy have advanced further, the goal of studying WNT/CTNNB1 signaling at endogenous expression levels in living cells now is within reach. Maintaining endogenous expression levels is important, as overexpression may lead to altered stoichiometry of signaling components, as well as changes in subcellular localization (*Gibson et al., 2013*; *Mahen et al., 2014*). Indeed, it has been shown that exogenously expressed CTNNB1 is less signaling competent, probably due to its post-translational modification status (*Hendriksen et al., 2008*).

Here, we generated functional HAP1$^{SGFP2-CTNNB1}$ knock-in cell lines to study the dynamic behavior and subcellular complex state of endogenous CTNNB1 in both a physiological and oncogenic context. Importantly, this allowed us to measure hitherto unknown biophysical parameters of WNT/CTNNB1 in individual living human cells for the first time. Using live-cell microscopy and automated cell segmentation, we observe that endogenous CTNNB1 levels increase only 1.7-fold in the cytoplasm and 3.0-fold in the nucleus after WNT3A treatment, which is consistent with the literature (*Jacobsen et al., 2016*; *Kafri et al., 2016*; *Massey et al., 2019*). Next, we used state-of-the-art, quantitative microscopy to measure the absolute concentration of CTNNB1 within different subcellular compartments and in different complex states in living cells. The findings from these experiments definitively challenge the still prevailing view that mainly monomeric CTNNB1 accumulates upon WNT pathway stimulation (*Nusse and Clevers, 2017*). Moreover, our integrative approach of quantitative imaging and computational modeling revealed three critical nodes of CTNNB1 regulation,

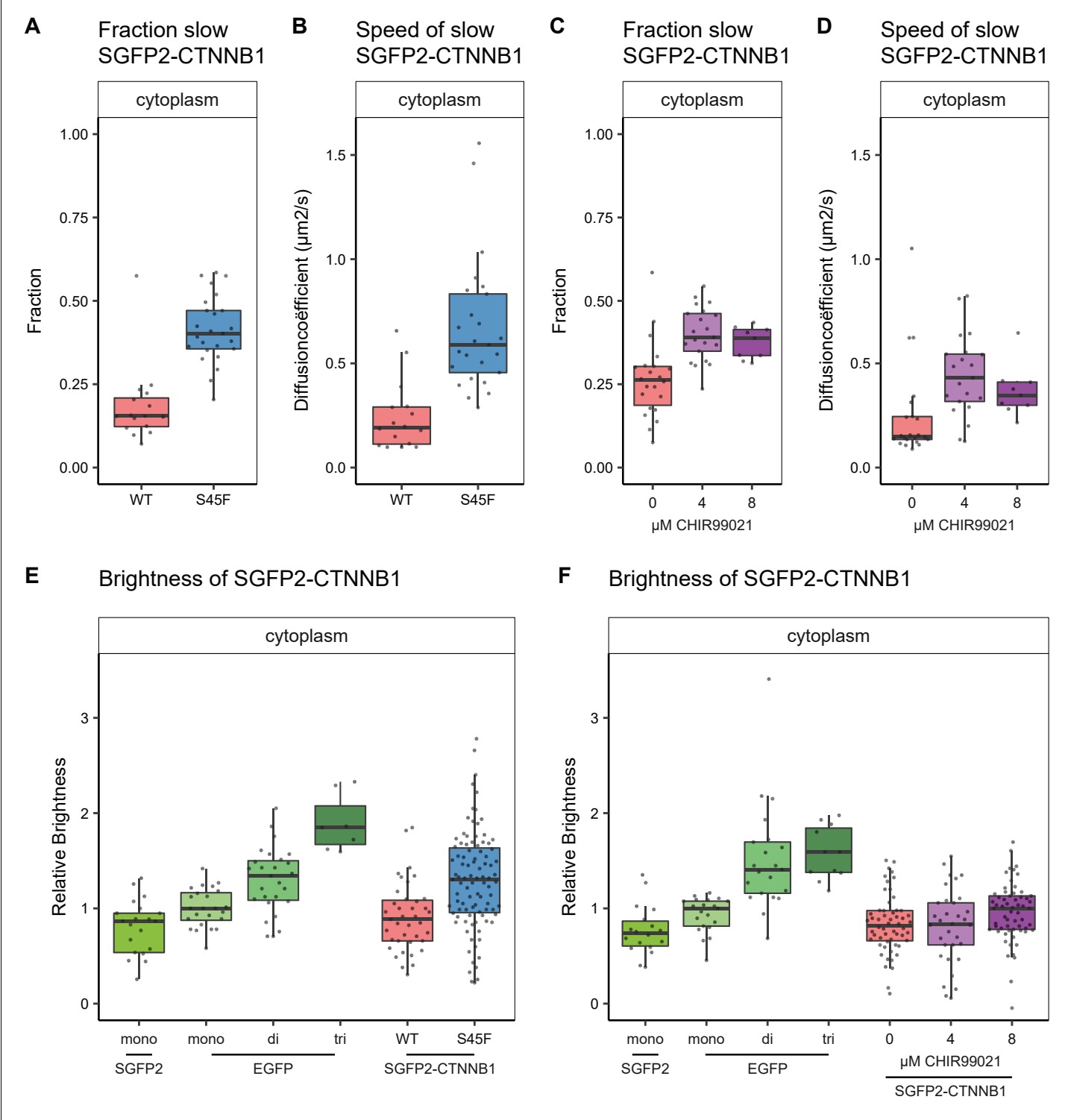

**Figure 7.** Cytoplasmic complex characteristics in absence of SGFP2-CTNNB1 N-terminal phosphorylation. The S45F mutant was introduced using CRISPR (see *Figure 7—figure supplement 1*) and CHIR treated and control cells were measured after 24 hr. Details on sample size and statistics can be found in *Supplementary file 1*. (A) Graph depicting the fraction of particles with the second diffusion component (i.e. SGFP2-CTNNB1 containing complex) measured by FCS for S45F mutant (B) Graph depicting the speed of the second diffusion component (i.e. SGFP2-CTNNB1 containing complex) measured by FCS for S45F mutant. (C).Graph depicting the fraction of particles with the second diffusion component (i.e. SGFP2-CTNNB1 containing complex) measured by FCS after 24 hr treatment with CHIR99021 (C) Graphs depicting the speed of the second diffusion component (i.e. SGFP2-CTNNB1 containing complex) measured by FCS after 24 hr treatment with CHIR99021. (E–F) Graphs depicting the molecular brightness of

*Figure 7 continued on next page*

*Figure 7 continued*

SGFP2-CTNNB1 in the cytoplasm relative to controls as measured with N and B in the same subcellular compartments for S45F mutant CTNNB1 (**E**) or after 24 hr of CHIR99021 treatment (**F**). EGFP monomer was used for normalization and EGFP dimer and trimer as controls for N and B measurements. The online version of this article includes the following figure supplement(s) for figure 7:

**Figure supplement 1.** Generation and characterization of a S45F mutant cell line (HAP1^SGFP2-CTNNB1(S45F)).

namely CTNNB1 degradation, nuclear shuttling and nuclear retention, which together describe the CTNNB1 turnover, subcellular localization and complex status under both physiological and oncogenic conditions.

## Cytoplasmic regulation of CTNNB1

Using FCS, we determined that in unstimulated HAP1 cells a substantial fraction (~30–40%) of SGFP2-CTNNB1 is associated with a very large, slow-diffusing cytoplasmic complex (*Figures 4– 5* and *7*). The main known cytoplasmic complex containing CTNNB1 is the destruction complex. The combined weight of the individual destruction complex components (AXIN, APC, CSNK1A1, and GSK3) would be expected to result in a much higher mobility than that displayed by the cytoplasmic CTNNB1-containing complex we observed. However, evidence is growing that the destruction complex forms large phase separated assemblies (also termed biomolecular condensates) (reviewed in *Schaefer and Peifer, 2019*). Oligomerization of AXIN and APC underlies the formation of these assemblies, and this in turn appears to be required for efficient degradation of CTNNB1 (*Fiedler et al., 2011*; *Kunttas-Tatli et al., 2014*; *Pronobis et al., 2017*; *Spink et al., 2000*). There is some evidence that these biomolecular condensates form at (near) endogenous levels (*Fagotto et al., 1999*; *Faux et al., 2008*; *Mendoza-Topaz et al., 2011*; *Pronobis et al., 2015*; *Schaefer et al., 2018*; *Thorvaldsen et al., 2015*), but it is still an open question what the exact composition and size of the destruction complex is in a physiological context. It should be noted that our imaging does not visualize punctae, which are typically associated with these biomolecular condensates (*Figure 3A*). In addition, our N and B data indicate that most of the slow diffusing CTNNB1 complexes contain one or very few SGFP2-CTNNB1 molecules in either the absence or presence of WNT3A stimulation. In view of the above-mentioned destruction complex oligomerization and its presumed multivalency, this finding was quite unexpected. Several mechanisms could explain this apparent discrepancy. On the one hand, destruction complex multimerization at endogenous levels might be more subtle than previously thought. For example, quantification of AXIN polymerization in vitro showed that even at exceedingly high concentration (24 µM), AXIN polymers typically contained only eight molecules (*Kan et al., 2020*). On the other hand, even if the multivalent destruction complex offers multiple CTNNB1 binding sites, occupancy at any one time might be low, due to the continuous and high turnover of CTNNB1. In this respect, the CTNNB1 bindings sites in the destruction complex could be envisioned to act similar to the wooden vanes in the paddle wheel of an old-fashioned watermill: like the water in the analogous example, CTNNB1 would be continuously scooped up (for phosphorylation) and dropped off (for degradation).

Only following the introduction of an S45F mutation, which results in constitutive inhibition of CTNNB1 phosphorylation and degradation, we observe a brightness increase that would be compatible with the accumulation of multiple SGFP2-CTNNB1^S45F molecules in a single cytoplasmic complex. This indicates that while the destruction complex might be multivalent in both a physiological and an oncogenic context, CTNNB1 occupancy of the complex is low under physiological conditions, but increased in oncogenic signaling. This has major impacts on how we conceptualize the workings of the CTNNB1 destruction machinery – especially in the context of cancer, since mutations in CTNNB1 (affecting occupancy) may have very different biochemical consequences than mutations in APC (affecting multimerization and valency of the destruction complex itself).

The mechanism on destruction complex deactivation remains controversial (*Tortelote et al., 2017*; *Verkaar et al., 2012*). The current literature suggests that the destruction complex is sequestered to the FZD-LRP receptor complex upon WNT pathway stimulation. Several models exist for how membrane sequestration inhibits CTNNB1 degradation, including LRP-mediated GSK3 inhibition (*Stamos et al., 2014*), sequestration of GSK3 in multi vesicular bodies (*Taelman et al., 2010*),

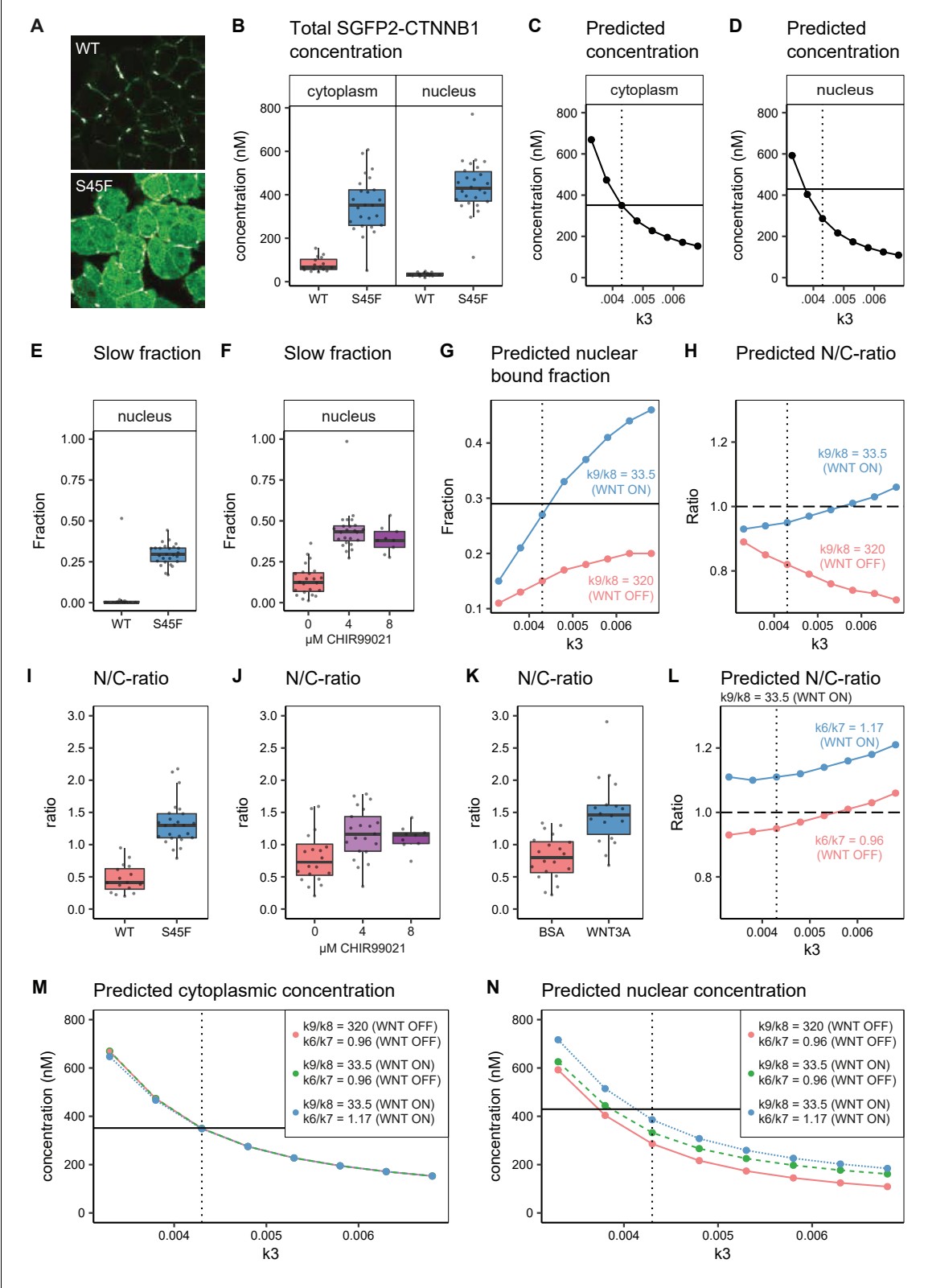

**Figure 8.** In silico and experimental perturbation of WNT signaling. Details on experimental sample size and statistics can be found in **Supplementary file 1**. (**A**) Representative confocal images of HAP1-SGFP2WT (WT, top) and HAP1-SGFP2S45F (S45F, bottom) cells acquired with the same image settings. The S45F mutation leads to the accumulation and nuclear enrichment of CTNNB1 in the cell. (**B**) Graph depicting the total concentration of SGFP2-CTNNB1 particles (monomeric plus complexed) as measured with FCS. (**C–D**) Inhibition of CTNNB1 degradation is modelled

*Figure 8 continued on next page*

*Figure 8 continued*

as a reduction in the value of k3. (C) Graph depicting the predicted total cytoplasmic CTNNB1 concentration as a function of k3. A reduction in k3 from 0.0068 (*Table 5*, WNT ON and WNT OFF conditions) to ~0.0043 (dotted line) corresponds to the cytoplasmic concentration observed (solid line). (D) Graph depicting the predicted total nuclear CTNNB1 concentration as a function of k3. The solid horizontal line indicates the concentration measured for the S45F mutant by FCS. Note that the value of k3 that matches the observed cytoplasm concentration (dotted line) does not match the experimentally determined concentration in the nucleus (solid line). (E–F) Graphs depicting the fraction of particles with the second diffusion component (i.e. SGFP2-CTNNB1 containing complex) measured by FCS for wild-type and S45F mutant (E) and after 24 hr CHIR99021 treatment (F). The increase in the bound fraction in the oncogenic mutant or after GSK3 inhibition we find, is comparable to what we observed in WNT3A stimulated cells (*Figure 5D*). (G) Graph showing the predicted nuclear bound fraction of CTNNB1as a function of k3 with the TCF/CTNNB1 binding affinity of the model (*Table 4*) for WNT OFF (k9/k8 = 320, pink line) and for WNT ON (k9/k8=33.5, blue line). Note that for WNT ON, the value for the nuclear bound fraction approximates the experimentally determined slow fraction for the S45F mutant (solid line, panel E) at the value for k3 that matches the cytoplasmic concentration of CTNNB1 (dotted line). (H) Graph showing the predicted nuclear/cytoplasmic (N/C)-ratio as a function of k3 with TCF/CTNNB1 binding affinity of the model (*Table 4*) for WNT OFF (k9/k8=320, pink line) and WNT ON (k9/k8=33.5, blue line). Note that, although for WNT ON the value of the N/C-ratio increases with k3, there is still nuclear exclusion (N/C-ratio lower than 1, dashed line) at the value of k3 that matches the cytoplasmic CTNNB1-concentration (dotted line). (I–K) The N/C-ratio as measured by FCS for wild-type and S45F mutant (I), after 24 hr CHIR99021 treatment (J) and after 4 hr WNT3A treatment (K). Note that all perturbations lead to nuclear accumulation (N/C-value exceeding 1). (L) Graph showing the predicted N/C-ratio as a function of k3 with the WNT ON value for k9/k8 with the nuclear shuttling ratio of the model (k6/k7 *Table 4*),corresponding to WNT OFF (k6/k7=0.96, pink line) and WNT ON (k6/k7=1.17, blue line), respectively. Note that the WNT ON value of k6/k7 increases the N/C-ratio to nuclear accumulation at the value for k3 that matches the cytoplasmic concentration (dotted line). (M) Graph depicting the predicted total cytoplasmic CTNNB1 concentration as a function of k3 with WNT ON and WNT OFF values for k9/k8 and k6/k7. Note that modulation of k9/k8 and k6/k7 has virtually no effect on the predicted cytoplasmic concentration of CTNNB1, resulting in overlapping points and lines in the graph. The horizontal solid line is the experimentally determined cytoplasmic CTBNN1 concentration (cf. panel B); the vertical dotted line is at the value of k3 that best reproduces this experimental finding in the model. (N) Graph depicting the predicted total nuclear CTNNB1 concentration as a function of k3 for WNT ON and WNT OFF values for k9/k8 and k6/k7. Note that if both k9/k8 and k6/7 are changed from their WNT OFF values the predicted nuclear concentration of CTBNN1 better matches the experimentally determined concentration (horizontal solid line) at the value for k3 that matches the cytoplasm concentration (vertical dotted line).

The online version of this article includes the following figure supplement(s) for figure 8:

**Figure supplement 1.** Live imaging of HAP1[SGFP2-CTNNB1] upon CHIR99021 stimulation.

**Figure supplement 2.** additional biophysical properties of SGFP2-CTNNB1[S45F] and SGFP2-CTNNB1 under CHIR99021 stimulation.

(partial) dissociation of the destruction complex (*Liu et al., 2005*; *Tran and Polakis, 2012*), and saturation of CTNNB1 within an intact destruction complex (*Li et al., 2012*). Our data clearly show that a substantial fraction of CTNNB1 in the cytoplasm remains bound upon pathway stimulation (*Figure 5D*). This is not captured by textbook models and cartoons that typically still depict the view that mainly monomeric CTNNB1 accumulates. It is, however, in line with the notion that CTNNB1 still accumulates in a destruction complex after WNT signaling. One line of evidence proposes that WNT traps CTNNB1 in a complex by inhibiting its transfer to the E3 ligase, rather than by inhibiting its phosphorylation and release (*Li et al., 2012*; *Pronobis et al., 2015*; *Schaefer et al., 2020*). However, other reports strongly suggest that CTNNB1 phosphorylation is at least partially inhibited during the WNT response (*Hernández et al., 2012*; *Mukherjee et al., 2018*).

The substantial fraction of slow diffusing CTNNB1 that remains upon physiological and oncogenic stimulation of the pathway, is also consistent with the previously proposed role for cytoplasmic retention by the destruction complex (*Krieghoff et al., 2006*; *Roberts et al., 2011*; *Yamulla et al., 2014*). These studies also show that both nuclear and cytoplasmic retention have an important role in determining the subcellular distribution of CTNNB1. Moreover, cytoplasmic retention of CTNNB1 can contribute to reducing downstream pathway activation even in the presence of oncogenic APC mutations (*Kohler et al., 2008*; *Li et al., 2012*; *Schneikert et al., 2007*; *Yang et al., 2006*), which further highlights the importance of this process.

We show that the cytoplasmic CTNNB1 complex in WNT3A or CHIR99021 treated cells as well as in S45F mutant cells has an over threefold increased mobility compared to control cells. Therefore, while the diffusion coefficient is still very low (indicating that the remaining complex is still very large), this implies it is a vastly different complex than that observed in the absence of WNT stimulation. The precise nature of these complexes remains unknown, but could be consistent with a reduced destruction complex size after WNT treatment, as also recently observed in *Drosophila* for AXIN complexes (*Schaefer et al., 2018*), or with the formation of inactivated

destruction complexes ('transducer complexes') in response to WNT/CTNNB1 pathway activation (*Hagemann et al., 2014*; *Lybrand et al., 2019*). The fact that cells in which GSK3 phosphorylation is inhibited through S45F mutation or CHIR99021 treatment show similar behavior, suggests that the size of the cytoplasmic complex is directly linked to the phosphorylation status of CTNNB1. The destruction complex has been shown to associate with (parts of) the ubiquitin and proteasome machinery (*Li et al., 2012*; *Lui et al., 2011*; *Schaefer et al., 2020*; *Thorvaldsen et al., 2015*). One interesting possibility, therefore, is that phosphorylated CTNNB1 is required for coupling the destruction complex to the ubiquitination and proteasome machinery. In fact, although not explicitly mentioned in the main text, supplementary table 1 of *Li et al., 2012* shows that in HEK293 cells, which harbor no mutation in the core components of the WNT pathway, CTNNB1 was found to interact with subunits of the proteasome, whereas in the S45F-CTNNB1 mutant cell line Ls174T these interactions were not detected. In conclusion, although we do not directly determine its identity, our measured biophysical parameters of the cytoplasmic CTNNB1 complex are consistent with it representing a large, multivalent destruction complex that is coupled to the proteasome as long as CTNNB1 is being phosphorylated.

## Nuclear regulation of CTNNB1

The key function of CTNNB1 downstream of WNT is to regulate transcription of TCF/LEF target genes (*Doumpas et al., 2019*; *Schuijers et al., 2014*). Proteomic analyses have shown that the WNT enhanceosome consists of CTNNB1, TCF/LEF, Pygopus Homologs 1 and 2 (PYGO) and B-cell CLL/lymphoma nine protein (BCL9) and several other large proteins (*Fiedler et al., 2015*; *van Tienen et al., 2017*). Using FCS, we show that CTNNB1 resides in a nuclear complex with a diffusion coefficient that is compatible with such a DNA-bound transcriptional complex (*Figure 5E*; *Lam et al., 2012*).

Although CTNNB1 is known to associate with TCF/LEF factors in response to WNT/CTNNB1 signaling to drive transcription (*Franz et al., 2017*; *Schuijers et al., 2014*), we also detect low levels of nuclear CTNNB1 complex in the absence of a WNT stimulus (*Figure 5C*). The diffusion coefficient of the nuclear CTNNB1 complex does not change upon the addition of WNT3A (*Figure 5E*), suggesting that some CTNNB1 is already associated with the DNA even in the absence of a WNT stimulus. At this point, we cannot exclude the contribution of TCF/LEF independent DNA binding (*Armstrong et al., 2012*; *Essers et al., 2005*; *Kormish et al., 2010*), or anomalous subdiffusion in the nucleus, either due to physical obstruction, transient DNA-binding events protein or protein complex formation (*Dross et al., 2009*; *Kaur et al., 2013*; *Wachsmuth et al., 2000*), as FCS only allows us to probe the speed of this complex.

However, upon pathway activation through WNT3A, CHIR99021 or S45F mutation we see a consistent increase in the fraction and absolute levels of this slow-diffusing nuclear CTNNB1 complex (*Figure 5E*, *Figure 8E–F*), compatible with increased CTNNB1 binding to its target sites. Upon WNT stimulation, the concentration of bound SGFP2-CTNNB1 in the nucleus increased to ~90 nM, which corresponds to something in the order of 20,000 bound CTNNB1 molecules in one nucleus, assuming a nuclear volume of 0.36 picoliter (*Tan et al., 2012*). Published CHIPseq studies report many CTNNB1 DNA binding sites, ranging from several hundred to several thousand sites in mammalian cells (*Cantù et al., 2018*; *Doumpas et al., 2019*; *Schuijers et al., 2014*). It is therefore highly likely that at least part of the slow-diffusing CTNNB1 particles we measure indeed represents CTNNB1 that is associated with the WNT enhanceosome.

## Regulation of CTNNB1 nuclear accumulation

In HAP1 cells, endogenous CTNNB1 is excluded from the nucleus in the absence of WNT. Our live imaging data reveal an immediate and preferential increase in nuclear CTNNB1 upon WNT3A stimulation, until an equilibrium is reached between the cytoplasmic and nuclear levels (*Figure 3D*). This is consistent with previous observations in HEK293 cells stably overexpressing low levels of YFP-CTNNB1 (*Kafri et al., 2016*).

Intriguingly, CTNNB1 does not contain nuclear import or export signals and can translocate independently of classical importin and exporter pathways (*Fagotto et al., 1998*; *Wiechens et al., 2001*; *Yokoya et al., 1999*). Hence, the molecular mechanism of CTNNB1 subcellular distribution remains incompletely understood. Evidence from Fluorescence Recovery After Photobleaching (FRAP)

studies suggest that the increase in nuclear CTNNB1 is due to changes in binding to its interaction partners in the cytoplasm and nucleus (retention) rather than active changes in nuclear import and export rates (shuttling) (*Jamieson et al., 2011*; *Krieghoff et al., 2006*). We argue that the two are not mutually exclusive, as our experimental data and computational model show that WNT regulates both nucleocytoplasmic shuttling and nuclear retention of CTNNB1. Indeed, we see an increase of nuclear CTNNB1 complexes in the nucleus (*Figure 5C–D*) and the dissociation of CTNNB1 from TCF is reduced almost 10-fold in WNT signaling conditions in our computational model (*Table 5*). Our model predicts that this increased nuclear retention indeed also increases the nuclear/cytoplasmic ratio (*Figure 8H*). However, to reconcile our computational prediction with our experimental observations, we additionally need to include a shift from nuclear export to nuclear import upon pathway activation (*Figure 6*, *Figure 8*). Our integrated experimental biology and computational modeling approach thus reveals that WNT signaling not only regulates the absolute levels of CTNNB1 through destruction complex inactivation, but also actively changes its subcellular distribution through nuclear retention and shuttling. The fact that direct inhibition of GSK3-mediated phosphorylation of CTNNB1 results in the same behavior, indicates that the phosphorylation status of CTNNB1 plays a critical role. This further emphasizes the importance of posttranslational modifications and conformational changes in CTNNB1 for its subcellular localization and function (*Gottardi and Gumbiner, 2004*; *Sayat et al., 2008*; *Valenta et al., 2012*; *van der Wal and van Amerongen, 2020*; *Wu et al., 2008*).

## Differences and similarities in physiological and oncogenic WNT signaling

As discussed above, several behaviors of CTNNB1 are conserved between the different modes of stimulation. For instance, WNT3A treatment, GSK3 inhibition and the oncogenic S45F mutation all result in (1) increased overall levels of CTNNB1; (2) a substantial fraction of CTNNB1 in a faster, albeit still very large complex in the cytoplasm; (3) increased nuclear accumulation of CTNNB1; and (4) increased retention of CTNNB1 in nuclear complexes. Our computational modeling further confirms that in addition to regulation of CTNNB1 turnover – either by removal of activated destruction complex or through inhibition of phosphorylation and ubiquitination – nuclear shuttling and nuclear retention are equally important regulatory nodes in oncogenic (CHIR99021 treatment or S45F mutation) and physiological (WNT3A stimulation) signaling.

However, the absolute levels of CTNNB1 and the resulting transcriptional activation are distinct in each of these conditions: Cells treated with a GSK3 inhibitor continue to accumulate CTNNB1 after 4 hr, when WNT3A treated cells reach a plateau. The latter is likely due to the fact that negative feedback mechanisms kick in, such as reconstitution of the destruction complex by AXIN2 or internalization of WNT-bound receptor complexes (*Agajanian et al., 2019*; *Lustig et al., 2002*), both of which function upstream of GSK3. Alternatively, it could reflect the notion that physiological WNT signaling does not turn the destruction complex off completely, but rather 'turns it down', as our N and B data support the fact that under physiological conditions the destruction complex itself provides a surplus reservoir of CTNNB1-binding sites that may only become occupied when WNT signaling is hyperactivated. As a combined result, GSK3 inhibition or S45F mutation of CTNNB1 can result in higher total intracellular levels of CTNNB1. Indeed, concentrations of SGFP2-CTNNB1 in S45F mutated cells exceed those in WNT3A treated cells in both the cytoplasm (1.6-fold) and the nucleus (1.8-fold). This subtle increase in CTNNB1 levels is likely amplified at the transcriptional level (*Jacobsen et al., 2016*), consistent with the well-known fact that constitutive activation of the pathway through different mechanisms, including APC mutation, results in higher pathway activation than physiological stimuli.

## Challenges and opportunities for fluorescence fluctuation spectroscopy techniques

Using fluorescence fluctuation spectroscopy techniques (FCS and N and B), we have quantified endogenous CTNNB1 concentrations and complexes in living cells for the first time, which provided novel and long-awaited biophysical parameters for computational modeling. Moreover, our approach has also yielded novel insights into CTNNB1 regulation that challenge current dogmas in the field. If we are correct, this has important consequences. First, if only part of the cytoplasmic

CTNNB1 pool is present in an uncomplexed (i.e. free or monomeric) state, regardless of whether the WNT/CTNNB1 pathway is off or on (either via physiological WNT3A stimulation or via oncogenic activation), this is a rewrite of the textbook model. Second, if the slow-diffusing cytoplasmic CTNNB1 complex indeed represents a proteasome-associated destruction complex, this would fuel a debate that has remained unresolved for many years (*Li et al., 2012*; *Verkaar et al., 2012*). As more studies will use image-based techniques to determine biophysical properties of WNT/CTNNB1 signaling events (*Ambrosi et al., 2020*; *Eckert et al., 2020*), the field will undoubtedly learn how to interpret these findings.

While it is tempting to speculate about the implications of our findings, as with any technique, there are several limitations to consider. First of all, to deduce absolute concentrations several variables need to be considered, including autofluorescence, bleaching and dark states. Although the FCS data are corrected for autofluorescence and bleaching (see *Equation 3* in FCS data acquisition and analysis section in the Materials and methods), this could potentially introduce some errors. It should also be noted that a small portion of SGFP2-CTNNB1 could be in a non-fluorescent state. Although our FCS analysis model already accounts for dynamic dark states such as the triplet state, non-matured fluorophores could lead to a slight underestimation of our concentrations. However, this is expected to be a very small fraction as SGFP2 has very good maturation kinetics (*Kremers et al., 2007*). Secondly, our findings concerning the diffusion kinetics are limited by the assumptions we make in the FCS fitting model. Although obvious mistakes in underlying assumptions immediately become clear due to bad fitting results and can therefore be excluded, not every wrong assumption will stand out accordingly. Our data clearly shows that assuming only one diffusion speed for CTNNB1 in HAP1 cells would be incorrect (*Figure 4*). However, whether with the second, slower diffusion speed we measure a single distinct, large complex, or rather an average of multiple different CTNNB1 containing complexes cannot be determined in our current set-up. Moreover, the measured diffusion coefficients do not reveal the identity of the complexes. Previous studies have shown that a significant pool of CTNNB1 is associated with destruction complex components in presence and absence of WNT signaling (*Gerlach et al., 2014*; *Kitazawa et al., 2017*; *Li et al., 2012*), and it is therefore likely that at least part of the slow fraction of CTNNB1 we measure does indeed represent this destruction complex bound pool. As we discuss above, association and dissociation of the destruction complex with the proteasome offers one potential explanation for the different diffusion coefficients measured in the cytoplasm in WNT 'ON' and 'OFF' conditions. However, in both conditions other processes, such as transient association with intracellular structures (e.g. vesicular membranes or cytoskeletal components), could contribute to the diffusion coefficients we observe.

In addition, we assume that CTNNB1 is present as a free-floating monomer (as fixed for our first component), based on previous observations (*Gottardi and Gumbiner, 2004*; *Maher et al., 2010*) and further supported by unbiased fitting of our data. However, at least one report suggests that CTNNB1 is not present as a monomer but rather in small cytoplasmic complexes of ~200 kDa (*Gerlach et al., 2014*). As diffusion speed is relatively insensitive to differences in size (e.g. an eight-fold increase in protein mass is expected to result in only a twofold reduction of the diffusion coefficient for a spherical particle), it is possible that we do not measure truly free-floating CTNNB1, but rather one or more of these smaller complexes. In addition, point FCS is limited to a single position in the cell. Therefore, in addition to the intercellular differences in the WNT signaling response of individual cells, our measurements also sample intracellular heterogeneity caused by the presence of organelles and molecular crowding. Notwithstanding these limitations, we have been able to show that a large portion of CTNNB1 is present in a very large complex in both stimulated and unstimulated conditions and that this complex has a statistically and biologically significant different speed after WNT3A treatment and upon oncogenic mutation of CTNNB1.

The biophysical parameters we obtained from point FCS and N and B have taught us more about the speed and occupancy of the SGFP2-CTNNB1 complexes in living cells. Moreover, using different stimuli and perturbations of the pathway we have been able to link this to the phosphorylation status of CTNNB1. However, FCS and N and B do not provide conclusive evidence on the identity and composition of these complexes. An exciting possibility would be to label additional components presumed to be present in the CTNNB1-containing complexes at the endogenous level to uncover the precise composition and stoichiometry of protein complexes involved in WNT signaling. For instance, Fluorescence Cross Correlation Spectroscopy (FCCS) could be employed to test if two

proteins reside within the same complex (*Elson, 2011*; *Hink, 2014*; *Macháň and Wohland, 2014*). Ultimately, a combination of such quantitative functional imaging techniques, biochemical and proteomic approaches, together with additional perturbations will need to be employed to further our understanding of the dynamic composition of endogenous CTNNB1 complexes, as well as to help us resolve the molecular mechanism underlying nucleocytoplasmic shuttling and nuclear retention. As both genome editing and live cell imaging techniques continue to improve, additional possibilities will open up to address longstanding questions in cellular signaling in a physiological context with high spatial and temporal resolution. New opportunities and challenges await as these investigations extend to 3D organoid cultures, developing embryos and living organisms.

# Materials and methods

## Key resources table

| Reagent type (species) or resource | Designation | Source or reference | Identifiers | Additional information |
|---|---|---|---|---|
| Cell line (*Homo sapiens*) | HAP1 | Whitehead Institute | Cellosaurus: CVCL_Y019 | kind gift from Thijn Brummelkamp (NKI) |
| Cell line (*Homo sapiens*) | HAP1^SGFP2-CTNNB1 | This paper | | |
| Cell line (*Homo sapiens*) | HAP1^SGFP2-CTNNB1(S45F) | This paper | | |
| Transfected construct (*Homo sapiens*) | pSpCas9(BB)—2A-Puro (PX459) V2.0 | *Ran et al., 2013* | RRID:Addgene_62988 | |
| Transfected construct (*Homo sapiens*) | MegaTopflash | *Hu et al., 2007* | | kind gift from Christophe Fuerer and Roel Nusse, Stanford University |
| Transfected construct (*Homo sapiens*) | CMV Renilla | Promega | E2261 | |
| Transfected construct (*Homo sapiens*) | pSGFP2-C1 | *Kremers et al., 2007* | RRID:Addgene_22881 | kind gift from Dorus Gadella |
| Transfected construct (*Homo sapiens*) | pmScarlet-i_C1 | *Bindels et al., 2017* | RRID:Addgene_85044 | kind gift from Dorus Gadella |
| Transfected construct (*Homo sapiens*) | pSYFP2-C1 | *Kremers et al., 2006* | RRID:Addgene_22878 | kind gift from Dorus Gadella |
| Transfected construct (*Homo sapiens*) | mTurquoise2-C1 | *Goedhart et al., 2012* | RRID:Addgene_54842 | kind gift from Dorus Gadella |
| Transfected construct (*Homo sapiens*) | pEGFP | Clontech | | |
| Transfected construct (*Homo sapiens*) | pEGFP$_2$ | *Pack et al., 2006* | | kind gift from Masataka Kinjo |
| Transfected construct (*Homo sapiens*) | pEGFP$_3$ | *Pack et al., 2006* | | kind gift from Masataka Kinjo |
| Transfected construct (*Homo sapiens*) | pBluescript II KS(+) | Stratagene | | |
| Transfected construct (*Homo sapiens*) | pX459-CTNNB1-ATG | This paper | RRID:Addgene_153429 | |

*Continued on next page*

*Continued*

| Reagent type (species) or resource | Designation | Source or reference | Identifiers | Additional information |
|---|---|---|---|---|
| Transfected construct (*Homo sapiens*) | pX459-CTNNB1-S45 | This paper | RRID:Addgene_164587 | |
| Transfected construct (*Homo sapiens*) | pRepair-SGFP2-CTNNB1 | This paper | RRID:Addgene_153430 | |
| Recombinant DNA reagent | pRepair-mScI-CTNNB1 | This paper | RRID:Addgene_153431 | |
| Recombinant DNA reagent | pRepair-SYFP2-CTNNB1 | This paper | RRID:Addgene_153432 | |
| Recombinant DNA reagent | pRepair-mTq2-CTNNB1 | This paper | RRID:Addgene_153433 | |
| Chemical compound, drug | CHIR99021 | Biovision | 1677–5 | 6 mM stock in DMSO |
| Peptide, recombinant protein | Recombinant Mouse Wnt-3a | R and D systems | 1324-WN-002 | 10 µg/ml stock solution in 0.1% BSA in PBS |
| Chemical compound, drug | Dapi | Invitrogen | D1306 | |
| Chemical compound, drug | Vybrant DyeCycle Violet Stain | Invitrogen | V35003 | |
| Chemical compound, drug | Vybrant DyeCycle Ruby Stain | Invitrogen | V10273 | |
| Antibody | Non-phosphorylated (Active) β-catenin clone D13A1 (Rabbit monoclonal) | Cell Signaling | 8814S RRID:AB_11127203 | WB (1:1000) |
| Antibody | Total β-catenin clone 14 (mouse monoclonal) | BD | 610153 RRID:AB_397554 | WB (1:2000) |
| Antibody | α-Tubulin clone DM1A (mouse monoclonal) | Sigma-Aldrich | T9026 RRID:AB_477593 | WB (1:1000) |
| Antibody | GFP Antibody (Rabbit polyclonal) | Invitrogen | A-6455 RRID:AB_221570 | WB (1:1000) |
| Antibody | IRDye 680LT Goat anti-Rabbit IgG | LI-COR | 926–68021 RRID:AB_10706309 | WB (1:20,000) |
| Antibody | IRDye 800CW Donkey anti-Mouse IgG | LI-COR | 926–32212 RRID:AB_621847 | WB (1:20,000) |
| Chemical compound, drug | SiR-DNA | Spirochrome | SC007 | |
| Chemical compound, drug | Alexa Fluor 488 NHS Ester | Molecular probes | A20000 | |
| Software, algorithm | FlowJo | | | |
| Software, algorithm | CellProfiler pipeline | This paper | | available at https://osf.io/6pmwf/ |
| Software, algorithm | FIJI/ImageJ | | | |
| Software, algorithm | FFS Dataprocessor version 2.3 | SSTC | | |
| Software, algorithm | *ptu* converter | *Crosby et al., 2013* | | |
| Software, algorithm | ImageJ macro script | modified from *Crosby et al., 2013* | | available at https://osf.io/ys5qw/ |
| Software, algorithm | PlotsOfDifferences | *Goedhart, 2019* | | https://huygens.science.uva.nl/PlotsOfDifferences/ |
| Software, algorithm | RStudio | | | |

*Continued on next page*

*Continued*

| Reagent type (species) or resource | Designation | Source or reference | Identifiers | Additional information |
|---|---|---|---|---|
| Software, algorithm | R script | This paper | | available at https://osf.io/sxakf/ |
| Software, algorithm | R shiny app | This paper | WNT_minimal_ model_v2.4.R | app available at https://wntlab.shinyapps.io/ WNT_minimal_model/, source script available at https://osf.io/27ya6/ |

## DNA constructs

The following constructs were used: pSpCas9(BB)−2A-Puro (PX459) V2.0 (*Ran et al., 2013*, a kind gift from Feng Zhang, available from Addgene, plasmid #62988), MegaTopflash (*Hu et al., 2007*, a kind gift from Christophe Fuerer and Roel Nusse, Stanford University), CMV Renilla (E2261, Promega, Madison, WI), pSGFP2-C1 (*Kremers et al., 2007*, a kind gift from Dorus Gadella, available from Addgene, plasmid #22881), pmScarlet-i_C1 (*Bindels et al., 2017*, a kind gift from Dorus Gadella, available from Addgene, plasmid #85044), pSYFP2-C1 (*Kremers et al., 2006*, a kind gift from Dorus Gadella, available from Addgene, plasmid #22878), mTurquoise2-C1 (*Goedhart et al., 2012*, a kind gift from Dorus Gadella, available from Addgene, plasmid #54842), pEGFP (Clontech, Mountain View, CA), pEGFP$_2$ and pEGFP$_3$ (*Pack et al., 2006*, a kind gift from Masataka Kinjo), and pBluescript II KS(+) (Stratagene, La Jolla, CA).

The gRNA targeting the start codon in exon2 of human *CTNNB1* was designed using the MIT webtool (crispr.mit.edu) and cloned into pX459. Oligos RVA567 and RVA568 (*Table 6*), encoding the gRNA, were annealed and ligated into BbsI-digested pX459 plasmid as previously described (*Ran et al., 2013*) to obtain pX459-CTNNB1-ATG. The gRNA targeting codon 3 of *CTNNB1* for mutagenesis of Serine 45 to Phenylalanine (S45F) was similarly designed and cloned by introducing RVA561 and RVA562 (*Table 6*) into pX459, yielding pX459-CTNNB1-S45.

The repair plasmid for SGFP2-CTNNB1 (pRepair-SGFP2-CTNNB1) was cloned using Gibson cloning (*Gibson et al., 2009*). SGFP2 was chosen for its favorable brightness, maturation and photo-stability (*Kremers et al., 2007*). First, a repair plasmid including the Kozak sequence from the pSGFP2-C1 plasmid was generated (pRepair-Kozak-SGFP2 -CTNNB1). For this, 5' and 3' homology arms were PCR amplified from genomic HEK293A DNA with primers RVA618 and RVA581 (5' arm) or RVA619 and RVA584 (3' arm). SGFP2 was amplified with Gibson cloning from pSGFP2-C1 with primers RVA582 and RVA583 and the backbone was amplified from SacI digested pBlueScript KS(+) with primers RVA622 and RVA623. The final repair construct (pRepair-SGFP2-CTNNB1) contains the endogenous *CTNNB1* Kozak sequence before the SGFP2 ATG. To obtain pRepair-SGFP2-CTNNB1, the backbone and homology regions were amplified from pRepair-SGFP2-Kozak-CTNNB1 with primers RVA1616 and RVA1619, and an SGFP2 without the Kozak sequence was amplified from pSGFP2-C1 with primers RVA1617 and RVA1618. To generate color variants of the repair plasmid SYFP2, mScarlet-i and mTurquoise2 (not used in this publication, but available from Addgene, see below) were also amplified from their respective C1 vectors with primers RVA1617 and RVA1618. PCR products were purified and assembled with a Gibson assembly master mix with a 1:3 (vector: insert) molar ratio. Gibson assembly master mix was either purchased (E2611S, NEB) or homemade (final concentrations: 1x ISO buffer (100 mM Tris-HCL pH 7.5, 10 mM MgCl2, 0.2M dNTPs (R0181, Thermo Scientific), 10 mM DTT (10792782, Fisher), 5% PEG-8000 (1546605, Sigma-Aldrich, St Louis, MO), 1 mM NAD+ (B9007S, NEB)), 0.004 U/μl T5 exonuclease (M0363S, NEB), 0.5 U/μl Phusion DNA Polymerase (F-530L, Thermo Scientific) and 4 U/μl Taq DNA ligase (M0208S, NEB)).

The following plasmids are available from Addgene: pX459-CTNNB1-ATG (#153429), pX459-CTNNB1-S45 (#164587), pRepair-SGFP2-CTNNB1 (#153430), pRepair-mScl-CTNNB1 (#153431), pRepair-SYFP2-CTNNB1 (#153432), pRepair-mTq2-CTNNB1 (#153433).

## Cell culture, treatment, and transfection

HAP1 cells (a kind gift from Thijn Brummelkamp (NKI), acknowledging the Whitehead Institute as the source of the material) were maintained in full medium (colorless IMDM (21056023, Gibco, Thermo

Fisher Scientific, Waltham, MA) supplemented with 10% FBS (10270106, Gibco) and 1X Glutamax (35050061, Gibco)) under 5% $CO_2$ at 37°C in humidifying conditions and passaged every 2–3 days using 0.25% Trypsin-EDTA (25200056, Gibco). Cells were routinely tested for mycoplasma. We verified the haploid identity of the parental HAP1[WT] by karyotyping of metaphase spreads. To maintain a haploid population, cells were resorted frequently (see below) and experiments were performed with low passage number cells.

Where indicated, cells were treated with CHIR99021 (6 mM stock solution in DMSO) (1677–5, Biovision, Milpitas, CA) or Recombinant Mouse Wnt-3a (10 µg/ml stock solution in 0.1% BSA in PBS) (1324-WN-002, R and D systems, Bio-Techne, Minneapolis, MN) with DMSO and 0.1% BSA in PBS as vehicle controls, respectively. In *Figure 3*, the range of WNT3A used was based on previous experiments in HEK293T cells (*Jacobsen et al., 2016*).

Cells were transfected using Turbofect (R0531, ThermoFisher, Thermo Fisher Scientific, Waltham, MA), X-tremeGene HP (6366546001, Roche, Basel, Switzerland), or Lipofectamine 3000 (L3000001, Invitrogen, Thermo Fisher Scientific, Waltham, MA) in Opti-MEM (31985070, Gibco) according to the manufacturer's instructions.

## HAP1[SGFP2-CTNNB1] and HAP1[SGFP2-CTNNB1(S45F)] generation

800,000 HAP1 cells/well were plated on six-well plates. The following day, cells were transfected with Turbofect and 2000 ng DNA. pX459-CTNNB1-ATG and pRepair-SGFP2-CTNNB1 were transfected in a 2:1, 1:1, or 1:2 ratio. pSGFP2-C1, pX459, or pX459-CTNNB1-ATG were used as controls. From 24 to 48 hr after transfection, cells were selected with 0.75 µg/ml puromycin (A1113803, Gibco). Next, cells were expanded and passaged as needed until FACS sorting at day 9. For FACS analysis and sorting, cells were washed, trypsinized, resuspended with full medium and spun down at 1000 rpm for 4 min. For sorting, cells were stained with 1 µg/ml Dapi (D1306, Invitrogen) in HF (2% FBS in HBSS (14175053, Gibco)), washed with HF and resuspended in HF. To determine the haploid population, a separate sample of cells was stained with 5 µM Vybrant DyeCycleTM Violet Stain (V35003, Invitrogen) in full medium for 30 min at 37°C and kept in vibrant containing medium. Cells were filtered with a 70 µm filter and then used for FACS sorting and analysis on a FACSARIA3 (BD, Franklin Lanes, NJ). Vybrant-stained cells were analyzed at 37° and used to set a size gate only containing haploid cells. Dapi-stained cells were single cell sorted at 4°C into 96-well plates, that were previously coated overnight with 0.1% gelatin (G9391, Sigma-Aldrich) in MQ and contained full medium supplemented with 1% penicillin/streptomycin (15140122, Gibco) and 0.025 M HEPES (H3375 Sigma-Aldrich, 1 M stock solution, pH 7.4, filter sterilized). The three independent clones used in this study were obtained from separate transfections of the same parental cell line. Clones were genotyped and sanger sequenced using primers RVA555 and RVA558 (*Table 6*).

HAP1[SGFP2-CTNNB1(S45F)] were generated from HAP1[SGFP2-CTNNB1] clone 1. The same procedure as above was followed with slight adaptations; Cells were transfected 1000 ng pX459-CTNNB1-S45 or pX459 with 2 or 4 µl 10 mM repair oligo (RVA 2540) with Turbofect, selected with puromycin and expanded as described above. Haploid single cells were sorted after 11 days as described above. For haploid size discrimination Vybrant DyeCycle Ruby Stain (V10273, Invitrogen) was used. The five clones used in this study were obtained from two separate transfection (clones 2, 3, 16, 24 from the same transfection, clone 27 from a second transfection). Clones were genotyped using primers RVA555 and RVA558 (*Table 6*), followed by HpaII (ER0511, ThermoFisher) restriction as per the manufacturer's instruction. RVA555 was used for sanger sequencing.

Resorting of the cell lines was also performed with the same FACS procedure, with collection of cells in 15 mL tubes containing full medium with 1% penicillin and 0.025 M HEPES.

FACS data were analyzed and visualized with FlowJo.

## Luciferase assay

For luciferase assays, 100,000 cells per well were seeded on a 24-well plate. Cells were transfected with 1 µl X-tremeGene HP and 400 ng MegaTopflash reporter and 100 ng CMV-Renilla, or 500 ng SGFP2-C1 as a negative control 24 hr later. Cells were treated with the indicated concentration of CHIR99021 24 hr after transfection and after another 24 hr medium was removed and the cells were harvested with 50 µl Passive Lysis Buffer (E1941, Promega). Luciferase activity was measured on a GloMax Navigator (Promega), using 10 µl lysate in a black OptiPlate 96-well plate (6005279, Perkin

**Table 6.** primers/oligonucleotides used in this study.

| primer | sequence |
| --- | --- |
| RVA24 | CAAGTTTGTTGTAGGATATGCCC |
| RVA25 | CGATGTCAATAGGACTCCAGA |
| RVA124 | AGTGTGAGGTCCACGGAAA |
| RVA125 | CCGTCATGGACATGGAAT |
| RVA555 | GCCAAACGCTGGACATTAGT |
| RVA558 | AGACCATGAGGTCTGCGTTT |
| RVA561 | CACCGTTGCCTTTACCACTCAGAGA |
| RVA562 | AAACTCTCTGAGTGGTAAAGGCAAC |
| RVA567 | CACCGTGAGTAGCCATTGTCCACGC |
| RVA568 | AAACGCGTGGACAATGGCTACTCAC |
| RVA581 | tgctcaccatggtgg<br>GATTTTCAAAACAGTTGTATGGTATACTTC |
| RVA582 | actgttttgaaaatcCCACCATGGTGAGCAAGGGC |
| RVA583 | agtagccattgtccaCTTGTACAGCTCGTCCATGCCG |
| RVA584 | gacgagctgtacaagTGGACAATGGCTACTCAAGGTTTG |
| RVA618 | atacgactcactatagggcgaattggagct<br>GATGCAGTTTTTTTCAATATTGC |
| RVA619 | ttctagagcggccgccaccgcggtggagct<br>CTCTCTTTTCTTCACCACAACATTTTATTTAAAC |
| RVA622 | AAGAGAGAGCTCCACCGCGGTGGCGGCCG |
| RVA623 | TGCATCAGCTCCAATTCGCCCTATAGTGAGTCG |
| RVA1616 | tgtccacgctgGATTTTCAAAACAGTTGTATGG |
| RVA1617 | atacaactgttttgaaaatccagcgtggaca<br>ATGGTGAGCAAGGGCGAG |
| RVA1618 | cacaaaccttgagtagccatCTTGTACAGCTCGTCCATGC |
| RVA1619 | ATGGCTACTCAAGGTTTGTGTCATTAAATC |
| RVA2540 | CTTACCTGGACTCTGGAATCCATTCTGGTGCCAC<br>TACCACAGCTCCTTTCCTGTCCGGTAAAGGCAAT<br>CCTGAGGAAGAGGATGTGGATACCTCCCAAGT |

Elmer, Waltham, MA) and 50 µL homemade firefly and luciferase reagents (according to *Fuerer et al., 2014*; *Hampf and Gossen, 2006*).

For luciferase assays, three technical replicates (i.e. three wells transfected with the same transfection master mix) were pipetted and measured for each sample in each experiment. For each technical triplicate, the average MegaTopflash activity was calculated and depicted as a single dot in *Figure 2C* and *Figure 7—figure supplement 1G*. Three independent biological experiments, each thus depicted as an individual dot, were performed. To calculate MegaTopflash activity, Renilla and Luciferase luminescence values were corrected by subtracting the average background measured in the SGFP2-transfected control. MegaTopflash activity was calculated as the ratio of corrected Firefly and Renilla luminescence and normalized to the average reporter activity of the relative DMSO control (*Figure 2C*) or WT DMSO control (*Figure 7—figure supplement 1G*).

## Western blot

The remaining lysates from the technical triplicates of the luciferase assay were combined and they were cleared by centrifugation for 10 min at 12,000 g at 4°C. Western blot analysis was performed and quantified as previously described (*Jacobsen et al., 2016*). Antibodies were used with the following dilutions, 1:1000 Non-phosphorylated (Active) β-catenin clone D13A1 (8814S, Cell Signaling, Danvers, MA), 1:2000 total β-catenin clone 14 (610153, BD), 1:1000 α-Tubulin clone DM1A (T9026,

Sigma-Aldrich), 1:1000 GFP polyclonal (A-6455, Invitrogen), 1:20,000 IRDye 680LT Goat anti-Rabbit IgG (926–68021, LI-COR, Lincoln, NE), 1:20,000 IRDye 800CW Donkey anti-Mouse IgG (926–32212, LI-COR). Raw data for all blots have been made available at https://osf.io/vkexg/.

## qRT-PCR

For qRT-PCR analysis, 100,000 HAP1 cells per well were seeded on a 24-well plate. After 48 hr, cells were treated with indicated concentrations of CHIR99021. Cells were harvested 24 hr after treatment. RNA was isolated with Trizol (15596018, Invitrogen) according to the manufacturer's instructions. cDNA was synthesized using SuperScriptIV (18090010, Invitrogen) according to the manufacturer's instructions. qRT-PCR was performed with SyberGreen (10710004, Invitrogen). The endogenous WNT target gene *AXIN2* was amplified using primers RVA124 and RVA125, and HPRT housekeeping control was amplified using primers RVA24 and RVA25.

For qRT-PCR experiments, three technical replicates (i.e. three reactions with the same cDNA) were pipetted and measured for each sample in each experiment. For each technical triplicate, the mean fold-change in AXIN2 expression was calculated and depicted as a single dot in *Figure 2D* and *Figure 7—figure supplement 1H*. Three independent biological experiments, each thus depicted as an individual dot, were performed. Relative expression levels of *AXIN2* were calculated using the comparative Delta-Ct method (*Livak and Schmittgen, 2001*; *Schmittgen and Livak, 2008*). Briefly, *AXIN2* expression was normalized for *HPRT* expression and then the relative fold-change to a WT DMSO sample was calculated for all clones and conditions.

## Time-lapse imaging

The day before imaging, 88,000 cells/well were seeded on an eight-well chamber slide with glass bottom (80827–90, Ibidi, Gräfelfing, Germany). HAP1$^{SGFP2-CTNNB1}$ clone 2 was used for the main *Figure 3*, all three clones were used for *Figure 3—figure supplement 2*. HAP1$^{SGFP2-CTNNB1(S45F)}$ clone 2 was imaged for *Figure 8A*. Approximately 6 hr before imaging, medium was replaced with full medium supplemented with 1% penicillin/streptomycin, 0.025M HEPES and 500 nM SiR-DNA (SC007, Spirochrome, Stein am Rhein, Switzerland). Time lapse experiments were performed on an SP8 confocal microscope (Leica Microsystems, Wetzlar, Germany) at 37°C with a HC PL APO CS2 63x/1.40 oil objective (15506350, Leica), 488 and 633 lasers, appropriate AOBS settings, using HyD detectors for fluorescent signal with a 496–555 bandpass for SGFP2-CTNNB1 and 643–764 bandpass for SiR-DNA, and a transmission PMT. Using multi-position acquisition, up to 24 images were captured every 5 min. Focus was maintained using AFC autofocus control on demand for every time point and position. Automated cell segmentation and intensity quantification was performed using a custom CellProfiler pipeline (made available at https://osf.io/6pmwf/). Output data was further analyzed in R/RStudio. Cells with a segmented cytoplasmic area of less than 10 pixels were excluded. Intensities were normalized per position to the average intensity in the cellular compartment (nucleus or cytoplasm) for that position before the addition of the compounds. The imaging settings resulted in low signal in regions not occupied by cells (~10% of the nuclear intensity, and ~5% of the cytoplasmic intensity in untreated cells), and the data was therefore not background corrected. The nuclear cytoplasmic ratio was calculated by dividing the raw nuclear intensity by the raw cytoplasmic intensity. Videos and still images were extracted with FIJI/ImageJ.

## FCS and N and B cell preparation and general settings

Two days before FCS and N and B experiments, 44,000 cells/well were seeded on an eight-well chamber slide with a glass bottom (80827–90, Ibidi). For low, FFS-compatible expression of control samples, HAP1$^{WT}$ cells were transfected with ~5 ng pSGFP2-C1, pEGFP (monomer), pEGFP$_2$ (dimer), or pEGFP$_3$ (trimer) and ~200 ng pBlueScript KS(+) per well with Turbofect, X-tremeGene HP or Lipofectamine 3000 the day before the experiment. Lipofectamine 3000 yielded the best transfection efficiency. For *Figures 4*, *5* and *8K* and accompanying supplements, HAP1$^{SGFP2-CTNNB1}$ clone 2 was used. For *Figures 7* and *8* and accompanying supplements, CHIR99021 data was recorded and pooled for all three HAP1$^{SGFP2-CTNNB1}$ clones, and S45F data was recorded and pooled from HAP1$^{SGFP2-CTNNB1(S45F)}$ clones 2, 24, and 27 and HAP1$^{SGFP2-CTNNB1}$ clone 1 (the parental line for these S45F mutant clones) was used as the wild-type control.

FCS and N and B measurements were performed on an Olympus FV-1000 equipped with SepiaII and PicoHarp 300 modules (Picoquant, Berlin, Germany) at room temperature. An Olympus 60x water immersed UPLS Apochromat (N.A. 1.2) objective was used for FCS acquisition and *Figure 3—figure supplement 2E*, and an Olympus 60x silicon immersed UPLS Apochromat (N.A. 1.4) objective was used for N and B measurements. Green fluorophores were excited with a 488 nm diode laser (Picoquant) pulsing at 20 MHz and detected through a 405/480-488/560/635 nm dichroic mirror (Chroma, Bellows Falls, VT) and 525df45 nm bandpass filter (Semrock, Rochester, NY) with an Avalanche Photodiode (APD) (MPD, Bolzano, Italy). For *Figure 2—figure supplement 1E* and for FCS and N and B reference images, the same laser and dichroic were used, but the signal was detected through a 505–540 bandpass filter with an internal PMT of the FV-1000 Olympus microscope.

## FCS data acquisition and analysis

For FCS measurements, a confocal image was recorded. In this reference image, a single pixel was set as region of interest (ROI), based on its localization in the cytoplasm or nucleus as judged by the transmission image. In this ROI, the fluorescence intensity was measured over time using an APD, for typically 120 s.

FCS measurements were analyzed in FFS Dataprocessor version 2.3 (SSTC, Minsk, Belarus). The autocorrelation curve ($G()$) was calculated from the measured intensity (I) according to *Equation 1*. Intensity traces with significant photobleaching, cell movement or focal drift were excluded from further analysis (see *Supplementary file 1* – tab FCS measurements and fitting). From other traces, a portion of the trace with minimal (less than 10%) intensity drift or bleaching was selected to generate autocorrelation curve (AC).

$$G() = 1 + \frac{<\delta I(t)*\delta I(t+\tau)>}{<I>^2} \tag{1}$$

The resulting AC was fitted with a Triplet-state-diffusion model, described in *Equation 2*. $G_\infty$ accounts for offset in the AC for example by intensity drift. N is the average of the number of particles that reside in the confocal volume. $F_{trip}$ and $\tau_{trip}$ describe the fraction of molecules in the dark state and the relaxation of this dark state respectively. Of note, in this case, $F_{trip}$ and $\tau_{trip}$ account both for blinking of the fluorescent molecules and for the afterpulsing artefact of the APD. $\tau_{diff,i}$ describes the diffusion rate of the fluorescent molecules with the corresponding fraction, $F_i$. This diffusion time depends on the structural parameter (sp), which is defined as the ratio of the axial ($\omega_z$) over the radial axis ($\omega_{xy}$) of the observation volume.

$$G(\tau) = G_\infty + \frac{1}{<N>} * \frac{F_{trip}}{1-F_{trip}} e^{\frac{-\tau}{\tau_{trip}}} * \sum_j \frac{F_i}{\left(1+\frac{\tau}{\tau_{diff,i}}\right)\sqrt{1+\frac{\tau}{\tau_{diff,i}*sp^2}}} \tag{2}$$

The apparent particle numbers ($N_{apa}$) for SGFP2-CTNNB1 were corrected for autofluorescence and bleaching (*Equation 3*). The autofluorescence ($I_{autofluorescence}$) of HAP1 cells in the nucleus and cytoplasm was measured in untransfected HAP1 cells using the same settings as for FCS measurements. The correction for moderate bleaching is based on the intensity of the selected portion of the intensity trace for AC calculation ($I_{ana}$) and the intensity at the start of the measurement ($I_{start}$).

The size and shape of the observation volume was calibrated daily by measuring Alexa Fluor 488 NHS Ester (A20000, Molecular probes, Thermo Scientific, stock dilution in MQ) in PBS in a black glass-bottom cell imaging plate with 96 wells (0030741030, Eppendorf, Hamburg, Germany). From the FCS measurements of Alexa488, the $\tau_{diff}$ and sp were determined by fitting with a single diffusion and blinking component. The diffusion coefficient (D) of Alexa488 in aqueous solutions at 22.5° C is 435 $\mu m^2 s^{-1}$ (*Petrásek and Schwille, 2008*). From these parameters, the axial diameter can be determined with *Equation 4* and the volume can be approximated by a cylinder (*Equation 5*). This allows for transformation of particle numbers to concentrations (*Equation 6*) and diffusion times to diffusion coefficients (*Equation 4*) that are independent of measurement settings and small daily changes in alignment of the microscope.

$$N_{corr} = N_{apa} * \left[1 - \frac{I_{autofluorescence}}{I_{total}}\right]^2 * \left[\frac{I_{start}}{I_{ana}}\right] \tag{3}$$

$$\tau_{diff} = \frac{\omega_{xy}^2}{4D} \tag{4}$$

$$V = 2\pi\omega_{xy}^3 * sp \tag{5}$$

$$C = \frac{N_{corr}}{V * N_A} \tag{6}$$

The model to fit SGFP2-CTNNB1 measurements contained two diffusion components. The first diffusion component was fixed to the speed of monomeric SGFP2-CTNNB1. To estimate the speed of monomeric SGFP2-CTNNB1, the speed of free floating SGFP2, transfected in HAP1 cells, was measured to be 24.1 $\mu m^2 s^{-1}$ using FCS. Subsequently, this speed was used to calculate the speed of monomeric SGFP2-CTNNB1 with Einstein-Stokes formula (*Equation 7*).

$$D = \frac{k_B T}{6\pi\eta r} \tag{7}$$

As the temperature (T), dynamic viscosity ($\eta$) and Boltzmann's constant ($k_B$) are equal between SGFP2 and SGFP2-CTNNB1 measurements, the expected difference in diffusion speed is only caused by the radius (r) of the diffusing molecule assuming a spherical protein. The difference in radius was approximated by the cubic root of the ratio of the molecular weight of the SGFP2-CTNNB1 fusion protein (88 + 27=115 kDa) and the size of the SGFP2 protein (27 kDa), thus expecting a 1.62 times lower diffusion coefficient (compared to free floating SGFP2) of 14.9 $\mu m^2 s^{-1}$ for SGFP2-CTNNB1.

It must be noted that, especially for larger protein complexes, the linearity between the radius of the protein and the speed is not ensured, if the shape is not globular, and due to other factors such as molecular crowding in the cell and hindrance from the cytoskeletal network. We therefore did not estimate the size of the measured CTNNB1 complexes (as this would inescapably introduce errors, given that the ideal circumstances underlying the Einstein-Stokes formula are not met in the cellular environment for a complex of this size), but rather compared them to measurements from other FCS studies. However, it is likely that the 3.5-fold change in the second diffusion coefficient of SGFP2-CTNNB1 in response to WNT3A treatment is indicative of a larger than 3.5-change to complex size.

In the fitting model, the structural parameter was fixed to the one determined by the Alexa488 measurements of that day. To ensure good fitting, limits were set for other parameters; $G_\infty$[0.5–1.5], N [0.001, 500], $\tau_{trip}$ [$1*10^{-6}$-0.05 ms], $\tau_{diff2}$[10–150 ms]. This model was able to fit most Autocorrelation Curves from FCS measurements. In case of clear misfits, as judged by the distribution of residuals around the fitted curves, the measurement was excluded (see *Supplementary file 1* – tab FCS measurements and fitting).

To validate the obtained first diffusion coefficient of 14.9 $\mu m^2 s^{-1}$ for SGFP2-CTNNB1, the data were tested with an unfixed two-component model where both the first and the second diffusion coefficient were fitted (shown in *Figure 4—figure supplement 1*). The following limits were set; $G_\infty$[0.5–1.5], N [0.001, 500], $\tau_{trip}$ [$1*10^{-6}$-0.05 ms], $\tau_{diff1}$[0.5–10 ms], $\tau_{diff2}$[10–150 ms]. This resulted in a median diffusion time for the first component of 14.8 $\mu m^2 s^{-1}$ (*Figure 4—figure supplement 1*), which was in line with our calculated diffusion coefficient of 14.9 $\mu m^2 s^{-1}$. All analyses were performed with the two-component model with the fixed first component to reduce variability.

## N and B data acquisition and analysis

As a control, and to optimize acquisition settings, HAP1 cells transfected with SGFP2, EGFP monomer, dimer or trimer were measured alongside HAP1[SGFP2-CTNNB1] cells treated with BSA, WNT3A, DMSO or CHIR99021, or HAP1[SGFP2-CTNNB1(S45F)] cells. SGFP2 and EGFP are highly similar in sequence (with only four amino acid changes) and in spectral and biochemical characteristics. SGFP2 has a slightly higher quantum yield (+7%), lower extinction coefficient (−27%), and enhanced protein expression and maturation compared to EGFP (*Kremers et al., 2007*). The resulting brightness of monomeric SGFP2 in comparison to monomeric EGFP is slightly lower (−5%). In cellular

measurements, this difference is within the biological and technical variation and therefore SGFP2 and EGFP controls can be considered synonymous in these experiments.

For N and B analysis, 50 images were acquired per measurement with a pixel time of 100 µs/pixel and a pixel size of 0.138–0.207 µm. The fluorescent signal was acquired with the APD described above for the FCS measurements. APD readout was converted to a TIF stack using a custom build .*ptu* converter (*Crosby et al., 2013*). This TIF stack was further analyzed using an ImageJ macro script (modified from *Crosby et al., 2013*, made available at https://osf.io/ys5qw/) based on *Digman et al., 2008*. Within the script, average brightness and particle numbers were calculated for nuclear or cytoplasmic ROIs, which were set based on transmission image (see *Figure 5F*). Static or slow-moving particles, including membrane regions, were excluded by thresholding and/or ROI selection, since they can severely impact the brightness measured.

Data were further analyzed in R/RStudio. Brightness was normalized to the median value of the EGFP-monomer brightness measured on the same day in the same cellular compartment (nucleus/cytoplasm). Of note, FCS and N and B analysis models assume a different confocal volume. In FCS we assume a cylinder with factor $\gamma=1$, whereas in N and B we assume a 3D-Gauss with factor $\gamma=0.3536$. To be able to compare particle numbers obtained with both techniques, particle numbers obtained with N and B were divided by the factor $\gamma=0.3536$.

## Data representation and statistical analysis

Data processing and representation were performed in RStudio (version 1.1.456 running R 3.5.1 or 3.6.1). 95% confidence intervals of the median mentioned in the text and shown in *Tables 1–2* and *Supplementary file 1* were calculated using PlotsOfDifferences (*Goedhart, 2019*). The p-values in *Table 3* and *Supplementary file 1* were also calculated using PlotsOfDifferences, which uses a randomization test and makes no assumption about the distribution of the data. Representation of the imaging data in *Figure 3—figure supplement 1* and in Videos 4-6 were generated in RStudio using a script based on PlotsOfDifferences (made available at https://osf.io/sxakf/).

## Model description

We developed a minimal model for WNT signaling based on a previous model from the Kirschner group (*Lee et al., 2003*). The model is available as an interactive app at and the R https://wntlab.shi-nyapps.io/WNT_minimal_model/ source code of the model is available at https://osf.io/27ya6/ (WNT_minimal_model_v2.4.R).

Our minimal model comprises the following reactions:

$$\mathrm{CB} + \mathrm{DC} \underset{\mathrm{k1}}{\overset{\mathrm{k2}}{\rightleftharpoons}} \mathrm{CB}^* - \mathrm{DC} \; \text{Binding of cytoplasmic CTNNB1 (CB) to destruction complex} \tag{1a}$$

$$\mathrm{CB}^* - \mathrm{DC} \overset{\mathrm{k3}}{\rightarrow} \mathrm{DC} + \mathrm{CB}^* \; \text{Release of phosphorylated CB (CB*) and recycling of the destruction complex} \tag{2a}$$

$$\mathrm{DVL} + \mathrm{DC} \underset{\mathrm{k4}}{\overset{\mathrm{k5}}{\rightleftharpoons}} \mathrm{DC}^* \; \text{Inactivation of the destruction complex by DVL} \tag{3a}$$

$$\mathrm{CB} \underset{\mathrm{k6}}{\overset{\mathrm{k7}}{\rightleftharpoons}} \mathrm{NB} \; \text{Nucleocytoplasmic shuttling of CB to and from the nucleus} \tag{4a}$$

$$\mathrm{NB} + \mathrm{TCF} \underset{\mathrm{k8}}{\overset{\mathrm{k9}}{\rightleftharpoons}} \mathrm{NB} - \mathrm{TCF} \; \text{Binding of NB to TCF} \tag{5a}$$

Below, we show the differential equations that govern the concentrations of the different compounds over time for the reactions described above. *Table 4* in the main text gives the correspondence between the variables (i.e. $x_1$) in the differential equations and the model name (i.e. CB) in the reactions. The parameter $w$ in *Equation 7a* and *Equation 9* is $w=0$ in the absence of WNT and $w=1$ if WNT is present, that is in our minimal model the inactive form of the destruction complex (DC*) is only present if WNT is present. The parameter $b$ in *Equation 6a* represents the constant production of CTNNB1, corresponding to $v_{12}$ in *Lee et al., 2003*.

$$\frac{dx_1}{dt} = -k_1 x_1 x_2 + k_2 x_3 - k_6 x_1 + k_7 x_5 + b \tag{6a}$$

$$\frac{dx_2}{dt} = -k_1 x_1 x_2 + (k_2 + k_3) x_3 - w(k_4 x_2 - k_5 x_4) \tag{7a}$$

$$\frac{dx_3}{dt} = k_1 x_1 x_2 - (k_2 + k_3) x_3 \tag{8}$$

$$\frac{dx_4}{dt} = w(k_4 x_2 - k_5 x_4) \tag{9}$$

$$\frac{dx_5}{dt} = k_6 x_1 - k_7 x_5 - k_8 x_5 x_6 + k_9 x_7 \tag{10}$$

$$\frac{dx_6}{dt} = -k_8 x_5 x_6 + k_9 x_7 \tag{11}$$

$$\frac{dx_7}{dt} = k_8 x_5 x_6 - k_9 x_7 \tag{12}$$

## Equilibrium conditions without WNT

The parameters in our model can in part be determined from our measurements of the equilibrium concentrations of CB, NB and their complexes, see *Tables 4–5* in the main text. Where we could not determine the parameters from our measurements, we used published values as indicated.

Under equilibrium conditions, the concentrations of the compounds do not change with time and the left-hand side of *Equations 6a - 12* is zero. From *Equations 10 and 11,* we can determine the ratio of the rate constants $k_6$ and $k_7$ from the measured values of $x_1$ and $x_5$:

$$k_6 x_1 = k_7 x_5 \Leftrightarrow \frac{k_6}{k_7} = \frac{x_5}{x_1} = \frac{87}{91} = 0.96 \tag{13}$$

From *Equations (6a), (8), (10) and (11)* we have:

$$-k_3 x_3 + b = 0 \Leftrightarrow k_3 = \frac{b}{x_3} = \frac{0.423}{62.5} = 0.0068 \text{min}^{-1} \tag{14}$$

Our reaction (1) corresponds closely to step 8 in Lee et al. therefore, we use the value of the dissociation constant $K_8 = 120$ nM from Lee et al. for our dissociation constant $K_1 = \frac{k_2}{k_1}$.

The concentration of the destruction complex is obtained from *Equation (1a)* under equilibrium conditions using *Equations (6a), (8), (10), (11) and (14)*

$$-k_1 x_1 x_2 + k_2 x_3 + b = 0$$

The value of $b$ is assumed to be small compared to the two other terms, so we calculate the concentration of the destruction complex as:

$x_2 = K_1 \frac{x_3}{x_1} = 120 \frac{62.5}{91} = 82.4 nM$. It was then verified in our interactive app that this value for the destruction complex is indeed consistent with the equilibrium conditions without WNT stimulation.

To calculate the dissociation constant for the NB-TCF complex, we estimate an equilibrium concentration for free TCF ($x_6$) from *Tan et al., 2012*. From their Figure 11, it is seen that the bound TCF concentration in equilibrium in the presence of WNT has about the same value as the initial free TCF concentration and that no initial bound TCF is present. However, we measured NB-TCF also in the initial state. Therefore, we consider the free TCF concentration value from Tan et al. as a lower bound for the estimate of total TCF. Also, from Figure 11 of *Tan et al., 2012*, we estimate that of the initial free TCF, a fifth remains in the nucleus as free TCF after WNT is turned on. We measured 86 nM NB-TCF in the nucleus after the application of WNT. This leads to an estimate of the total concentration of TCF, $TCF^0$, in the nucleus of: $[TCF^0] = 86 + 0.2 \times 86 = 103 nM$. If we assume that the

total TCF concentration does not change by the application of WNT, we calculate the dissociation constant of the NB-TCF complex from *Equation 12*:

$$k_8 x_5 \left(TCF^0 - x_7\right) = k_9 x_7 \Rightarrow \frac{k_9}{k_8} = K_2 = \frac{x_5(TCF^0 - x_7)}{x_7} = \frac{87 * 81}{22} = 320\,\text{nM} \tag{15}$$

## Equilibrium conditions with WNT

We model the action of WNT by deactivation of the destruction complex by DVL through reaction 3 by setting $w = 1$ in *Equations 7a and 9*. The dissociation constant of CB*-DC, $K_1$, is assumed not to change in the presence of WNT. The measurements of free CB and NB in equilibrium (see *Table 2*) give for the ratio of $k_6$ and $k_7$:

$$k_6 x_1 = k_7 x_5 \Leftrightarrow \frac{k_6}{k_7} = \frac{x_5}{x_1} = \frac{170}{145} = 1.17 \tag{16}$$

The value of the rate of decay of the phosphorylated complex CB*-DC, $k_3$, is found to be the same for the 'without WNT' situation:

$$-k_3 x_3 + b = 0 \Leftrightarrow k_3 = \frac{b}{x_3} = \frac{0.423}{62.5} = 0.0068\,\text{min}^{-1} \tag{17}$$

To uniquely determine the ratio of $k_4$ and $k_5$, we need the concentrations of the destruction complex DC and DC* neither of which we have access to. We can, however, fit this ratio with our model to the measured values of $x_1$ and $x_7$ and find $k_4/k_5 = 1.7$.

We again calculate the dissociation constant of the NB-TCF complex from *Equation (12)*, using the concentrations for NB and NB-TCF obtained with FCS.

$$k_8 x_5 \left(TCF^0 - x_7\right) = k_9 x_7 \Rightarrow \frac{k_9}{k_8} = K_2 = \frac{x_5(TCF^0 - x_7)}{x_7} = \frac{170 * 17}{86} = 33.6\,\text{nM} \tag{18}$$

Notice that we determined the ratios of the rate constants from the measured equilibrium values of free and bound CTNNB1 in the cytoplasm and the nucleus. This means that our rate constants are determined up to a multiplicative factor: the equilibrium equations do not change if all rate constants $k_i$ and the parameter $b$ are multiplied by the same factor, *Rate*. The factor *Rate* determines how fast our model system reaches equilibrium. By comparing the times equilibrium was reached by the cytoplasmic and nuclear SGFP2-CTNNB1 signals (*Figure 3B, C*) of about 4.5 hr, we fitted a factor *Rate* = 20 for our model.

Our model shows that the ratios of $k_6/k_7$ and $k_9/k_8$ are different for the conditions without and with WNT stimulation, suggesting a change in mechanism for nuclear shuttling of CTNNB1 and nuclear retention of CTNNB1 in going from the WNT 'off' situation to the WNT 'on' situation. It seems likely that such changes do not occur instantaneously. In our model we therefore allow a gradual rise in $k_5/k_4$ and a gradual transition of the ratios of $k_6/k_7$ and $k_9/k_8$ from WNT 'off' to the WNT 'on'. In our model, this is included by setting a parameter ('*Steep*') that indicates the time after application of WNT the transition from WNT 'off' parameter values to WNT 'on' parameter values is complete. The value that gives a good approximation of the experimentally observed concentration curves is *Steep = 150* minutes (*Figure 6B-F*).

## Acknowledgements

We thank the van Leeuwenhoek Centre for Advanced Microscopy (LCAM, Section Molecular Cytology, Swammerdam Institute for Life Sciences, University of Amsterdam) for the use of their facilities and LCAM staff for sharing their expertise and providing technical support, Gonzalo Congrains Sotomayor for support with FACS sorting, Jasmijn Span for cloning the color variants of the repair plasmid that have been deposited at Addgene and Sanne Lith for gRNA cloning as students in our lab, Marten Postma and Joachim Goedhart for assistance with data handling and analysis, Dorus Gadella for carefully reading the manuscript, and all colleagues for stimulating discussions and suggestions.

## Additional information

### Funding

| Funder | Grant reference number | Author |
|---|---|---|
| University of Amsterdam | MacGillavry fellowship | R van Amerongen |
| KWF Kankerbestrijding | ANW 2013-6057 | R van Amerongen |
| KWF Kankerbestrijding | 2015-8014 | R van Amerongen |
| Nederlandse Organisatie voor Wetenschappelijk Onderzoek | VIDI 864.13.002 | R van Amerongen |
| Nederlandse Organisatie voor Wetenschappelijk Onderzoek | OCENW.KLEIN.169 | R van Amerongen |

The funders had no role in study design, data collection and interpretation, or the decision to submit the work for publication.

### Author contributions

Saskia MA de Man, Conceptualization, Data curation, Formal analysis, Validation, Investigation, Visualization, Writing - original draft, Project administration; Gooitzen Zwanenburg, Conceptualization, Data curation, Software, Formal analysis, Validation, Investigation, Visualization, Methodology, Writing - original draft, Project administration, Writing - review and editing; Tanne van der Wal, Investigation, Project administration, Writing - review and editing; Mark A Hink, Resources, Software, Supervision, Validation, Writing - review and editing; Renée van Amerongen, Conceptualization, Software, Supervision, Funding acquisition, Validation, Methodology, Writing - original draft, Project administration, Writing - review and editing

### Author ORCIDs

Saskia MA de Man (iD) https://orcid.org/0000-0003-0906-5276
Gooitzen Zwanenburg (iD) https://orcid.org/0000-0002-2745-0285
Tanne van der Wal (iD) https://orcid.org/0000-0002-3042-6948
Renée van Amerongen (iD) https://orcid.org/0000-0002-8808-2092

### Decision letter and Author response

Decision letter https://doi.org/10.7554/eLife.66440.sa1
Author response https://doi.org/10.7554/eLife.66440.sa2

## Additional files

### Supplementary files

• Source data 1. Numerical data for *Figure 3B,C,D*, *Figure 3—figure supplement 2*, *Figure 3—figure supplement 3* and *Figure 8—figure supplement 1B,C,D*.

• Supplementary file 1. Tables of all summary statistics (mean, median, 95% confidence intervals, differences, p-values) of the FCS and N and B parameters show in *Figures 5*, *7* and *8* and accompanying supplements.

• Transparent reporting form

### Data availability

Source data: for numerical data points in Figures 2-5,7-8 are attached to this article. In addition a comprehensive overview of all numerical data (summary statistics; median, mean and 95% CI's) for the FCS and N&B experiments depicted in Figures 5, 7, 8 plus accompanying supplements and in Tables 1, 2 and 3 is provided in summary tables as Supplementary File 1. Raw data: Original FACS data (.fcs), Western blot data (.tif), confocal images (.tif), FCS data (.ptu- and. oif reference images), N&B data (.ptu/.tif and. oif reference images) have been provided on Open Science Framework (https://osf.io/dczx8/). Source code: scripts for the following have been made publicly available on

Open Science Framework (https://osf.io/dczx8/), as referenced in the materials and methods section: Cell profiler segmentation pipeline (Figure 3), R script based on PlotsOfDifference to generate Figure 3 supplement 2 and supplementary movies 4-6, ImageJ N&B analysis script (Figures 5,7 and 8), R source code for the computational model (Figure 6).

The following dataset was generated:

| Author(s) | Year | Dataset title | Dataset URL | Database and Identifier |
|---|---|---|---|---|
| de Man SMA, Zwanenburg G, van der Wal T, Hink MA, van Amerongen R | 2021 | Quantitative Imaging of CTNNB1 | https://osf.io/dczx8/ | Open Science Framework, 10.17605/OSF.IO/DCZX8 |

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
