## [Decision Letter]

**Acceptance summary:**

Wnt signaling plays critical roles in cell fate determination in essentially every tissue in all animals, regulates tissue homeostasis in many adult tissues, and is inappropriately activated in many human cancers. It has been the focus of research for decades, and we have an outline of signal transduction. However, remarkably, key questions remain controversial. Central among these are questions about the nature of the negative regulatory destruction complex, its mechanism of action and how it is turned down by Wnt signaling. Here Saskia and colleagues take a novel and very exciting approach to these questions, combining innovative quantitative live-cell imaging and computational modelling.

**Decision letter after peer review:**

[Editors’ note: the authors submitted for reconsideration following the decision after peer review. What follows is the decision letter after the first round of review.]

Thank you for submitting your work entitled "Quantitative live-cell imaging yields novel insight into endogenous WNT/CTNNB1 signaling dynamics" for consideration by *eLife*. Your article has been reviewed by 3 peer reviewers, and the evaluation has been overseen by a Reviewing Editor and a Senior Editor.

Our decision has been reached after consultation between the reviewers. Based on these discussions and the individual reviews below, we regret to inform you that your work will not be considered further for publication in *eLife*.

The authors investigate how cells respond to WNT signaling by altering β catenin (CTNNB1) dynamics. They generated a number of cell lines in which they use different light microscopy techniques (such as FCS and number and brightness measurements) to quantitatively investigate the diffusion behavior and complex formation of intracellular CTNNB1. The results are in general well explained, reasoned and technically well-controlled (except for some, which raised concerns that were pointed out by the reviewers). The main finding of the paper is that CTNNB1 seems to reside in slow-moving complexes (that exist both in the presence and absence of WNT) that become slightly more mobile after WNT addition. As pointed out by the reviewers, these results can be interpreted in different ways, and it is not clear whether these complexes represent the destruction complex (cytoplasm) or enhanceosome (nucleus). In summary, yet the work shows some technical proficiency which could address some critical issues in Wnt signaling, the authors would need to identify the issues that could be resolved by the technique and then design experiments to resolve them in the future.

So, given the relatively large number of major points of the reviews, in particular those that would require extensive extra work (namely, Rev. 1's point 2and3; Rev 2's specific comments 2and3; and Rev 3's points 1and3), we proceed at this point with a formal rejection of the manuscript so then the authors can decide whether they want to work out all these points and proceed with a new submission in the future, or in case they want this paper to be published in a more timely manner, they can submit elsewhere with the reviews herein included, which will be publicly posted in the biorxiv.

*Reviewer #1:*

CTNNB1 is a core component of canonical Wnt signalling that is frequently mutated in cancers. A constitutively active destruction complex (degradosome) binds and phosphorylates CTNNB1 earmarking it for proteasomal degradation, this complex is inactivated upon Wnt3a/GSK3β inhibition leading to CTNNB1 stabilisation and nuclear translocation. The authors have successfully employed CRISPR mediated endogenous tagging of CTNNB1 and determined its cellular concentration and diffusion dynamics in HAP1 cells, in both the cytoplasm and nucleus by live-cell imaging and analysis. They provide the relative subcellular CTNNB1 concentration for the nucleus and cytoplasm, like previous studies in other cell lines (Tan et al., 2012) and in *Xenopus* (Lee et al., 2003). In addition their results suggest CTNNB1 resides in slow moving complexes that persist upon Wnt but become slightly more mobile, these results are intriguing but raise several unanswered questions, such as whether these complexes represent the destruction complex (cytoplasm) or enhanceosome (nucleus). The work has been completed to a high standard but I have several concerns listed below.

1) The authors acknowledge significant cell-cell heterogeneity. This is particularly noticeable in Figure 4A upon Wnt3a and CHIR99021 treatment. Figure 4B suggests all cells are analysed regardless of heterogeneity and the only exclusion criteria mentioned in the methodology is cells with a cytoplasm of less than 10pixels. Figure 4C/D does not seem to reflect the variation observed in Figure 4A? What is the spread pre-normalisation before and after treatment? How is the relative increase in nuclear/cytoplasmic intensity affected by cell size? Nuclear and cytoplasmic area? This may affect the relative fold increase and the cytoplasmic area seems highly variable at the confluence of cells shown.

2) Using point FCS the authors determined two diffusion speeds corresponding to monomer and complexed CTNNB1 in both the nucleus and cytoplasm. A modest increase in cytoplasmic diffusion speed of complexed CTNNB1 was observed after Wnt3a (0.461μm2/s^-1^) but far from the speed of the monomer (14.9μm2/s^-1^) suggesting it remains complexed upon Wnt3a. In addition the fraction of complexed CTNNB1 (~40%) remains largely unaltered. Is the same true under CHIR299021 treatment? Point FCS samples a very small area of the cell cytoplasm/nucleus and therefore gives a small representation of the subcellular pool (which is likely heterogeneous), only a single point appears to have been analysed per-cell and within the 21 cells analysed clear outliers can be observed (Figure 6A/B), this has not been adequately discussed. What is the variation in diffusion measured at different points within a single cell? Some discussion has been made as to these complexes reflecting the destruction complex/proteasome or the enhanceosome but this really needs to be tested in order to make any conclusions about these observations. Especially as cytoplasmic complexes are maintained under Wnt conditions, this would challenge the notion that CTNNB1 disassociates from the destruction complex upon Wnt. Ideally endogenous tagging of other destruction complex components with a different flourophore would be done to address this, if these complexes do represent the destruction complex and remain bound after Wnt this would have significant implications for our understanding of complex inactivation and greatly enhance the manuscript.

3) The NandB analysis averages out monomeric and complexed CTNNB1 intensity across an image stack around a single ROI within each cell. The authors interpret Figure 6C to mean SGFP2-CTNNB1 is present as a monomer whether in a complex or not. This is based on the fact the relative brightness averages at 1.0 similar to a monomeric GFP control. However, the spread of relative brightness is large, and often less than <1 so a relative brightness of 1 cannot refer to a monomeric SGFP2-CTNNB1? Does cellular concentration affect relative brightness? If so, transiently expressed monomer and dimer GFP may not be the best controls. Aggregation is spatially homogeneous and limited by the diffusion rate of protein/complexes – which your FCS measurements suggest is consistent with a large complex. Thus a single average may not represent the diversity of protein complexes, eNandB could be used (Cutrale et al., 2019). As mentioned in point 3, like FCS, you are only sampling a small region of the cell, which may or may not contain a destruction complex for example. Super-resolution imaging techniques such a STORM or LLSM may help with visualisation of cell complex heterogeneity and give a different impression of complex occupancy. I don't think the NandB data is sufficient to say complexes don't exist that contain more than one SGFP2-CTNNB1 molecule.

4) The computational model relies on a number of assumptions determined in other studies that may not reflect the HAP1 cells used in this study. Lee et al., was performed in *Xenopus* and Tan et al., 2012 found a number of differences in their mammalian cell studies. Important information regarding the concentration of destruction complex components has also been omitted, this information is important for future comparisons of cell-type specific behaviours.

*Reviewer #2:*

The manuscript by S.M.A. de Man et al., presents a study on the cellular response to Wnt activation and on the intracellular kinetics of β catenin (CTNNB1). The authors have developed cell lines expressing GFP reporters of CTNNB1 using CRISPR CAS9. They present different convincing controls on the specificity of the reporter and decided to analyze the temporal behavior of the best reacting clone. Then, they investigate the temporal evolution of fluorescent signal in the cell cytoplasm and nucleus upon Wnt signaling activation. They quantify the kinetics of the relocalization of CTNNB1 from the cytoplasm to the nucleus upon different strength of activation of the Wnt signaling and GSK3 inhibition. Using FCS, they identify that a dual diffusion model fit better the experimental data than a classical single diffusion model, suggesting the presence of complexes of different sizes. They measure the diffusion parameters and concentrations of the complexes in the nucleus and in the cytoplasm.

Using a dynamical model, the authors reveal that, to recapitulate the experimental observations, the regulation of CTNNB1 upon Wnt signaling has to be controlled at three levels, the destruction complex, the nuclear transport and the binding affinity to the chromatin.

Overall, the study is solid, presenting novel information on the kinetics of CTNNB1 during Wnt signaling. The results are consistent with the classical view on the regulation of β catenin during Wht signaling. I believe it is of interest for a general cell and developmental biology readership such as the one of *eLife*. I have few comments essentially on the methodology.

– The authors have designed a new cell line allowing for tracing the kinetics of betacatenin over time following Wnt signaling activation. They follow the relative changes in concentration in the nucleus and cytoplasm upon activation of Wnt signaling. Normalized changes render difficult to evaluate if the difference in the increase in the cytoplasm and the nucleus is due to a higher increase in the nucleus or simply due the absence of betacatenin in the nucleus at the onset of the process therefore enhancing the quantification. A non-normalized plot showing the increase in grey levels in the nucleus and cytoplasm should be added to complement the quantification and identify the differences between nuclear and cytoplasmic β catenin. It would also help the reader to compare with the results of concentrations extracted from the FCS.

– The response in figure 4 upon Wnt signaling activation and GSK3 inhibition are different (with the absence of a plateau in the case of GSK3 inhibition). The explanation of this difference is unclear as it is. I would suggest the authors to details a bit more their thoughts on the reason of the difference.

Could this simply be that Wnt activation clusters just a subset of GSK3 at the membrane and that inhibition can reach higher level of depletion of GSK3 in the cytoplasm?

– How GSK3 inhibition treatment affects the FCS measurements, particularly concentrations and different complexes compositions? The differences with Wnt3 activation could provide additional information on the nature of the identified complexes.

– The dynamical model presented in the paper shows a non-monotonous change in the concentration of betacatenin in the cytoplasm after activation. This seems to be due to the kinetics of nuclear transport and does not seems to be present in the experimental observations. Can the authors comment on this point? Is there a way by modulating parameters associated to transport to suppress this discrepancy?

– Finally, the model is consistent with the experimental observations but the authors di not checked with any type of perturbation how the model would compare with the experiments. For instance, how does the model would compare with experiments in the case of GSK3 inhibition, or when nuclear transport is affected. Adding a perturbation case would significantly strengthen the connection between model and experiment and the message of the manuscript.

*Reviewer #3:*

1) As I state below the paper is carefully done using a difficult and sophisticated biophysical technique, FCS to assess the changes in β catenin diffusion within the cell following Wnt signaling. So it passes the test on being an original piece of work executed well. However what has been learned is quite limited. A few interactions, such as the slow diffusion in the cytoplasm can be interpreted several ways. It is very helpful to have concentrations in the nucleus and cytoplasm for β catenin for future modeling. They could have tried to use single cross correlation with labeled APC or axin or the proteasome to derive more important information about the path through the destruction sequence. But that may be too hard to ask for at this stage. They could have combined their measurements with appropriate mutants or knockouts. I come down close to the line, high on the importance of the problem and the methods and execution; lower on the current take home lesson.

2) The support for the somewhat limited conclusions is strong as it is.

3) There are some technical issues. There is some concern with the FCS data. Itself. Figure 5F and 5G are of some concern. The curve doesn't drop to 1 at long correlation time (>100ms) and there are big fluctuations in the region of short correlation times (<0.1 ms). This could be due to the very long time course (120s) used in the experiment. Have the authors tried to image the same spot multiple times in short interval (etc 10s), or try to analyze 10s sub-trace of the original long trace to see if the conclusions hold? This type of error could influence the calculation of the diffusion coefficient of complexes of CNNTB1. They also affect the quantification of concentration. In line 352-353 the authors mentioned the nuclear concentration of CNNTB1 increases 2.1 fold based on FCS measurement, which is smaller than the fluorescent intensity change. Is this the result of errors such as this?

[Editors’ note: further revisions were suggested prior to acceptance, as described below.]

Thank you for submitting your article "Quantitative live-cell imaging and computational modelling shed new light on endogenous WNT/CTNNB1 signaling dynamics" for consideration by *eLife*. Your article has been reviewed by 2 peer reviewers, and the evaluation has been overseen by a Reviewing Editor and Aleksandra Walczak as the Senior Editor. The following individual involved in review of your submission has agreed to reveal their identity: Mark Peifer (Reviewer #3).

Essential Revisions:

1) For free β-catenin, the authors have calculated the theoretical diffusion coefficient based on the measured diffusion coefficient of SGFP2 and the size difference between them. But it is possible that β-catenin will interact with other proteins in a specific or non-specific manner, which will make the β-catenin diffuse substantially slower than SGFP2. The free β-catenin diffusion coefficient will affect the fitted value of the slow diffusion component. I wonder if the authors could fit the autocorrelation curve directly with 2 component model without assuming a free β-catenin diffusion coefficient. On the other hand, if a predetermined free β-catenin diffusion coefficient is preferred to fit the autocorrelation curves, maybe it is better to make a ΔARM-β-catenin and use the diffusion coefficient of that protein as the diffusion coefficient of the fast component, or measure the diffusion coefficient of SGFP2-β-catenin directly in the cell extract. In addition, some of the free receptors have a surface diffusion coefficient in the range of 0.1 um2/s. Have the authors considered the possibility that the β-catenin complexes bind some large structure inside the cell, like ER membrane, showing a very slow diffusion coefficient?

2) The model that the author proposed has some logical flaws. In line 1042, equation (9), the authors write the equation w(k4 χ2 – k5 x4) as the mathematical form of Wnt activation, where w is applied to both k4 χ2 term and the k5 x4 term. The χ2 term represents the active destruction complex, and the x4 term represents the inactive destruction complex. This means Wnt will regulate both forms of the destruction complex the same way. And the ratio of χ2/x4 will not change after Wnt pathway activation. This way to model is logically flawed. The authors do show a response of the model after Wnt treatment, but this is because the destruction complex also has a substrate-bound fraction: x3. This fraction is not controlled directly by Wnt input.

3) It's better to use SGFP2 dimer and trimer as control instead of EGFP, in Figure 5H and 7E/F. The dimer and trimer constructs are important control for the average size and the percentage of bright fluorescent proteins will vary for different fluorescent proteins. I have no idea how SGFP2 compares to EGFP.

4) The data on the concentrations of β-catenin in the cytoplasm and nucleus before and after Wnt signaling and after inhibition of GSK3 are a major advance. However, I share the opinion of previous reviewer 2 (comment 1) that too much attention is paid to relative differences and not enough to absolute differences. There is strong evidence in the literature that one function of the destruction complex is to retain β-catenin in the cytoplasm (e.g. https://pubmed.ncbi.nlm.nih.gov/21471006/), and thus reduce the amount that can enter the nucleus. Their quantitative imaging (e.g. Figure 3A) reveals this quite clearly and combined with their FCS data suggest that the cytoplasmic pool includes this "bound" β-catenin and that this bound pool remains after Wnt stimulation. I would suggest making this clearer and putting it in the context of earlier work on cytoplasmic retention.

5) One model currently in the field is that the key regulated step in β-catenin destruction is not its phosphorylation but instead its transfer to the E3 ligase, an event that may be controlled by GSK3 phosphorylation of APC (e.g. https://pubmed.ncbi.nlm.nih.gov/22682247/;. https://pubmed.ncbi.nlm.nih.gov/26393419/; https://pubmed.ncbi.nlm.nih.gov/32129710/). In talking about this, the authors state: "Our data clearly show that a substantial fraction of CTNNB1 in the cytoplasm remains bound upon pathway stimulation (Figure 5D). This is not predicted by any of the above-mentioned models and challenges the long-held view that mainly monomeric CTNNB1 accumulates." However, in my mind the retention of high MW complexes after Wnt signaling seems like it is consistent with the model in the manuscripts cited above. Further, while model cartoons (I use them myself) indicate accumulation of monomeric β-catenin in the cytoplasm, in fact what seems more likely is that monomeric β-catenin that accumulates after Wnt signaling rapidly enters the nucleus and is "trapped" as part of high MW transcriptional complexes like those their data appear to document, and thus monomeric cytoplasmic levels may be relatively low. I think a more nuanced description of these data would be useful.

6) One major controversy in the field is the nature of the active destruction complex and its multimeric state. Some data (e.g. https://pubmed.ncbi.nlm.nih.gov/26124443/; https://pubmed.ncbi.nlm.nih.gov/26393419/; https://pubmed.ncbi.nlm.nih.gov/29641560/) suggest large multimeric assemblies while in vitro studies have suggested much smaller assemblies (e.g. https://pubmed.ncbi.nlm.nih.gov/32297861/). It was quite surprising that the authors did not review this literature-in fact the latter paper isn't even in the reference list. Their data, while not definitive, certainly has something to say about this discussion. That having been said, however, I do NOT agree with Reviewer 1 that extending this work to visualize tagged destruction complex components is essential. Of course, this is a very important next step, but this manuscript is already rich in novel data. The reduction in "size" of the potential destruction complexes after Wnt treatment is also consistent with recent observations in *Drosophila* (https://pubmed.ncbi.nlm.nih.gov/29641560/)

7) I think Reviewer 2 comment 2 (previous submission) and the authors response should be more clearly laid out in the manuscript. While many of us said for years that Wnt signaling "turns the destruction complex off", it is now abundantly clear that it only "turns the destruction complex down". For example, in both *Drosophila* and in cultured mammalian cells complete loss of GSK3 or APC leads to much higher levels of β-catenin than are seen after Wnt signaling.

8) The fact that their Number and Brightness data suggest few destruction complexes bind multiple β-catenins is very surprising. It doesn't fit my preconceptions and that is OK! I like the idea that usually throughput is rapid enough that multiple β-catenins don't accumulate (waterwheel model). It is surprising that Wnt signaling doesn't alter this but those are the data.

---

## [Author Response]

[Editors’ note: the authors resubmitted a revised version of the paper for consideration. What follows is the authors’ response to the first round of review.]

The authors investigate how cells respond to WNT signaling by altering β catenin (CTNNB1) dynamics. They generated a number of cell lines in which they use different light microscopy techniques (such as FCS and number and brightness measurements) to quantitatively investigate the diffusion behavior and complex formation of intracellular CTNNB1. The results are in general well explained, reasoned and technically well-controlled (except for some, which raised concerns that were pointed out by the reviewers). The main finding of the paper is that CTNNB1 seems to reside in slow-moving complexes (that exist both in the presence and absence of WNT) that become slightly more mobile after WNT addition. As pointed out by the reviewers, these results can be interpreted in different ways, and it is not clear whether these complexes represent the destruction complex (cytoplasm) or enhanceosome (nucleus). In summary, yet the work shows some technical proficiency which could address some critical issues in Wnt signaling, the authors would need to identify the issues that could be resolved by the technique and then design experiments to resolve them in the future.So, given the relatively large number of major points of the reviews, in particular those that would require extensive extra work (namely, Rev. 1's point 2and3; Rev 2's specific comments 2and3; and Rev 3's points 1and3), we proceed at this point with a formal rejection of the manuscript so then the authors can decide whether they want to work out all these points and proceed with a new submission in the future, or in case they want this paper to be published in a more timely manner, they can submit elsewhere with the reviews herein included, which will be publicly posted in the biorxiv.

After careful consideration, we felt increasingly confident that we could adequately and comprehensively address all of the reviewer comments – including the main criticisms that led to a formal rejection of our first submission. We have therefore decided to perform the extra work required and to proceed with a new submission for consideration by *eLife*.

Before addressing each of the specific reviewer comments in point-by-point detail (Rev.1: p.3-18, Rev.2: p.19-27, Rev.3 p.28-38) , we want to highlight the following:

We have performed extensive extra work to address Rev.1’s point’s 2and 3, Rev.2’s specific comments 2and 3 and Rev. 3’s points 1 and 3. We specifically want to highlight the following new data, which are shown in Figures 7 and 8 (and accompanying supplements):

1. To directly compare physiological (Wnt_off_/Wnt_on_) and oncogenic (i.e. constitutively active) signaling, we generated a second cell line using CRISPR/Cas9 genome editing, harboring an oncogenic point mutant form of CTNNB1 (SGFP2-CTNNB1^S45F^).

2. To further quantify the levels, complex state and multimerization status when WNT/CTNNB1 signaling is hyperactivated, we performed additional FCS and NandB experiments in the new mutant cell line and upon GSK3B inhibition by CHIR99021 treatment (as requested by reviewers 1 and 2).

3. We use these same perturbations to strengthen the link between our experimental data and the computational model (as suggested by reviewer 2) and provide access to the model in the form of an interactive app (available at https://wntlab.shinyapps.io/WNT_minimal_model/).

We also suspect that we didn’t sufficiently highlight the novel aspects of our work, which caused some questions about the knowledge increase and biological insights. We want to emphasize the technical and conceptual advances our study brings about. In this current submission, we have clearly highlighted the newly obtained biological insights that are supported by the new data mentioned above. While we do not want to claim that this study provides the final answer as to the working mechanism of WNT/CTNNB1 signaling, we are confident that our study pushes the technical and conceptual boundaries in the field. For your convenience, we summarize our main findings (and how they relate to current views) below:

1. Our study is the first to measure changes in the absolute concentration of endogenous CTNNB1 in its relevant functional pools in living human cells in response to physiological or oncogenic WNT signaling. This work is at the forefront of what is technically possible.

2. We use our newly acquired biophysical parameters to build a computational model that predicts the levels and subcellular distribution of CTNNB1 across the different cytoplasmic and nuclear pools. This wholistic picture reveals that, in addition to destruction complex inactivation, nuclear shuttling and nuclear retention are critical regulatory nodes within the pathway in both physiological and oncogenic settings.

3. Our data challenge current views of the core WNT/CTNNB1 signaling mechanism (i.e. turnover of CTNNB1 by the destruction complex) in three ways.

a. First, we show that upon pathway 35-40% of cytoplasmic CTNNB1 remains bound within a complex, regardless of whether signaling is activated via WNT3A (physiological stimulation), GSK3 inhibition (hyperactivation) or a CTNNB1^S45F^ point mutation (oncogenic activation). These data challenge (and essentially overthrow) a longstanding dogma in the field, namely that CTNNB1 accumulates as a free and monomeric protein in the cytoplasm upon WNT stimulation.

b. Second, we show that the cytoplasmic CTNNB1 complex undergoes a major reduction in size depending on the phosphorylation status of CTNNB1. The nature of destruction complex inactivation has been debated for almost a decade. Our data are consistent with a model in which the destruction complex is bound to the ubiquitination and proteasome machinery when CTNNB1 is phosphorylated, but in which this association is lost when CTNNB1 phosphorylation is blocked.

c. Third, despite the large size of this cytoplasmic complex (compatible with a multivalent destruction complex) our data do not support high-occupancy binding of CTNNB1 under physiological conditions. However, occupancy does increase when a phospho-dead form (CTNNB1^S45F^) is present. This has major impacts on how we conceptualize the workings of the CTNNB1 destruction machinery – especially in the context of cancer, since mutations in CTNNB1 (affecting occupancy) may have different biochemical consequences than mutations in APC (affecting multimerization and valency of the destruction complex itself).

Reviewer #1:CTNNB1 is a core component of canonical Wnt signalling that is frequently mutated in cancers. A constitutively active destruction complex (degradosome) binds and phosphorylates CTNNB1 earmarking it for proteasomal degradation, this complex is inactivated upon Wnt3a/GSK3β inhibition leading to CTNNB1 stabilisation and nuclear translocation. The authors have successfully employed CRISPR mediated endogenous tagging of CTNNB1 and determined its cellular concentration and diffusion dynamics in HAP1 cells, in both the cytoplasm and nucleus by live-cell imaging and analysis. They provide the relative subcellular CTNNB1 concentration for the nucleus and cytoplasm, like previous studies in other cell lines (Tan et al., 2012) and in *Xenopus* (Lee et al., 2003). In addition their results suggest CTNNB1 resides in slow moving complexes that persist upon Wnt but become slightly more mobile, these results are intriguing but raise several unanswered questions, such as whether these complexes represent the destruction complex (cytoplasm) or enhanceosome (nucleus). The work has been completed to a high standard but I have several concerns listed below.

We thank Reviewer 1 for their critical reading and evaluation of our manuscript and we are happy that they found our work to have been completed to a high standard. We will take the opportunity to address their concerns in more detail below. Before doing so, however, we want to point out two things. First, unlike previous studies (as the reviewer mentions), we perform the first absolute measurements of subcellular CTNNB1 concentrations in living human cells, for both the unbound and complexed pools of the protein. This, in turn, allowed us to build a computational model based on these measured biophysical parameters, which is a major improvement over existing analysis. Second, the shift in complex mobility upon Wnt stimulation is, in fact, far from ‘slight’ or ‘modest’ (see Reviewer 1, point 2).

1) The authors acknowledge significant cell-cell heterogeneity. This is particularly noticeable in Figure 4A upon Wnt3a and CHIR99021 treatment. Figure 4B suggests all cells are analysed regardless of heterogeneity and the only exclusion criteria mentioned in the methodology is cells with a cytoplasm of less than 10pixels. Figure 4C/D does not seem to reflect the variation observed in Figure 4A? What is the spread pre-normalisation before and after treatment? How is the relative increase in nuclear/cytoplasmic intensity affected by cell size? Nuclear and cytoplasmic area? This may affect the relative fold increase and the cytoplasmic area seems highly variable at the confluence of cells shown.

The reviewer poses several questions concerning the heterogeneity of the response to the WNT3A and CHIR99021 treatments previously shown in Figure 4. Of note, these data are now found in Figure 3 (WNT3A) and accompanying supplements and in Figure 8 supplement 1 (CHIR99021). To clearly address all the reviewer’s questions, we have organized them as follows and will address them in this order:

A) Apparent lack of variation in Figure 4C/D (now Figure 3B/C and Figure 8 supplement 1B/C)

B) Scaling of variation after normalization

C) Influence of cell size on CTNNB1 accumulation

A. Apparent lack of variation: We agree that Figure 4C/D (now Figure 3B/C) does not fully show the cell-to-cell heterogeneity that is visible in Figure 4A (now Figure 3A). The reason for this apparent discrepancy is that Figure 4C/D (now Figure 3B/C) shows the 95% confidence interval of the mean of all individually segmented cells. Due to the large sample size (n=155-400 cells per timepoint for each condition) the 95% confidence interval becomes very determined even though there is cell to cell variation. We have chosen this representation because it allows an easy comparison of the general trends of CTNNB1 subcellular accumulation between different conditions.

The cell-to-cell heterogeneity that Reviewer 1 refers to is better observed in Figure supplement 3A/B (previously Figure 4 supplement 2A/B and depicted in Author response image 1), where the data for all individual cells are plotted after 4 hours of WNT3A or CHIR99021 stimulation. From these graphs it is clear that there is indeed a large spread in the extent to which individual cells accumulate CTNNB1 at any given time point, but that almost all cells do react, which results in combination with the high number of cells to a very determined confidence interval of the mean in Figure 3B/C (previously Figure 4C/D).

B. Scaling of variation after normalization: In our previous submission, we indeed only showed normalized fluorescence data for Figure 4 (now Figure 3). Note that this normalization was needed to combine different biological experiments in one graph. However, when we plot an example from an individual biological experiment (i.e. part of the data of Figure 4 (now Figure 3)), we find that the spread in the unnormalized data is similar to the spread of the normalized data (Author response image 1). The reason for this is that for each imaging position (i.e. a specific field of view of several cells) we normalize to the average intensity in the relevant cellular compartment (i.e. cytoplasm or nucleus) prior to treatment – this allows normalization per position but preserves intensity differences within the position so as not to mask cell-to-cell heterogeneity.

**Author response image 1. sa2fig1:** Boxplots and individual datapoints from a single biological experiment (that is part of Figure 3 previously Figure 4) at 0 and 4 hours after treatment. The upper panels (raw) show the unnormalized intensity and the bottom panels (normalized) the normalized data. Note that although the axis is very different (0-750 compared to 0-5), the spread in the data is highly similar before and after normalization.

C. Influence of cell size: As per Reviewer 1’s suggestion, we have looked at the relation between cell size and WNT responsiveness (Author response image 2). When we divide the cells into 3 different size categories based on their cytoplasmic area (small: <500 pixels, middle: 500-1500 pixels, large: >1500 pixels), we do not find any meaningful differences in the cytoplasmic intensity, nuclear intensity and nuclear/cytoplasmic ratio after 4 hours of treatment between the different size categories.

Of course, there still could be more subtle effects of cell shape and size on CTNNB1 levels (and vice versa) and lots of other data can still be extracted from these live cell imaging experiments as well. We think this is a very interesting research direction, which will certainly get follow-up in our lab, but this is beyond the scope of the current study, which focusses on quantifying the subcellular behavior of CTNNB1 rather than on the causes underlying cell-to-cell heterogeneity.

**Author response image 2. sa2fig2:** Cells were divided into 3 size categories (small: <500 pixels, middle: 500-1500 pixels, large: >1500 pixels). Graphs show the normalized cytoplasmic intensity (top), normalized nuclear intensity (middle) and the nuclear/cytoplasmic ratio (bottom) for cells treated for 4 hours with BSA (control) 100 ng/ml WNT3A or 8 µM CHIR99021.

2) Using point FCS the authors determined two diffusion speeds corresponding to monomer and complexed CTNNB1 in both the nucleus and cytoplasm. A modest increase in cytoplasmic diffusion speed of complexed CTNNB1 was observed after Wnt3a (0.461μm2/s^-1^) but far from the speed of the monomer (14.9μm2/s^-1^) suggesting it remains complexed upon Wnt3a. In addition the fraction of complexed CTNNB1 (~40%) remains largely unaltered. Is the same true under CHIR299021 treatment? Point FCS samples a very small area of the cell cytoplasm/nucleus and therefore gives a small representation of the subcellular pool (which is likely heterogeneous), only a single point appears to have been analysed per-cell and within the 21 cells analysed clear outliers can be observed (Figure 6A/B), this has not been adequately discussed. What is the variation in diffusion measured at different points within a single cell? Some discussion has been made as to these complexes reflecting the destruction complex/proteasome or the enhanceosome but this really needs to be tested in order to make any conclusions about these observations. Especially as cytoplasmic complexes are maintained under Wnt conditions, this would challenge the notion that CTNNB1 disassociates from the destruction complex upon Wnt. Ideally endogenous tagging of other destruction complex components with a different flourophore would be done to address this, if these complexes do represent the destruction complex and remain bound after Wnt this would have significant implications for our understanding of complex inactivation and greatly enhance the manuscript.

We thank Reviewer 1 for their detailed and critical evaluation of our FCS experiments and we fully understand that our unexpected findings raise several questions. Again, we have subdivided the different discussion points so we can address each of these more clearly in that same order.

A) Diffusion speed and bound fraction under WNT3A and CHIR99021 stimulation

B) Heterogeneity of FCS measurements in living cells

C) Implications of our findings for the destruction complex

A. Diffusion characteristics of WNT3A vs CHIR99021: First, the reviewer is indeed correct that our data show that a large fraction of CTNNB1 remains complexed upon WNT3A stimulation, which directly challenges the textbook model for WNT/CTNNB1 signaling. The reviewer is also correct that this complex remains much slower than monomeric CTNNB1.

However, we want to point out that a 3-fold change in mobility (“a modest increase”, according to Reviewer 1) is, in fact, quite substantial: the speed of a protein is proportional to the cube root of its size (i.e. a 3-fold increase in speed means a 3^3^ = 27-fold decrease in size) for spherical particles. Therefore, the cytoplasmic complex present after WNT3A stimulation clearly has a different identity than the one present before.

In our manuscript, we have purposely refrained from estimating complex sizes, since the shape of the complex, molecular crowding and the meshwork of the cytoskeleton in the cytoplasm (and the chromatin in the nucleus) will also influence the speed of larger proteins. Therefore, the cube root relationship likely cannot be applied across our entire range of measurements. Nevertheless, in our new submission we now emphasize these findings as follows (page 18, lines 339-344): “Because a 3.5-fold change in speed would result in 3.53-change in size for a spherical particle (assuming Einstein-Stokes, see equation 7 in the material and methods section for details), this indicates that the size of the cytoplasmic CTNNB1 complex drastically changes when the WNT pathway is activated. Thus, although the fraction of CTNNB1 that resides in a complex remains the same, the identity of the cytoplasmic complex is quite different in unstimulated and WNT3A stimulated cells.”

At Reviewer 1’s request (and as also suggested by Reviewer 2, see our reply on below), we have now also performed FCS experiments upon CHIR99021 treatment (new Figure 7). Interestingly, we find that under these circumstances a large portion of CTNNB1 is also retained in a large cytoplasmic complex. The same is true in cells in which we introduced an S45F mutation in CTNNB1, resulting in constitutive activation of the pathway. Moreover, we find that the speed of this cytoplasmic complex is similarly increased as observed with WNT3A treatment, both upon mutational (S45F) or pharmacological (CHIR99021) inhibition of GSK3. We will further address the implications on the identity of the cytoplasmic CTNNB1 complex at point C on page 10 below.

B. Heterogeneity of FCS measurements in living cells: The reviewer is correct that there is variation in our data and our current datasets indeed contain some outliers. However, we do not think it is warranted (or informative for the reader) to discuss individual datapoints. That being said, we have always checked these outliers and we think we can explain some of them (For instance: In cases where the slow-moving fraction of CTNNB1 is very small, the diffusion times are less well defined. Other cases might be related to the intra- and intercellular heterogeneity the reviewer referred to). We have explicitly chosen to include (and show) all data for transparency and we have only excluded outliers if there was an obvious technical reason to do so (e.g. a clear misfit of the fitting model, which was very rare but has now been included in supplementary file 1).

In an ideal world, we would both have measured multiple points within the same cell and many more cells overall. However, an inherit limitation of point FCS is that it is quite laborsome and time-consuming. This is the reason why we only record a single point within either cellular compartment, especially since in case of WNT3A stimulation we have a small time window in which to record cells (as stated in the methods, all recordings were made 3-5 hours after stimulation and it takes approximately 10 minutes to measure the cytoplasm and the nucleus in a single cell when including all technical actions). While measuring multiple points per cell would indeed allow us to determine the intracellular variation, we found it more important to obtain sufficient independent measurements of both control and stimulated cells within the same experiment and our data demonstrate that this was the right choice: in our current set-up we are able to determine clear and statistically significant changes in the concentration, fractions of bound and unbound CTNNB1 and the relative diffusion speeds of CTNNB1 in unstimulated and stimulated cells despite including both inter- and intracellular variation. This has allowed us to focus on general differences – and meaningful biological changes – between different experimental conditions. The addition of new FCS data on S45F mutant or CHIR99021 treated cells (Figure 7), further strengthen our conclusions.

In summary, we only discuss biological differences that are upheld between different conditions in spite of inter- and intracellular variation and some outliers. In the Discussion section “Challenges and opportunities for fluorescence fluctuation spectroscopy techniques” we have now included a statement of the impact of using a single point measurement on the variability in our measurements to discuss this more adequately as requested by the reviewer: (page 37-38, lines 730-736): “In addition, point FCS is limited to a single position in the cell. Therefore, in addition to the intercellular differences in the WNT signaling response of individual cells, our measurements also sample intracellular heterogeneity caused by the presence of organelles and molecular crowding. Notwithstanding these limitations, we have been able to show that a large portion of CTNNB1 is present in a very large complex in both stimulated and unstimulated conditions and that this complex has a statistically and biologically significant different speed after WNT3A treatment.”.

C. Implications of our findings for the destruction complex: Reviewer 1 rightfully recognizes that our data challenge a long-standing dogma in the field, namely that CTNNB1 disassociates from the destruction complex upon WNT stimulation. Even without knowing the precise identity of the cytoplasmic complex, our data are very clear on this point: We find that 35-40% of cytoplasmic CTNNB1 is bound within a cytoplasmic complex, regardless of whether the WNT/CTNNB1 pathway is activated via WNT3A (physiological stimulation), CHIR99021 treatment (hyperactivation) or an CTNNB1S45F point mutation (oncogenic activation). This alone is an exciting novel biological insight that requires an update of the WNT signaling mechanism.

We also share the reviewer’s enthusiasm and interest in determining the precise nature of the slow-moving CTNNB1 complexes. As also suggested by the reviewer, we have begun the generation of double-tagged cell lines to further pinpoint the composition of the cytoplasmic complex. While we do already have a cell line in which we tagged GSK3B with a red fluorophore (mScarlet-i) in addition to our SGFP2-CTNNB1 allele (see Author response image 3), our genome editing approach has not yet been successful for other destruction complex components and we suspect that low expression levels are a bottleneck. Unfortunately, this is especially true for AXIN1, which would be the most relevant candidate to tag in order to confidently identify the destruction complex.

**Author response image 3. sa2fig3:** These images show a double-tagged line that has endogenously tagged SGFP2-CTNNB1 (left panel, green) and endogenously tagged mScI-GSK3B (middle panel, magenta).

In addition, we have also run into some of the technical challenges that are well known to complicate in vivo FCCS measurements, including “various sources of errors caused by instrumental or optical limitations such as imperfect overlap of detection volumes or detector cross talk” (Štefl et al., 2020). Thus, while ideal on paper for determining cooccupancy of CTNNB1 and other destruction complex components, in reality we have encountered roadblocks that prevent us from obtaining meaningful biological insights about the cytoplasmic complex identity using an FCCS approach. Although this may change in the future, at present these challenges unfortunately outweigh the time and resources we can invest in this area.

We do want to highlight, however, that by performing additional FCS measurements on the S45F CTNNB1 mutant and CHIR99021 treated cells (new Figure 7), as briefly discussed on page 8 above, we obtained more information on the identity of this cytoplasmic complex that further challenge the textbook model for WNT/CTNNB1 signaling: The fact that the increase in speed (i.e. reduction in size) of this cytoplasmic complex is consistent between WNT3A stimulation, CHIR99021 treatment and S45F mutational activation of the pathway, allows us to pinpoint that the size of this complex is correlated to the phosphorylation status of CTNNB1. We think that these biophysical parameters we measure are in agreement with a model in which the destruction complex is bound to the ubiquitination and proteasome machinery when CTNNB1 is being phosphorylated, but loses this association when CTNNB1 phosphorylation is blocked. See page 33, lines 627-636: “One interesting possibility, therefore, is that phosphorylated CTNNB1 is required for coupling the destruction complex to the ubiquitination and proteasome machinery. In fact, although not explicitly mentioned in the main text, supplementary table 1 of Li et al., 2012 shows that in HEK293 cells, which harbor no mutation in the core components of the WNT pathway, CTNNB1 was found to interact with subunits of the proteasome, whereas in the S45F-CTNNB1 mutant cell line Ls174T these interactions were not detected. In conclusion, although we do not directly determine its identity, our measured biophysical parameters of the cytoplasmic CTNNB1 complex are consistent with it representing a large, multivalent destruction complex that is coupled to the proteasome as long as CTNNB1 is being phosphorylated.”

3) The NandB analysis averages out monomeric and complexed CTNNB1 intensity across an image stack around a single ROI within each cell. The authors interpret Figure 6C to mean SGFP2-CTNNB1 is present as a monomer whether in a complex or not. This is based on the fact the relative brightness averages at 1.0 similar to a monomeric GFP control. However, the spread of relative brightness is large, and often less than <1 so a relative brightness of 1 cannot refer to a monomeric SGFP2-CTNNB1? Does cellular concentration affect relative brightness? If so, transiently expressed monomer and dimer GFP may not be the best controls. Aggregation is spatially homogeneous and limited by the diffusion rate of protein/complexes – which your FCS measurements suggest is consistent with a large complex. Thus a single average may not represent the diversity of protein complexes, eNandB could be used (Cutrale et al., 2019). As mentioned in point 3, like FCS, you are only sampling a small region of the cell, which may or may not contain a destruction complex for example. Super-resolution imaging techniques such a STORM or LLSM may help with visualisation of cell complex heterogeneity and give a different impression of complex occupancy. I don't think the NandB data is sufficient to say complexes don't exist that contain more than one SGFP2-CTNNB1 molecule.

We thank the reviewer for their comments and suggestions concerning our NandB data. Below we address all of them, and expect that this should alleviate their concerns. We also understand the reservation the reviewer might have, as our finding that the cytoplasmic CTNNB1 complex usually holds only one or very few CTNNB1 molecules seems to be in direct contrast with the notion that the CTNNB1 destruction complex a very large, phase-separated, multivalent complex. We also further discuss these implications on page 14-16.

For clarity we have subdivided the comment in several points and will address them in that order

(A) Brightness quantification and controls

(A1) Spread in the data

(A2) Relation number of particles and brightness

(A3) Spatial heterogeneity

(A4) eNandB and super resolution

(B) Implications of the NandB data on the occupancy of the destruction complex

A1. A1. Spread in the data: As with any technique, the measurements of relative brightness are subject to experimental variation, which is seen as the spread Reviewer 1 refers to. This is especially true for NandB measurements in living cells, compared to isolated proteins in solution. For normalization purposes, we set the average of the control measurements to 1. By definition, some relative-brightness values will therefore be <1 and others will be >1, both in our control sample as well as in our experimental sample.

However, when we look at the statistics of these measurements, we can see that the brightness of SGFP2-CTNNB1 (in either the cytoplasm or the nucleus and with or without WNT3A treatment) is not statistically different from the EGFP monomer control. This is in contrast to the EGFP dimer, the brightness of which is statistically different from the EGFP monomer (indicated with * in the table below). We are therefore confident that we can use these controls to draw conclusions on the multimerization status of SGFP2-CTNNB1. Although these statistics were already available in the Supplementary File 1 of the previous and current version of the manuscript, we have now added this in the main text as Table 3 to strengthen the conclusions within the body of the text.

A2.A2. Relation number of particles and brightness: Reviewer 1 is completely correct that high concentrations are incompatible with NandB measurements (as is true for any FFS technique). At high concentrations, fluctuations between the pixels in time will become very small, reducing the apparent (i.e. measured) relative brightness. To avoid this, we have been careful to use very low amounts of DNA in our transfections as described in the Materials and methods section (page 46-47, lines 924-927): “For low, FFS-compatible expression of control samples, HAP1WT cells were transfected with ~5 ng pSGFP2-C1, pEGFP (monomer), pEGFP2 (dimer) or pEGFP3 (trimer) and ~200 ng pBlueScript KS(+) per well with Turbofect, X-tremeGene HP or Lipofectamine 3000 the day before the experiment.”

To further appease the reviewer, we have plotted the relative (i.e. normalized) brightness versus the number of particles to verify that the levels obtained with our transfection are indeed within a range where they do not influence the relative brightness. This plot (Author response image 4) confirms that we do not see any dampening of the relative brightness even in measurements with the highest number of particles (200-250 particles). Thus, while the reviewer’s concern is theoretically valid, we have verified that the low plasmid-based transient expression of our controls is suitable for our analysis.

**Author response image 4. sa2fig4:** Figure displaying the relation between the particle number (x-axis) and the normalized brightness (y-axis). The number of particles is up to 3fold higher than on average in our endogenously tagged SGFP2CTNNB1 cell lines. However, we see no apparent effect of these higher particle numbers on the normalized brightness.

Spatial heterogeneity: Reading the full comment of reviewer 1, we assume that they meant to write that aggregation can be spatially heterogenous (rather than homogenous). In case we misunderstand this, we kindly ask the reviewer to clarify their concern so we can address it more accurately.

In our NandB analysis, we include much larger ROI’s than the point measurements we use in FCS. We apologize that this was not made sufficiently clear in our previous submission. In our new submission, we have added an additional panel showing typical NandB measurement ROI’s to the new Figure 5 (was Figure 6). We hope the reviewer agrees that a sufficiently large ROI of the nucleus and cytoplasm is included in the NandB analysis and therefore this covers the spatial heterogeneity more so than an FCS measurement.

A4. eNandB and super resolution: Reviewer 1 also suggests two additional approaches (eNandB and super resolution imaging) to investigate the brightness of SGFP2-CTNNB1 taking into account the potential heterogeneity of brightness between different complexes. One of us (Mark Hink) has in fact tried the eNandB method in our lab for known ratios of EGFP monomer, dimer, trimer or tetramer. However, the eNandB analysis was unable to extract these known brightness ratios and we were thus unable to reproduce the result of the original publication ((Cutrale et al., 2019)– also cited by Reviewer 1). As a control, the relative brightness obtained with our current NandB analysis did produce the expected average brightness in these experiments. So, while this approach looked promising, it did not work in our hands and we therefore decided to stick with the NandB analysis that provides a single average brightness, since this works reliably in our experience. We also think that our findings with the S45F mutation (see below) further confirm that we can detect brightness changes where they exist.

Although we agree that the use of super resolution imaging techniques offers exciting opportunities, the use of these techniques for brightness quantification is still very much in development and far less straightforward and commonplace than Reviewer 1 seems to imply.

B. Implications of the NandB data on the occupancy of the destruction complex: We agree with the reviewer that we cannot completely exclude that there are complexes that contain more than one SGFP2-CTNNB1 molecule. However, we think we have been careful not to overinterpret our findings (page 19-20, lines 367-369: “Because we found a large fraction of SGFP2-CTNNB1 to reside in a complex using point FCS (Figure 5C-D), this suggests that few, if any, of these complexes contain multiple SGFP2-CTNNB1 molecules.”).

In support of this interpretation, we also want to point out the following: if complexes with multiple SGFP2-CTNNB1 were present in our measurements, these would very quickly increase the average normalized brightness (as well as the apparent (i.e. measured) fraction that is in complex in FCS). This is due to the fact that in both FCS and NandB the intensity is multiplied by itself in the analysis, so it is the squared brightness of the fluorescent species that will contribute to the apparent brightness (and apparent fractions). This is best illustrated with simulations, of which we show some examples (see Author response table 1).

For example, if we assume that 50% of SGFP2-CTNNB1 resides in a complex, with a brightness (i.e. occupancy) of 1, 2 or 3, the average apparent brightness will increase from 1, to 1.67 and 2.5 respectively (Scenarios 1, 2 and 4 in Author response table 1). This immediately shows that it is unlikely that most of the complexes we measure with FCS contain more than one SGFP2-CTNNB1: in that case the average should increase compared to our monomeric EGFP control, which was not the case. This still holds true if we take into account that the brightness of the complex will also affect the apparent fraction of the complex (Scenarios 3 and 5 in Author response table 1).

Although this still leaves the possibility that some CTNNB1 complexes carry more than 1 SGFP2-CTNNB1, it is unlikely that even one complex would contain many CTNNB1 molecules. If, for example, only 1% of the complexes would carry 10 molecules of SGFP2CTNNB1, the apparent brightness would already increase to 1.83 (Scenario 6) – something we would have been able to detect in our NandB measurements. This scenario is also not consistent with the raw FCS traces, since the presence of a low number of complexes with very high brightness would be expected to give rather prominent spikes in the intensity traces, which is something we did not observe.

Summarizing, if highly saturated CTNNB1 complexes exist within the cell, these must be present in very low numbers such that we somehow missed them in both our FCS and NandB measurements. Even if this would be the case, this means that there is another cytoplasmic complex, which is much more abundant and has a low occupancy rate of SGFP2-CTNNB1 and which, moreover, changes in size upon WNT3A stimulation.

**Author response table 1. resptable1:** This table describes different scenarios of how different brightness species would impact the average apparent brightness.

Scenario 1	actual brightness	actual fraction (%)	apparant fraction (%)	average apparent brightness
monomer SGFP2-CTNNB1	1	50	50	-
complex containing 1 SGFP2-CTNNB1	1	50	50	-
				1

Scenario 2	actual brightness	actual fraction (%)	apparant fraction (%)	average apparent brightness
monomer SGFP2-CTNNB1	1	50	20	-
complex containing 2 SGFP2-CTNNB1	2	50	80	-
				1.67

Scenario 3	actual brightness	actual fraction (%)	apparant fraction (%)	average apparent brightness
monomer SGFP2-CTNNB1	1	80	50	-
complex containing 2 SGFP2-CTNNB1	2	20	50	-
				1.33
				
Scenario 4	actual brightness	actual fraction (%)	apparant fraction (%)	average apparent brightness
monomer SGFP2-CTNNB1	1	50	10	-
complex containing 3 SGFP2-CTNNB1	3	50	90	-
				2.5
				
Scenario 5	actual brightness	actual fraction (%)	apparant fraction (%)	average apparent brightness
monomer SGFP2-CTNNB1	1	90	50	-
complex containing 3 SGFP2-CTNNB1	3	10	50	-
				1.5
				
Scenario 6	actual brightness	actual fraction (%)	apparant fraction (%)	average apparent brightness
monomer SGFP2-CTNNB1	1	99	50	-
complex containing 10 SGFP2-CTNNB1	10	1	50	-
				1.83

Therefore, our data simply are not compatible with the prominent presence of a high occupancy complex (i.e. more than 1 or 2 – possibly 3 molecules of SGFP2-CTNNB1) under physiological conditions. We were equally surprised by this result (this is why we performed the simulations depicted in the Table shown above to begin with), as this is not what we would have expected from a large multivalent destruction complex, as appears to be supported by many current reports in the WNT field.

A possible explanation of this discrepancy is that CTNNB1 does reside in such multivalent complexes, but only very transiently. This would result in a situation where, on average, most of the potential binding sites for CTNNB1 are typically unoccupied – somewhat analogous to the wooden vanes in the paddle wheel of an old-fashioned watermill, or the rotating brushes on a Roomba vacuum cleaning robot: like the water and dust in these examples, CTNNB1 would continuously be scooped up (for phosphorylation) and dropped off (for degradation).

Interestingly, in the case of S45F mutation we do find an effect on the brightness (as shown in Figure 7), underscoring that our experimental approach is suitable for detecting these changes if and when they occur. This further suggests that in the presence of this oncogenic mutation the cytoplasmic complex does contain multiple SGFP2-CTNNB1 molecules.

**Author response image 5. sa2fig5:** Waterwheel model of CTNNB1 turnover.

4) The computational model relies on a number of assumptions determined in other studies that may not reflect the HAP1 cells used in this study. Lee et al., was performed in *Xenopus* and Tan et al., 2012 found a number of differences in their mammalian cell studies. Important information regarding the concentration of destruction complex components has also been omitted, this information is important for future comparisons of cell-type specific behaviours.

We appreciate the concern of the reviewer. Indeed, there are multiple parameters for which we had to rely on the literature. This underscores the need for studies like ours, which are dedicated to measuring absolute concentrations of proteins and protein complexes in different cell types. We also feel that measuring these parameters directly in living cells improves upon previous measurements in other settings (Lee et al., 2003; Tan et al., 2012), although ultimately such biochemical and biophysical approaches will also need to be compared.

With our approach we were able to obtain these specific parameters for endogenous CTNNB1, in a bound and unbound form, in both the cytoplasm and the nucleus of living human cells for the first time – which is of major importance for the field, as also remarked by reviewer 3. For all other parameters, however, we currently depend on other studies (such as (Lee et al., 2003; Tan et al., 2012)), to supplement our own data. Additional measurements of endogenous concentrations of other components in the future, will indeed help to compare cell-type specific behaviors and will also be needed for the continuous improvement of computational models.

We carefully checked all of the parameters we provided in our first submission and we could not detect and missing information. We apologize for any confusion that may have arisen with respect to details on the concentration of destruction complex components, but we can alleviate this concern: The reviewer is correct that these details were not provided, but this is because in our model we simply do not include individual destruction complex components or destruction complex assembly (in contrast to (Lee et al., 2003)). The reason for this decision was two-fold: First, we did not know the actual biological parameters of the individual components of the destruction complex in HAP1 cells (as Reviewer 1 also points out) and we wanted to minimize the number of assumptions we had to make based on other studies. Second, we realized we could simplify our computational model by only considering the fully formed destruction complex and solely estimating its concentration based on our own data obtained for the cytoplasmic complex as well as on the association of CTNNB1 K1=k1/k2 in our model is equal to K8=120 in the model of (Lee et al., 2003). We have now clarified this in the model description on page 54 lines 1056-1065:

“Our reaction (1) corresponds closely to step 8 in Lee et al. therefore, we use the value of the dissociation constant 𝐾_8_ = 120 nM from Lee et al. for our dissociation constant K1=K2K1. The concentration of the destruction complex is obtained from equation (1) under equilibrium conditions using equations (6), (8), (10), (11) and (14).

−𝑘_1_𝑥_1_𝑥_2_ + 𝑘_2_𝑥_3_ + 𝑏 = 0

The value of 𝑏 is assumed to be small compared to the two other terms, so we calculate the concentration of the destruction complex as:K1=K1x3x1=12062.591=82.4 nM. It was then verified in our interactive app that this value for the destruction complex is indeed consistent with the equilibrium conditions without WNT stimulation.”

For our purposes (i.e. describing the behavior of CTNNB1) this simplified computational model turned out to be sufficient. In the future, individual steps of destruction complex assembly and disassembly could be added to address different questions, but we think this will only be informative if actual biophysical parameters on complex assembly and complex component concentrations are obtained that are relevant for the system under study.

In this new submission, we now also include the (slightly updated) computational model as an interactive app (https://wntlab.shinyapps.io/WNT_minimal_model/) that allows users to change certain parameters of the model, to explore how this affects the levels of different species of CTNNB1. We hope this will make the computational model more accessible for non-computational biologists, who can intuitively “play with” the model. This might already provide readers with some insights into the regulatory nodes in their own specific experiments, similar to the new data we have included in the new Figure 8 of our manuscript. Here we strengthen the link between our data and the model by showing that the regulatory nodes responsible for nuclear retention (k8/k9) and nuclear shuttling (k6/k7) remain equally important during physiological (WNT3A stimulation) and oncogenic (S45F mutant) WNT/CTNNB1 signaling. In addition, we provide the raw scripts (https://osf.io/jx29z/ (WNT_minimal_model_v2.3.R)) so anyone wishing to adapt or add to any of the parameters or steps currently present can do so.

Reviewer #2:The manuscript by S.M.A. de Man et al., presents a study on the cellular response to Wnt activation and on the intracellular kinetics of β catenin (CTNNB1). The authors have developed cell lines expressing GFP reporters of CTNNB1 using CRISPR CAS9. They present different convincing controls on the specificity of the reporter and decided to analyze the temporal behavior of the best reacting clone. Then, they investigate the temporal evolution of fluorescent signal in the cell cytoplasm and nucleus upon Wnt signaling activation. They quantify the kinetics of the relocalization of CTNNB1 from the cytoplasm to the nucleus upon different strength of activation of the Wnt signaling and GSK3 inhibition. Using FCS, they identify that a dual diffusion model fit better the experimental data than a classical single diffusion model, suggesting the presence of complexes of different sizes. They measure the diffusion parameters and concentrations of the complexes in the nucleus and in the cytoplasm.Using a dynamical model, the authors reveal that, to recapitulate the experimental observations, the regulation of CTNNB1 upon Wnt signaling has to be controlled at three levels, the destruction complex, the nuclear transport and the binding affinity to the chromatin.Overall, the study is solid, presenting novel information on the kinetics of CTNNB1 during Wnt signaling. The results are consistent with the classical view on the regulation of β catenin during Wht signaling. I believe it is of interest for a general cell and developmental biology readership such as the one of eLife. I have few comments essentially on the methodology.

We thank reviewer 2 for their concise summary and positive assessment of our work. We will answer their comments in detail below. Please note that we have numbered them for clarity.

– The authors have designed a new cell line allowing for tracing the kinetics of betacatenin over time following Wnt signaling activation. They follow the relative changes in concentration in the nucleus and cytoplasm upon activation of Wnt signaling. Normalized changes render difficult to evaluate if the difference in the increase in the cytoplasm and the nucleus is due to a higher increase in the nucleus or simply due the absence of betacatenin in the nucleus at the onset of the process therefore enhancing the quantification. A non-normalized plot showing the increase in grey levels in the nucleus and cytoplasm should be added to complement the quantification and identify the differences between nuclear and cytoplasmic β catenin. It would also help the reader to compare with the results of concentrations extracted from the FCS.

We agree that the relation between the nuclear and cytoplasmic levels are relevant for the reader, since it allows discrimination between nuclear exclusion in the absence of WNT3A stimulation and/or nuclear enrichment in the presence of WNT3A stimulation. Note that in our first submission these data were already included in Figure 4E (now Figure 3D), which depicts a ratio of the raw (i.e. non-normalized) intensity values, with a nuclear/cytoplasmic ratio <1 indicating nuclear exclusion and a ratio >1 indicating nuclear enrichment. This indeed shows that before stimulation there is nuclear exclusion of CTNNB1 while upon treatment nuclear and cytoplasmic levels are very similar and perhaps only slightly enriched in the nucleus. The latter is in agreement with our FCS data, which also show a nuclear/cytoplasmic ratio greater than 1 following WNT3A stimulation or CHIR99021 treatment (new Figure 8I-J).

The remainder of the live-cell imaging analysis (Figures 4C-E, now Figure 3B-D) required normalization to be able to combine different biological experiments in a single graph. This was not true for the ratio in Figure 4E (now Figure 3D), since by dividing the two intensities to obtain the nuclear-cytoplasmic ratio, the ratio is independent of the raw values and can therefore be compared across different experiments without further normalization.

As per Reviewer 2’s suggestion we have now also added an additional Figure 3 supplement 3 with non-normalized (i.e. raw intensity) values for the cytoplasm and the nucleus from a single biological experiment, so the reader can more directly appreciate the nuclear exclusion at the beginning of the experiment, which may be more intuitive than the ratio depicted in Figure 4E (now Figure 3D).

– The response in figure 4 upon Wnt signaling activation and GSK3 inhibition are different (with the absence of a plateau in the case of GSK3 inhibition). The explanation of this difference is unclear as it is. I would suggest the authors to details a bit more their thoughts on the reason of the difference.Could this simply be that Wnt activation clusters just a subset of GSK3 at the membrane and that inhibition can reach higher level of depletion of GSK3 in the cytoplasm?

In our first submission we explained the different response upon WTN3A stimulation and CHIR99021 treatment in relation to AXIN2-mediated feedback: “The fact that intracellular SGFP2-CTNNB1 levels in the 8 μM CHIR99021 condition continued to accumulate, suggests that negative feedback, for example through *AXIN2* (Lustig et al., 2002), is overridden under these circumstances”. As AXIN2 mediated feedback is relevant to restore destruction complex functionality, it would not play a role in circumstances where functionality is already blocked by other mechanisms, such as direct inhibition of GSK3 activity. Therefore, we deem this a very likely mechanism for the observed discrepancy between the WNT3A and CHIR99021 treatment.

The alternative explanation provided by the reviewer is interesting, but less likely in our opinion: In the case of incomplete GSK3 inhibition by WNT3A, we would expect to observe a difference in the kinetics of CTNNB1 accumulation or a more apparent difference in the overall levels of CTNNB1 between these two conditions. However, it is possible that other feedback mechanisms, such as the removal of the WNT/receptor complex through endocytosis, does limit WNT3A-mediated GSK3 inhibition at later time points, which would be another explanation for the fact that WNT3A and CHIR99021 would start to differ at this time (>4h). Additionally, non-biological explanations may also play a role (e.g. the stability and denaturation of the purified WNT3A protein).

Please note that in our new submission, we have divided the WNT3A and CHIR99021 livecell imaging data over different figures (Figure 3 for WNT3A, Figure 8 supplement 1A- for CHIR99021) to accommodate other changes made to the manuscript. However, our observations and conclusions remain the same, with addition of receptor feedback now added within the text (page 26, lines 480-485*:”* However, in contrast to what is observed for WNT3A treatment, no plateau was reached at the highest concentration of CHIR99021 (8 µM). The fact that intracellular SGFP2-CTNNB1 levels in the 8 μM CHIR99021 condition continued to accumulate, suggests that negative feedback, for example through AXIN2 (Lustig et al., 2002) or through internalization of receptor complexes (Agajanian et al., 2019), is overridden under these circumstances”)

– How GSK3 inhibition treatment affects the FCS measurements, particularly concentrations and different complexes compositions? The differences with Wnt3 activation could provide additional information on the nature of the identified complexes.

At Reviewer 2’s request (and as also suggested by Reviewer 1, see our reply on page 7-8 above), we have now also performed FCS experiments upon CHIR99021 treatment (new Figure 7). Interestingly, we find that under these circumstances a large portion of CTNNB1 is also retained in a large cytoplasmic complex. The same is true in cells in which we introduced an S45F mutation in CTNNB1, resulting in constitutive activation of the pathway. Moreover, we find that the speed of this cytoplasmic complex is similarly increased as observed with WNT3A treatment, both upon mutational (S45F) or pharmacological (CHIR99021) inhibition of GSK3. Based on these additional experiments, we can now confidently conclude that the size of this cytoplasmic complex is therefore related to the phosphorylation status of CTNNB1: (page 24, lines 450-456): “Similar to the situation detected under physiological conditions (Figure 5D), we find a large fraction of SGFP2-CTNNB1^S45F^ to reside in a cytoplasmic complex (Figure 7A). As observed for physiological stimulation with WNT3A (Figure 5E), the speed of this complex is increased in SGFP2-CTNNB1^S45F^ compared to unstimulated HAP1^SGFP2-CTNNB1^ cells (Figure 7B). We find similar behavior when we block the GSK3 mediated phosphorylation of wild-type CTNNB1 using CHIR99021 (Figure 7C-D). The reduction in cytoplasmic complex size therefore must occur downstream of CTNNB1 phosphorylation.”.

To our opinion, this leaves only limited room for speculation as to its precise identity, as also mentioned in the discussion: (page 33, lines 627-636): “One interesting possibility, therefore, is that phosphorylated CTNNB1 is required for coupling the destruction complex to the ubiquitination and proteasome machinery. In fact, although not explicitly mentioned in the main text, supplementary table 1 of Li et al., 2012 shows that in HEK293 cells, which harbor no mutation in the core components of the WNT pathway, CTNNB1 was found to interact with subunits of the proteasome, whereas in the S45F-CTNNB1 mutant cell line Ls174T these interactions were not detected. In conclusion, although we do not directly determine its identity, our measured biophysical parameters of the cytoplasmic CTNNB1 complex are consistent with it representing a large, multivalent destruction complex that is coupled to the proteasome as long as CTNNB1 is being phosphorylated.”.

Interestingly, it were thus not the differences but rather the similarities observed between treatments and CTNNB1 activation status that provided additional information on the nature of the identified complexes and we thank Reviewer 1 for this suggestion.

– The dynamical model presented in the paper shows a non-monotonous change in the concentration of betacatenin in the cytoplasm after activation. This seems to be due to the kinetics of nuclear transport and does not seems to be present in the experimental observations. Can the authors comment on this point? Is there a way by modulating parameters associated to transport to suppress this discrepancy?

We thank the Reviewer for their detailed interest in this behavior of the model, which is explained by the way in which WNT/CTNNB1 signaling is ‘switched on’ computationally.

Originally, we modelled WNT addition by simultaneously and immediately changing all parameter values from the “off” state to those we determined for the equilibrium situation in the “on” state. As can be seen in Author response image 6, in this old situation the direct change of k8/k9 and k6/k7 causes an immediate drop in the cytoplasmic concentration (panel A) and an immediate jump in the nuclear concentration (panel C), which does not correspond with our live-cell imaging findings. This is also very clearly seen in the nuclear cytoplasmic ratio (panel E).

We reasoned that inside the cell, all of these parameters would gradually change over time. We therefore added a term to the equations (called “steep” in the new interactive model available via https://wntlab.shinyapps.io/WNT_minimal_model/) for both k6/k7 and k9/k8 as well as for k5/k4 to represent this gradual change. We fitted this term to match the timing of the change in nuclear/cytoplasmic ratio we observed with live-cell imaging. The resulting curves (panel B,D,F Author response image 6) are a better approximation of the experimental observations and we explain the inclusion of this term on page 56 lines 1101-1107: “It seems likely that such changes do not occur instantaneously. In our model we therefore allow a gradual rise in 𝑘_5_/𝑘_4_ and a gradual transition of the ratios of 𝑘_6_/𝑘_7_ and 𝑘_9_/𝑘_8_ from WNT ‘off’ to the WNT ‘on’. In our model this is included by setting a parameter (“Steep”) that indicates the time after application of WNT the transition from WNT ‘off’ parameter values to WNT ‘on’ parameter values is complete. The value that gives a good approximation of the experimentally observed concentration curves is Steep = 150 minutes (Figure 6 panels B-F).”

However, the change in the model is indeed still not constant as keenly spotted by reviewer 2. Further tweaking of delays in the activation of k9/k8 or k6/k7 and the rise of k5/k4 (the same term is now used for all) could perhaps further smoothen this line. However, we don’t think this will provide new biological information on the underlying regulatory mechanisms of WNT signaling and have therefore decided not to pursue this further at the moment.

**Author response image 6. sa2fig6:** Comparison of old model without steep (A,C,E) and new model with steep (B,D,F). In the old model the change in k6/k7 and k9/k8 is immediate at the addition of WNT3A. This results in a sudden loss of cytoplasmic CTNNB1 (**A**) and gain in the nucleus (**C**) as is also apparent from the nuclear-cytoplasmic ratio (**E**). Our live imaging experiment shows that the changes. This excluded the slow adjustment of the k6/k7 and k8/k9 (called steep in the newest version of the model). Without this adjustment the parameters in the nucleus adjust immediately resulting in an immediate change in the nuclear-cytoplasmic ratio (**C**) and the TCF-bound fraction (**E**). This also leads to a dip in the cytoplasmic concentrations of CTNNB1 (**B**) and immediate increase in cytoplasmic concentrations (**C**).

– Finally, the model is consistent with the experimental observations but the authors di not checked with any type of perturbation how the model would compare with the experiments. For instance, how does the model would compare with experiments in the case of GSK3 inhibition, or when nuclear transport is affected. Adding a perturbation case would significantly strengthen the connection between model and experiment and the message of the manuscript.

We thank the reviewer for this excellent suggestion. We have now added and extensively analyzed a perturbation case in our new Figure 8.

The mechanism of nuclear translocation of CTNNB1 remains to be resolved. In fact, CTNNB1 does not contain nuclear import or export signals and can translocate independently of classical importin and exporter pathways (Fagotto et al., 1998; Wiechens and Fagotto, 2001; Yokoya et al., 1999). Hence, we could not specifically perturb nuclear/cytoplasmic shuttling (which represents the first critical node in our model, namely k6/k7). Similarly, we empirically found that we could not fully perturb nuclear retention (which represents the second critical node in our model, namely k8/k9) with the small molecule inhibitors currently available (e.g., ICG-001). Therefore, we decided to perturb the third critical node in our model, namely destruction complex activity (k3), via two independent means of blocking the GSK3-mediated inhibition of CTNNB1 (CHIR99021 treatment and S45F mutation).

Using this perturbation case, we were able to establish that upon destruction complex inactivation we still need similar changes in k6/k7 and k8/k9 as in the case of WNT3A stimulation to match the predictions of the model with our experimental observations. These findings underscore the critical nature of all three regulatory nodes in both physiological and oncogenic conditions. We think that these data indeed strengthen the link between our experimental and computational findings and we hope the reviewer agrees.

Importantly, these perturbation experiments also revealed that the changes in nuclear retention and nuclear/cytoplasmic shuttling are directly correlated with the CTNNB1 phosphorylation status, which opens up the exciting possibility that the N-terminal posttranslational modification status of CTNNB1 plays an active and important role in regulating its subcellular distribution (see page 36 lines 690-695: “The fact that direct inhibition of GSK3 mediated phosphorylation of CTNNB1 results in the same behavior, indicates that the phosphorylation status of CTNNB1 plays a critical role. This further emphasizes the importance of posttranslational modifications and conformational changes in CTNNB1 for its subcellular localization and function (Gottardi and Gumbiner, 2004; Sayat et al., 2008; Valenta et al., 2012; van der Wal and van Amerongen, 2020; Wu et al., 2008).”).

Reviewer #3:1) As I state below the paper is carefully done using a difficult and sophisticated biophysical technique, FCS to assess the changes in β catenin diffusion within the cell following Wnt signaling. So it passes the test on being an original piece of work executed well. However what has been learned is quite limited. A few interactions, such as the slow diffusion in the cytoplasm can be interpreted several ways. It is very helpful to have concentrations in the nucleus and cytoplasm for β catenin for future modeling. They could have tried to use single cross correlation with labeled APC or axin or the proteasome to derive more important information about the path through the destruction sequence. But that may be too hard to ask for at this stage. They could have combined their measurements with appropriate mutants or knockouts. I come down close to the line, high on the importance of the problem and the methods and execution; lower on the current take home lesson.

We thank the reviewer for their careful assessment of our work and we are happy that they consider our work to be original, well executed and high on importance. At the same time, we also understand that the reviewer might have been left wondering what has been learned in terms of take-home lesson. We feel that in our first submission these take-home messages may have gotten lost amidst the complex biophysical details.

Here we will touch upon several points and suggestions made by Reviewer 3. For clarity we have subdivided them and will address them in this order.

A) New take-home messages from the revised manuscript

B) Fluorescence cross-correlation spectroscopy

C) S45F mutant and CHIR99021 perturbation case

A. New take-home messages from the revised manuscript: We recognize that our previous version of the manuscript did not allow us draw strong conclusions about the nature and identity of the complexes we identified, but we were careful not to overinterpret our results. We are confident that we have remedied this situation in our new submission. In particularly, we have included two new figures (Figure 7 and Figure 8) based on two distinct perturbation cases (CHIR99021 treatment and S45F mutation of CTNNB1, explained in more detail below). The combined live-cell imaging, FCS measurements and NandB analyses now allow us to summarize the novel findings and take-home messages of our study as follows, as also highlighted on page 1 and 2 of this rebuttal:

1. Our study is the first to measure changes in the absolute concentration of endogenous CTNNB1 in its relevant functional pools in living human cells in response to physiological or oncogenic WNT signaling. This work is at the forefront of what is technically possible.

2. Reviewer 3 acknowledges the relevance of obtaining biophysical data for modelling purposes. In fact, we think that these particular points of our study should not be underestimated: We use our newly acquired biophysical parameters to build a computational model that predicts the levels and subcellular distribution of CTNNB1 across the different cytoplasmic and nuclear pools. This wholistic picture reveals that, in addition to destruction complex inactivation, nuclear shuttling and nuclear retention are critical regulatory nodes within the pathway in both physiological and oncogenic settings.

3. Our data challenge current views of the core WNT/CTNNB1 signaling mechanism (i.e. turnover of CTNNB1 by the destruction complex) in three ways.

3.1 First, we show that upon pathway 35-40% of cytoplasmic CTNNB1 remains bound within a complex, regardless of whether signaling is activated via WNT3A (physiological stimulation), GSK3 inhibition (hyperactivation) or a CTNNB1^S45F^ point mutation (oncogenic activation). These data challenge (and essentially overthrow) a long-standing dogma in the field, namely that CTNNB1 accumulates as a free and monomeric protein in the cytoplasm upon WNT stimulation.

3.2 Second, we show that the cytoplasmic CTNNB1 complex undergoes a major reduction in size depending on the phosphorylation status of CTNNB1. The nature of destruction complex inactivation has been debated for almost a decade. Our data are consistent with a model in which the destruction complex is bound to the ubiquitination and proteasome machinery when CTNNB1 is phosphorylated, but in which this association is lost when CTNNB1 phosphorylation is blocked.

Third, despite the large size of this cytoplasmic complex (compatible with a multivalent destruction complex) our data do not support highoccupancy binding of CTNNB1 under physiological conditions. However, occupancy does increase when a phospho-dead form (CTNNB1^S45F^) is present. This has major impacts on how we conceptualize the workings of the CTNNB1 destruction machinery – especially in the context of cancer, since mutations in CTNNB1 (affecting occupancy) may have different biochemical consequences than mutations in APC (affecting multimerization and valency of the destruction complex itself).

We don’t want to presume that our current study is the be-all and end-all to the WNT/CTNNB1 signaling mechanism, but we are confident that it offers solid evidence to challenge different aspects of the textbook model and, in doing so, provides a significant conceptual advance as well as multiple leads for new and testable hypotheses.

B. Fluorescence cross-correlation spectroscopy: We also share the reviewer’s enthusiasm and interest in determining the precise nature of the slow-moving CTNNB1 complexes. As also suggested by Reviewer 3 (and reviewer 1, see page 7 for their comment and page 10 for our reply), we have begun the generation of double-tagged cell lines to further pinpoint the composition of the cytoplasmic complex. While we do already have a cell line in which we tagged GSK3B with a red fluorophore (mScarlet-i) in addition to our SGFP2CTNNB1 allele (Author response image 7), our genome editing approach has not yet been successful for other destruction complex components and we suspect that low expression levels are a bottleneck. Unfortunately, this is especially true for AXIN1, which would be the most relevant candidate to tag in order to confidently identify the destruction complex. C.

**Author response image 7. sa2fig7:** These images show a double-tagged line that has endogenously tagged SGFP2CTNNB1 (left panel, green) and endogenously tagged mScI-GSK3B (middle panel, magenta).

In addition, we have also run into some of the technical challenges that are well known to complicate in vivo FCCS measurements, including “various sources of errors caused by instrumental or optical limitations such as imperfect overlap of detection volumes or detector cross talk” (Štefl et al., 2020). Thus, while ideal on paper for determining cooccupancy of CTNNB1 and other destruction complex components, in reality we have encountered roadblocks that prevent us from obtaining meaningful biological insights about the cytoplasmic complex identity using an FCCS approach. Although this may change in the future, at present these challenges unfortunately outweigh the time and resources we can invest in this area.

C. S45F mutant and CHIR99021 perturbation case: At Reviewer 3’s suggestion we have also added FCS and NandB data obtained under mutant conditions. For this, we introduced an S45F mutation in CTNNB1 resulting in constitutive activation of the pathway. This mutation is one of the earliest identified oncogenic mutations in CTNNB1 (Morin et al., 1997), and it removes the CSNK1 priming phosphorylation site on CTNNB1 that is needed for sequential phosphorylation by GSK3, and thus blocks its proteasomal degradation (Amit et al., 2002; Liu et al., 2002). In addition, we also obtained new FCS and NandB data after CHIR99021 treatment as suggested by Reviewer 1 and 2 (see page 7 and 21 for their comments). Interestingly, we find that under both these circumstances a large portion of CTNNB1 is also retained in a large cytoplasmic complex (new Figure 7). Moreover, we find that the speed of this cytoplasmic complex is similarly increased as observed with WNT3A treatment, both upon mutational (S45F) or pharmacological (CHIR99021) inhibition of GSK3.

These additional FCS measurements, have given us more information on the identity of this cytoplasmic complex that further challenge the textbook model for WNT/CTNNB1 signaling: The fact that the increase in speed (i.e. reduction in size) of this cytoplasmic complex is consistent between WNT3A stimulation, CHIR99021 treatment and S45F mutational activation of the pathway, allows us to pinpoint that the size of this complex is correlated to the phosphorylation status of CTNNB1. We think that these biophysical parameters we measure are in agreement with a model in which the destruction complex is bound to the ubiquitination and proteasome machinery when CTNNB1 is being phosphorylated, but loses this association when CTNNB1 phosphorylation is blocked. See (page 33, lines 627-636): “One interesting possibility, therefore, is that phosphorylated CTNNB1 is required for coupling the destruction complex to the ubiquitination and proteasome machinery. In fact, although not explicitly mentioned in the main text, supplementary table 1 of Li et al., 2012 shows that in HEK293 cells, which harbor no mutation in the core components of the WNT pathway, CTNNB1 was found to interact with subunits of the proteasome, whereas in the S45F-CTNNB1 mutant cell line Ls174T these interactions were not detected. In conclusion, although we do not directly determine its identity, our measured biophysical parameters of the cytoplasmic CTNNB1 complex are consistent with it representing a large, multivalent destruction complex that is coupled to the proteasome as long as CTNNB1 is being phosphorylated.”.

Of note, we used the same perturbation to determine the effect of an oncogenic CTNNB1 mutation on brightness (and thus occupancy) of the cytoplasmic complex. While our data are not compatible with the prominent presence of a high occupancy complex (i.e. more than 1 or 2 – possibly 3 molecules of SGFP2-CTNNB1) under physiological conditions (quite contrary to what would be expected from a large multivalent destruction complex, as appears to be supported by many current reports in the WNT field), in the case of S45F mutation we do find an effect on the brightness, suggesting that in the presence of this oncogenic mutation the cytoplasmic complex does contain multiple SGFP2-CTNNB1 molecules.

A possible explanation of this discrepancy is that CTNNB1 does reside in such multivalent complexes, but only very transiently. This would result in a situation where, on average, most of the potential binding sites for CTNNB1 are typically unoccupied – somewhat analogous to the wooden vanes in the paddle wheel of an old-fashioned watermill, or the rotating brushes on a Roomba vacuum cleaning robot: like the water and dust in these examples, CTNNB1 would continuously be scooped up (for phosphorylation) and dropped off (for degradation).

2) The support for the somewhat limited conclusions is strong as it is.

We are happy that Reviewer 3 appreciates our thoroughness. We hope that in our answer under Major comment #1 above we have also convinced the reviewer that the conclusions, novel insights and conceptual advances presented in this work are equally strong as their experimental support and relevant for not just the WNT signaling field, but also a broader scientific community.

3) There are some technical issues. There is some concern with the FCS data. Itself. Figure 5F and 5G are of some concern. The curve doesn't drop to 1 at long correlation time (>100ms) and there are big fluctuations in the region of short correlation times (<0.1 ms). This could be due to the very long time course (120s) used in the experiment. Have the authors tried to image the same spot multiple times in short interval (etc 10s), or try to analyze 10s sub-trace of the original long trace to see if the conclusions hold? This type of error could influence the calculation of the diffusion coefficient of complexes of CNNTB1. They also affect the quantification of concentration. In line 352-353 the authors mentioned the nuclear concentration of CNNTB1 increases 2.1 fold based on FCS measurement, which is smaller than the fluorescent intensity change. Is this the result of errors such as this?

The reviewer touches upon several aspects of the FCS technique, which remains challenging in a complex environment such as a living mammalian cell, for example due to heterogeneous autofluorescence, cell movement etc. The reviewer refers to “errors”, but we think these points rather refer to inherent properties of this technique. For this reason, we have devoted an entire section in the discussion to discuss the challenges and opportunities of this technique (updated and modified in this submission).

We do thank the reviewer for all their questions and suggestions, because we do agree that they are relevant and important to consider. To clearly address them we have organized them as follows and will address them in this order:

A) Concerns relating to the offset of the autocorrelation curve

B) Concerns relating to large fluctuation at short correlation times

C) Concerns relating to duration of the FCS acquisition

D) The discrepancy between fluorescence and concentration

A. Offset of the autocorrelation curve: The reviewer expects the autocorrelation curve to drop to 1 after 100ms. First, please note that the decay seen between 10ms and 1000ms is still part of the slow diffusion component. In these experiments, long correlation times therefore are >1000ms, rather than 100ms. Second, there is indeed a small offset in the autocorrelation curve at these long correlation timepoints (>1000 ms). However, this offset is approximately 0.002 in the specific example depicted in Figure 5F-G (now Figure 4FG). If we look at all of our measurements, the offset of the curves is generally between 0.000 and 0.004, which is not expected to have a big influence on either the particle number or the diffusion times. We also show another curve without any offset from one of our datasets (Author response image 8), to show the difference with the autocorrelation curve shown in the manuscript that was chosen without cherry picking.

**Author response image 8. sa2fig8:** Autocorrelation curves from the same experiment with a small offset of 0. 002 (**A**) as original example in Figure 5G (now Figure 4G) or without offset (**B**).

For our analysis we removed any measurements with significant intensity drift Material and Methods, page 48, line 951-954: “Intensity traces with significant photobleaching, cell movement or focal drift were excluded from further analysis (see supplementary file 1 – tab FCS measurements and fitting). From other traces a portion of the trace with minimal (less than 10%) intensity drift or bleaching was selected to generate autocorrelation curve (AC).” This also means that even though our measurements are 120 seconds in total, analysis is sometimes performed on shorter portions of the trace. In the analyzed portion of the trace some minor movement, drift or bleaching (i.e. less than 10% per minute) might still be present and we think that this could be responsible for causing a minor offset of the autocorrelation curve. We therefore used a model that includes a term describing this offset (G∞), (Materials and methods page 48, line 956-957: “G∞ accounts for offset in the AC for example by intensity drift”). This ensures that the amplitude (used to fit the number of particles for the slow and fast fraction of SGFP2-CTNNB1) is not affected by the small offset in the autocorrelation curve. Therefore, the concentrations and diffusion kinetics from our analysis, should not be affected. We hope this addresses the reviewer’s concern.

B. Large fluctuations at short correlation times: The reviewer also wonders about the “big fluctuations in the region of short correlation times (<0.1 ms)”. At these short correlation times we observe two phenomena: first we have afterpulsing, which is a APD (avalanche photodiode detector) artefact that is specific for every detector, and second, we are dealing with the photophysical properties of the specific fluorophore (e.g. fast blinking). Indeed, the noise around this part of the curve seems quite large. However, this is quite normal for autocorrelation curves from a single APD detector, due to the statistics (i.e. the number of events measured by the APD) associate with these very short time intervals due to the pulsed laser excitation and gated detection (which results in the detection of only 1 event per 50ns).

When we look at the fitting of these two events, described by F_trip_ and τ_trip_(see Material and methods page 48 line 958-961: “F_trip_ and τ_trip_ describe the fraction of molecules in the dark state and the relaxation of this dark state respectively. Of note, in this case, F_trip_ and τ_trip_ account both for blinking of the fluorescent molecules and for the afterpulsing artefact of the APD”.) We see that the residual noise of the experimental curve is fitted nicely around the analysis model as can also be judged by the residual plot in Figure 4G, indicating that the fitted parameters are reliable. In addition, the typical times that are fitted as τ_trip_ are in the 1-50 µs range, whereas our first diffusion time is significantly removed at around 1 ms. therefore, the fitting of the particle numbers and diffusion times is not easily influenced by the noise at these very early timescales. We would welcome any additional questions or remarks if the reviewer’s concern remains.

C. Duration of the FCS acquisition: Although there are people in the field that prefer multiple short measurements, our own experience, with our experimental setup, is that at these low endogenous concentrations the autocorrelation curves we obtain with these very short traces are very noisy. In our hands, one single measurement with more photons tends to give more determined parameters than statistically combining multiple short measurements.

However, as per the reviewer’s suggestion we have compared the analysis of a large portion of the trace (90s) with shorter, 10s sub-traces. As can be seen in Author response image 9, the results of the 10s traces are indeed noisier, which results in less well-defined confidence intervals of the individually fitted parameters. This, in turn, contributes to the variation between the estimated particle numbers (N) and fraction of slow CTNNB1 (F2) for the different 10s measurements. The analysis of a larger portion of the FCS measurement results in more determined parameters, and also corresponds well to the average of the10s traces. Therefore, we expect that general application of shorter analysis intervals, will yield the same conclusions about concentration and diffusion coefficients as we get from the current analysis strategy. Thus, we have no reason to believe that our data are more error-prone than if we were to perform shorter measurements.

**Author response image 9. sa2fig9:** Comparison of analysis of short and long FCS traces. Autocorrelation curves were made from nine 10s portion of the measurement (trace), or a single 90s trace from the same FCS measurement of SGFP2-CTNNB1 and subsequently fitted with the same fitting model. The graphs depict the fitted values (dots) and confidence intervals determined by the goodness of fit in the FCS analysis software (lines) for the particle number (N, panel **A**) and _the fraction of slow component_ (F2, panel **B**) of the 10s and 90s.

D. Discrepancy between fluorescence and concentration: As for the specific example that Reviewer 3 refers to in line 352-353 page 14-15 lines 276-278 in our new submission:

“This increase was smaller than expected from fluorescence intensity measurements (Figure 3B), for which we currently have no explanation (Figure 5 supplement 1).”: We have, of course, been puzzled ourselves by the difference between the FCS measurement and the fluorescent intensity change and therefore described this discrepancy for clarity. We had also prepared a Figure supplement that addresses this issue, but by mistake this was not include (although it was available at BioRXIV). We have now corrected this mistake and included this supplement in our new submission. As outlined in the figure supplement, we have verified that the discrepancy is not related to bleaching or quenching. As detailed above, we also do not think that any of the reviewer’s listed concerns (the offset of the autocorrelation curve at long correlations times, the big fluctuations in the region of short correlation times or the time intervals used for FCS measurements) is responsible for this discrepancy. However, if the reviewer would have further suggestions, we would be happy to look into these, as we currently do not have an explanation ourselves.

In summary, we appreciate the critical feedback from the reviewer on this challenging biophysical technique and we hope to have alleviated their concerns.

[Editors’ note: what follows is the authors’ response to the second round of review.]

Essential Revisions:1) For free β-catenin, the authors have calculated the theoretical diffusion coefficient based on the measured diffusion coefficient of SGFP2 and the size difference between them. But it is possible that β-catenin will interact with other proteins in a specific or non-specific manner, which will make the β-catenin diffuse substantially slower than SGFP2. The free β-catenin diffusion coefficient will affect the fitted value of the slow diffusion component. I wonder if the authors could fit the autocorrelation curve directly with 2 component model without assuming a free β-catenin diffusion coefficient. On the other hand, if a predetermined free β-catenin diffusion coefficient is preferred to fit the autocorrelation curves, maybe it is better to make a ΔARM-β-catenin and use the diffusion coefficient of that protein as the diffusion coefficient of the fast component, or measure the diffusion coefficient of SGFP2-β-catenin directly in the cell extract. In addition, some of the free receptors have a surface diffusion coefficient in the range of 0.1 um2/s. Have the authors considered the possibility that the β-catenin complexes bind some large structure inside the cell, like ER membrane, showing a very slow diffusion coefficient?

This comment concerns two different points, which we will therefore address separately:

1) the free CTNNB1 diffusion coefficient and (2) the diffusion coefficient of the slower CTNNB1 pool.

1. Indeed, we made multiple assumptions to calculate the theoretical diffusion coefficient of free SGFP2-CTNNB1 (as detailed in lines 1097-1103 of the Materials and methods: “As the temperature (T), dynamic viscosity (η) and Boltzmann’s constant (kB) are equal between SGFP2 and SGFP2‑CTNNB1 measurements, the expected difference in diffusion speed is only caused by the radius (r) of the diffusing molecule assuming a spherical protein.”, already present in the previous version). In reality, some of these theoretical assumptions (such as a perfect spherical nature of the particle) will not be met completely. This is an inherent limitation of the approach (and, perhaps, quantitative biology in general). Therefore, the question as to what will happen if we do not fix the first (fast) component is valid. In fact, it had already occurred to us while we were performing our measurements. At the time, we noticed that fitting our two-component model without a fixed first component resulted in similar diffusion coefficients for the first (fast) component, but we had not formally analyzed this. As suggested by the reviewer, we have now reanalyzed the data from Figures 4 and 5. Specifically, we have fitted these measurements with a two-diffusion component model in which both the first diffusion coefficient and the second diffusion coefficient were unfixed. The first diffusion coefficient fitted by this model is highly similar (median 14.8 μm^2^/s) to the fixed value we determined theoretically for free SGFP2-CTNNB1 (14.9 μm^2^/s).

This analysis is included in the revised manuscript (new Figure 4 supplement 1).

As for the reviewer’s concern that “The free β-catenin diffusion coefficient will affect the fitted value of the slow diffusion component”, we have verified that the other parameters (concentration, second diffusion coefficient and slow fraction) are also not substantially different (nor any better) from those obtained using the model with the fixed first component (Author response image 10, top row shows the results using the unfixed two-component fitting model and bottom row shows the results with our first-fixed two-component fitting model). Therefore, our previous interpretations and conclusions remain unaltered.

**Author response image 10. sa2fig10:** SGFP2-CTNNB1 parameters fitted by a two-diffusion component model where T1 is free (A-C) or fixed at 14.9 um2/s (D-F). (**A** and **D**) Concentration of SGFP2-CTNNB1 (**B** and **E**) Fraction of slow SGFP2-CTNNB1 (**C** and **F**) Speed of slow SGFP2-CTNNB1.

We do want to point out that every degree of freedom in the fitting model, will increase the number of possible solutions that can fit the raw curve, leading to less well-defined parameters, especially in more noisy data. Therefore, we still prefer to fix the first component and we think this is justified as we have now shown that the data do not change.

We appreciate the additional suggestions from the reviewer to determine the free diffusion coefficient of monomeric SGFP2-CTNNB1. With both suggested alternatives, CTNNB1 could retain some interactions with other proteins, potentially generating multiple diffusing species (i.e. monomeric and complexes). In addition, the cellular context (i.e. cytoskeletal meshwork and crowding) might be quite different in a cellular extract, which would result in significant differences in the free diffusion coefficient measured ex cellulo as opposed to in cellulo. It is our honest opinion that none of these suggested alternatives is necessarily better, and each control will have different pros and cons. We chose our current control (SGFP2 transfected in HAP1 cells) because it allows us to measure the same fluorophore, which is by definition monomeric, in the same cellular (i.e. biologically relevant) context as our SGFP2-CTNNB1.

Taking everything together, we think the solid fit of our model to our data (Figure 4), and our reanalysis showing similar results when using unbiased fitting (new Figure 4 supplement 1) support the fact that our use of the estimated theoretical diffusion coefficient for free monomeric SGFP2-CTNNB1 is a valid simplification.

This point is also addressed in the main text of the revised manuscript, which now reads as follows:

Line 263-272: “We therefore tested the fit of a two-component model. To this end, we deduced the theoretical diffusion speed of monomeric, unbound SGFP2‑CTNNB1 to be 14.9 µm2/s. This theoretical speed was confirmed by fitting an unbiased two-component model to our experimental data (Figure 4 supplement 1). To limit variability due to noise in the measurements, we proceeded with the two component model in which the first diffusion component was fixed to the theoretically determined diffusion speed of monomeric SGFP2‑CTNNB1 (14.9 µm2/s) and with the second diffusion component limited to slower speeds compatible with point-FCS imaging (see Materials and methods for details). This model provided good fits for our autocorrelation curves obtained in both cytoplasmic and nuclear point FCS measurements (Figure 4G).”

Line 1119-1126: “To validate the obtained first diffusion coefficient of 14.9 µm2s^-1^ for SGFP2‑CTNNB1, the data were tested with an unfixed two-component model where both the first and the second diffusion coefficient were fitted (shown in Figure 4 supplement 1). The following limits were set; G∞ [0.5-1.5], N [0.001, 500], τtrip [1*10-6-0.05 ms], τdiff1 [0.5-10 ms], τdiff2 [10-150 ms]. This resulted in a median diffusion time for the first component of 14.8 µm2s^-1^ (Figure 4 supplement 1), which was in line with our calculated diffusion coefficient of 14.9 µm2s^-1^. All analyses were performed with the two-component model with the fixed first component to reduce variability.”

2. Regarding the speed of the slower SGFP2-CTNNB1 pool: the reviewer is correct that our slow diffusion coefficient is not far removed from those measured for certain membrane associated proteins. Typically, photobleaching is a significant issue in point FCS for proteins stably located at membrane, which we did not observe in our measurements (see lines 1062-1063 and 1075-1076 for how we control for this, already present in the previous version). That being said, we did consider the possibility that the slow diffusion coefficient could be explained by dynamic association with cellular structures. In fact, in our first submitted version of this manuscript (28-05-2020-RA-BX-*eLife*-59433), we actually addressed this issue specifically, but in streamlining the manuscript for our new submission (11-01-2021-RA-*eLife*-66440), this detail got lost. In the current revision, we have reintroduced this possibility in the section on “Challenges and opportunities for fluorescence fluctuation spectroscopy techniques”. (Line 818-828: “Moreover, the measured diffusion coefficients do not reveal the identity of the complexes. Previous studies have shown that a significant pool of CTNNB1 is associated with destruction complex components in presence and absence of WNT signaling (Gerlach et al., 2014; Kitazawa et al., 2017; Li et al., 2012), and it is therefore likely that at least part of the slow fraction of CTNNB1 we measure does indeed represent this destruction complex bound pool. As we discuss above, association and dissociation of the destruction complex with the proteasome offers one potential explanation for the different diffusion coefficients measured in the cytoplasm in WNT ‘ON’ and ‘OFF’ conditions. However, in both conditions other processes, such as transient association with intracellular structures (e.g. vesicular membranes or cytoskeletal components), could contribute to the diffusion coefficients we observe.)”

If the reviewer has any additional concerns or knowledge specifically concerning ER-binding, of which we are not currently aware, we would be happy to include any literature that has described this.

2) The model that the author proposed has some logical flaws. In line 1042, equation (9), the authors write the equation w(k4 χ2 – k5 x4) as the mathematical form of Wnt activation, where w is applied to both k4 χ2 term and the k5 x4 term. The χ2 term represents the active destruction complex, and the x4 term represents the inactive destruction complex. This means Wnt will regulate both forms of the destruction complex the same way. And the ratio of χ2/x4 will not change after Wnt pathway activation. This way to model is logically flawed. The authors do show a response of the model after Wnt treatment, but this is because the destruction complex also has a substrate-bound fraction: x3. This fraction is not controlled directly by Wnt input.

We respectfully disagree that our model contains some logical flaws and we would like to refute this remark as follows:

The reviewer correctly notes that in equation (9)

dx4dt=w(k4x2−k5x4) the parameter *w* multiplies both the terms k4x2 (representing the free/active destruction complex, DC) and k5x4 (representing the inactivated destruction complex, DC*). Note that these two terms are also present in equation (7), which describes the change in the concentration of DC (χ2).

The reviewer is not correct, however, that the ratio of χ2/x_4_ will not change after Wnt pathway activation, because both x_4_ and *w* itself are equal to zero in the absence of WNT (as was already mentioned in the methods section, paragraph ‘model description’). We suspect that this misunderstanding stems from the fact that in our previous submission we did not place sufficient emphasis on the fact that the term *w* only comes into play when the WNT pathway is switched on and we apologize for any confusion this may have caused. To remedy this, we now make explicit mention of the term w in Table 4, Table 5 and Figure 6 in addition to the methods section (lines 1172-1174). It is also addressed in the main text as follows:

Line 409-411: “When WNT is present in the system, we describe the inactivation of the destruction complex (‘DC*’) by DVL through the parameter w (see Materials and methods section model description).”

Put differently, our minimal model assumes that WNT acts by reducing the concentration of the active destruction complex (DC in Figure 6A, χ2 in Table 4). We do this by opening up a pathway through which the active destruction complex is removed only in the presence of WNT (WNT OFF, *w*=0; WNT ON, *w*=1). In the model, this pathway is represented by equation (9), which describes the increase in the concentration of the inactivated destruction complex (DC* in Figure 6A, x_4_ in Table 4) and through the same terms in equation (7), which describes the reduction in the concentration of the active destruction complex (χ2). The term k_4_x_2_ thus describes the appearance of inactivated destruction complex (DC*, x_4_) at the cost of the free destruction complex (DC, χ2).

Regarding x_3_, the reviewer correctly notes that in our model WNT does not directly control destruction complex bound to CTNNB1 (CB*-DC in Figure 6A, x_3_ in Table 4). A similar indirect mode of action of WNT treatment is described in the much more elaborate model from the Kirschner group (Lee et al., 2003), where the action of WNT is simulated by the formation of activated Dishevelled, which reduces the concentration of the active form of the destruction complex, resulting in an increase in CTNNB1 levels. Thus, in both our model and in the model from the Kirschner group, the concentration of the free/active destruction complex is reduced, thereby increasing CTNNB1 levels.

It is important to note that the removal of free/active destruction complex (i.e. an increase in DC* at the expense of DC) in and by itself does not necessarily lead to an increase in CTNNB1. In our model, and also indirectly in the Kirschner group model, this increase is a consequence of *(i)* the constant rate at which the destruction complex is released upon CTNNB1 degradation (i.e. k_3_ is not affected by WNT) and *(ii)* our experimental observation that the DC-bound phosphorylated CTNNB1 (CB*-DB in Figure 6A, x_3_ in Table 4) has the same concentration in the absence and presence of WNT 62.5 nM, as per the median concentration of the slow component in our FCS measurements

3) It's better to use SGFP2 dimer and trimer as control instead of EGFP, in Figure 5H and 7E/F. The dimer and trimer constructs are important control for the average size and the percentage of bright fluorescent proteins will vary for different fluorescent proteins. I have no idea how SGFP2 compares to EGFP.

We understand this point from the reviewer’s perspective. We insufficiently realized that the similarities between SGFP2 and EGFP would not be obvious to all and we apologize for not having been more explicit about the properties of SGFP2. As we will explain below, it is highly comparable to EGFP and therefore the use of EGFP controls is justified.

The superGFP2 (SGFP2) variant was developed by colleagues in our institute, who are experts in fluorescent protein biology. It is based on the EGFP template but has been optimized for maturation in mammalian cells (Kremers et al., 2007). SGFP2 and EGFP have the exact same size (239 amino acids) and only differ by 4 amino acids (Author response image 11). As described by Kremers et al., the spectral and biochemical characteristics of EGFP and SGFP2 are highly similar with a slightly lower extinction coefficient (-27%) and a slightly higher quantum yield (+7%) for SGFP2. When other factors are also taken into consideration (including the improved maturation and a lower fraction of dark state kinetics for SGFP2) this results in a slightly lower molecular brightness (-5%) of monomeric SGFP2 in comparison to monomeric EGFP (as can be judged from Figure 5H and Figure 7E/F).

To further illustrate this, we have performed additional NandB measurements of the two monomeric proteins in HeLa cells (Author response image 11). These data confirm that for cellular measurements any differences between EGFP and SGFP2 are well within the 95% confidence range and thus too small to have any significant impact on our measurements and interpretations. Based on the above, we believe our controls in Figure 5H and 7E/F are beyond dispute.

**Author response image 11. sa2fig11:** Differences between EGFP and SGFP2. (**A**) protein alignment of EGFP (top) and SGFP2 (bottom) showing the 4 amino acid difference. (**B**) Independent measurements in an additional mammalian cell line of monomeric EGFP and SGFP2 brightness. The same control constructs, measured in HAP1 cells, were already included in figures 5 and 7. The molecular brightness of sGFP2 is slightly lower, but well within the 95% confidence interval of the biological measurements. Any differences between the proteins are thus too small to be significant.

Please also note that because the maturation of SGFP2 is improved compared to EGFP (which was an important reason behind our choice to tag CTNNB1 with SGFP2 in the first place), this should only aid the detection of SGFP2 dimers, trimers or higher order oligomers in our SGFP2-CTNNB1 cells. Put differently: It is highly unlikely that we miss them. The fact that we only detect an increase in molecular brightness in the S45F mutant cell line, thus further strengthens our conclusion that few if any cytoplasmic complexes contain multiple CGFP2-CTNNB1 molecules under physiological conditions.

We have added the following text to the Materials and methods section to explain our choice for SGFP2 considering its fluorescent properties and the relation between EGFP and SGFP2 in the NandB controls:

Line 886-887: “SGFP2 was chosen for its favorable brightness, maturation and photo-stability (Kremers et al., 2007).”

Line 1130-1136: “SGFP2 and EGFP are highly similar in sequence (with only 4 amino acid changes) and in spectral and biochemical characteristics. SGFP2 has a slightly higher quantum yield (+7%), lower extinction coefficient (-27%), and enhanced protein expression and maturation compared to EGFP (Kremers et al., 2007). The resulting brightness of monomeric SGFP2 in comparison to monomeric EGFP is slightly lower (-5%). In cellular measurements this difference is within the biological and technical variation and therefore SGFP2 and EGFP controls can be considered synonymous in these experiments.”

4) The data on the concentrations of β-catenin in the cytoplasm and nucleus before and after Wnt signaling and after inhibition of GSK3 are a major advance. However, I share the opinion of previous reviewer 2 (comment 1) that too much attention is paid to relative differences and not enough to absolute differences. There is strong evidence in the literature that one function of the destruction complex is to retain β-catenin in the cytoplasm (e.g. https://pubmed.ncbi.nlm.nih.gov/21471006/), and thus reduce the amount that can enter the nucleus. Their quantitative imaging (e.g. Figure 3A) reveals this quite clearly and combined with their FCS data suggest that the cytoplasmic pool includes this "bound" β-catenin and that this bound pool remains after Wnt stimulation. I would suggest making this clearer and putting it in the context of earlier work on cytoplasmic retention.

As was already mentioned in the manuscript, we use our measurements and absolute differences to determine the appropriate biophysical parameters for our computational model.

We have now further emphasized this point in the discussion:

Line 602-603: “Importantly, this allowed us to measure hitherto unknown biophysical parameters of WNT/CTNNB1 in individual living human cells for the first time.”

We also thank the reviewer for their additional suggestions to incorporate these absolute values in different contexts, which we have done in our revision throughout:

First, we realize that we did not include absolute numbers and values for our S45F mutant experiments to the same extent as for wildtype CTNNB1 in the presence and absence of WNT. This has been remedied in the Results section as follows:

For the differences between physiological (WNT3A) and oncogenic (S45F) WNT/CTNNB1 signaling, we have more clearly emphasized the absolute concentrations in the Results section. Lines “514-525: With FCS and NandB we quantified the accumulation of CTNNB1 levels of mutant SGFP2‑CTNNB1^S45F^ (Figure 8B, Figure 8 supplement 2A) and wild-type SGFP2‑CTNNB1 upon CHIR99021 treatment (Figure 8 supplement 2B-C). Both exceeded the levels observed with physiological WNT3A stimulation (Figure 3A-C, Figure 5A). Specifically, the absolute concentration of SGFP2‑CTNNB1^S45F^ in the cytoplasm (median 351 nM, 95% CI 276-412) exceeded that of SGFP2‑CTNNB1 in WNT3A treated cells (median 221 nM, 95% CI 144-250). In the nucleus, the concentration of SGFP2‑CTNNB1^S45F^ reached 429 nM (median, 95% CI 387-481), as opposed to 240 nM (median, 95% CI 217-325) for SGFP2‑CTNNB1 in WNT3A treated cells as a result of losing its priming phosphorylation site. While this further increase in concentration is evident, it should be noted that in both the cytoplasm and the nucleus CTNNB1 levels thus rise less than 2-fold in an oncogenic setting compared to WNT3A treatment.”

For additional parameters we have similarly added more quantitative description of the data in the main text. For example, Line 535-538: “The fraction of bound SGFP2‑CTNNB1 in the nucleus was comparable between our HAP1^SGFP2‑CTNNB1(S45F)^ mutant cell line (median 0.29, 95% CI 0.27-0.33) (Figure 8E), 8 μM CHIR99021 (median 0.38, 95% CI 0.29-0.46) (Figure 8F) and WNT3A (median 0.32, 95% CI 0.30-0.34) (Figure 5D) treated wild-type HAP1^SGFP2‑CTNNB1^ cells.”

Second, in the discussion we put our FCS measurements in proper context of the existing literature on cytoplasmic retention:

Lines 668-677: “The substantial fraction of slow diffusing CTNNB1 that remains upon physiological and oncogenic stimulation of the pathway, is also consistent with the previously proposed role for cytoplasmic retention by the destruction complex (Krieghoff et al., 2006; Roberts et al., 2011; Yamulla et al., 2014). These studies also show that both nuclear and cytoplasmic retention have an important role in determining the subcellular distribution of CTNNB1. Moreover, cytoplasmic retention of CTNNB1 can contribute to reducing downstream pathway activation even in the presence of oncogenic APC mutations (Kohler et al., 2008; Li et al., 2012; Schneikert et al., 2007; Yang et al., 2006), which further highlights the importance of this process.”

5) One model currently in the field is that the key regulated step in β-catenin destruction is not its phosphorylation but instead its transfer to the E3 ligase, an event that may be controlled by GSK3 phosphorylation of APC (e.g. https://pubmed.ncbi.nlm.nih.gov/22682247/;. https://pubmed.ncbi.nlm.nih.gov/26393419/; https://pubmed.ncbi.nlm.nih.gov/32129710/). In talking about this, the authors state: "Our data clearly show that a substantial fraction of CTNNB1 in the cytoplasm remains bound upon pathway stimulation (Figure 5D). This is not predicted by any of the above-mentioned models and challenges the long-held view that mainly monomeric CTNNB1 accumulates." However, in my mind the retention of high MW complexes after Wnt signaling seems like it is consistent with the model in the manuscripts cited above. Further, while model cartoons (I use them myself) indicate accumulation of monomeric β-catenin in the cytoplasm, in fact what seems more likely is that monomeric β-catenin that accumulates after Wnt signaling rapidly enters the nucleus and is "trapped" as part of high MW transcriptional complexes like those their data appear to document, and thus monomeric cytoplasmic levels may be relatively low. I think a more nuanced description of these data would be useful.

We have rewritten this part of the discussion to be a bit more nuanced as per the reviewer’s suggestion. Lines 663-668: “Our data clearly show that a substantial fraction of CTNNB1 in the cytoplasm remains bound upon pathway stimulation (Figure 5D). This is not captured by textbook models and cartoons that typically still depict the view that mainly monomeric CTNNB1 accumulates. It is, however, consistent with the notion of the destruction complex remaining intact and transfer to the E3 ligase, rather than phosphorylation and release of CTNNB1 itself, being the key controlled step (Li et al., 2012; Pronobis et al., 2015; Schaefer et al., 2020).”

6) One major controversy in the field is the nature of the active destruction complex and its multimeric state. Some data (e.g. https://pubmed.ncbi.nlm.nih.gov/26124443/; https://pubmed.ncbi.nlm.nih.gov/26393419/; https://pubmed.ncbi.nlm.nih.gov/29641560/) suggest large multimeric assemblies while in vitro studies have suggested much smaller assemblies (e.g. https://pubmed.ncbi.nlm.nih.gov/32297861/). It was quite surprising that the authors did not review this literature-in fact the latter paper isn't even in the reference list. Their data, while not definitive, certainly has something to say about this discussion. That having been said, however, I do NOT agree with Reviewer 1 that extending this work to visualize tagged destruction complex components is essential. Of course, this is a very important next step, but this manuscript is already rich in novel data. The reduction in "size" of the potential destruction complexes after Wnt treatment is also consistent with recent observations in *Drosophila* (https://pubmed.ncbi.nlm.nih.gov/29641560/)

We thank the reviewers for their support of our work. We too look forward to future follow up studies and extension of this work by ourselves and others. Indeed, the multimerization state of the active destruction complex is one of the main outstanding questions in the field. We have included the suggested references in our revised discussion to better reflect the current state of the literature:

Line 678-686: “We show that the cytoplasmic CTNNB1 complex in WNT3A or CHIR99021 treated cells as well as in S45F mutant cells has an over 3-fold increased mobility compared to control cells. Therefore, while the diffusion coefficient is still very low (indicating that the remaining complex is still very large), this implies it is a vastly different complex than that observed in the absence of WNT stimulation. The precise nature of these complexes remains unknown, but could be consistent with a reduced destruction complex size after WNT treatment, as also recently observed in *Drosophila* for AXIN complexes (Schaefer et al., 2018), or with the formation of inactivated destruction complexes (‘transducer complexes’) in response to WNT/CTNNB1 pathway activation (Hagemann et al., 2014; Lybrand et al., 2019).”

Line 631-641 “In addition, our NandB data indicate that most of the slow diffusing CTNNB1 complexes contain one or very few SGFP2‑CTNNB1 molecules in either the absence or presence of WNT3A stimulation. In view of the above-mentioned destruction complex oligomerization and its presumed multivalency this finding was quite unexpected. Several mechanisms could explain this apparent discrepancy. On the one hand, destruction complex multimerization at endogenous levels might be more subtle than previously thought. For example, quantification of AXIN polymerization in vitro showed that even at exceedingly high concentration (24μM), AXIN polymers typically contained only 8 molecules (Kan et al., 2020). On the other hand, even if the multivalent destruction complex offers multiple CTNNB1 binding sites, occupancy at any one time might be low, due to the continuous and high turnover of CTNNB1.”

7) I think Reviewer 2 comment 2 (previous submission) and the authors response should be more clearly laid out in the manuscript. While many of us said for years that Wnt signaling "turns the destruction complex off", it is now abundantly clear that it only "turns the destruction complex down". For example, in both *Drosophila* and in cultured mammalian cells complete loss of GSK3 or APC leads to much higher levels of β-catenin than are seen after Wnt signaling.

We appreciate this suggestion and we have added a section in the discussion (Differences and similarities in physiological and oncogenic WNT signaling) to specifically address the differences between oncogenic and physiological signaling.

Line 762-78 “As discussed above, several behaviors of CTNNB1 are conserved between the different modes of stimulation. For instance, WNT3A treatment, GSK3 inhibition and the oncogenic S45F mutation all result in (1) increased overall levels of CTNNB1 (2) a substantial fraction of CTNNB1 in a faster, albeit still very large complex in the cytoplasm (3) increased nuclear accumulation of CTNNB1 and (4) increased retention of CTNNB1 in nuclear complexes. Our computational modelling further confirms that in addition to regulation of CTNNB1 turnover – either by removal of activated destruction complex or through inhibition of phosphorylation and ubiquitination -, nuclear shuttling and nuclear retention are equally important regulatory nodes in oncogenic (CHIR99021 treatment or S45F mutation) and physiological (WNT3A stimulation) signaling.

However, the absolute levels of CTNNB1 and the resulting transcriptional activation are distinct in each of these conditions: Cells treated with a GSK3 inhibitor continue to accumulate CTNNB1 after 4 hours, when WNT3A treated cells reach a plateau. The latter is likely due to the fact that negative feedback mechanisms kick in, such as reconstitution of the destruction complex by AXIN2 or internalization of WNT-bound receptor complexes (Agajanian et al., 2019; Lustig et al., 2002), both of which function upstream of GSK3. Alternatively, it could reflect the notion that physiological WNT signaling does not turn the destruction complex off completely, but rather “turns it down”, as our NandB data support the fact that under physiological conditions the destruction complex itself provides a surplus reservoir of CTNNB1 binding sites that may only become occupied when WNT signaling is hyperactivated. As a combined result, GSK3 inhibition or S45F mutation of CTNNB1 can result in higher total intracellular levels of CTNNB1. Indeed, concentrations of SGFP2‑CTNNB1 in S45F mutated cells exceed those in WNT3A treated cells in both the cytoplasm (1.6-fold) and the nucleus (1.8-fold). This subtle increase in CTNNB1 levels is likely amplified at the transcriptional level (Jacobsen et al., 2016), consistent with the well-known fact that constitutive activation of the pathway through different mechanisms, including APC mutation, results in higher pathway activation than physiological stimuli.”

8) The fact that their Number and Brightness data suggest few destruction complexes bind multiple β-catenins is very surprising. It doesn't fit my preconceptions and that is OK! I like the idea that usually throughput is rapid enough that multiple β-catenins don't accumulate (waterwheel model). It is surprising that Wnt signaling doesn't alter this but those are the data.

We appreciate this positive feedback and we still share the reviewers’ surprise. We appreciate their support for our waterwheel model and have incorporated this idea more clearly in the main text.

Line 639-645 “On the other hand, even if the multivalent destruction complex offers multiple CTNNB1 binding sites, occupancy at any one time might be low, due to the continuous and high turnover of CTNNB1. In this respect, the CTNNB1 bindings sites in the destruction complex could be envisioned to act similar to the wooden vanes in the paddle wheel of an old-fashioned watermill: like the water in the analogous example, CTNNB1 would be continuously scooped up (for phosphorylation) and dropped off (for degradation).”